# Towards large-scale single-shot millimeter-wave imaging for low-cost security inspection

Liheng Bian [1,2,6] ✉, Daoyu Li [1,6], Shuoguang Wang [3,4,6], Chunyang Teng[3], Jinxuan Wu[1], Huteng Liu[3], Hanwen Xu[1], Xuyang Chang [1], Guoqiang Zhao[3], Shiyong Li [3,5] ✉ & Jun Zhang[1] ✉

Millimeter-Wave (MMW) imaging is a promising technique for contactless security inspection. However, the high cost of requisite large-scale antenna arrays hinders its widespread application in high-throughput scenarios. Here, we report a large-scale single-shot MMW imaging framework, achieving low-cost high-fidelity security inspection. We first analyzed the statistical ranking of each array element through 1934 full-sampled MMW echoes. The highest-ranked elements are preferentially selected based on the ranking, building the experimentally optimal sparse sampling strategy that reduces antenna array cost by one order of magnitude. Additionally, we derived an untrained interpretable learning scheme, realizing robust and accurate MMW image reconstruction from sparsely sampled echoes. Last, we developed a neural network for automatic object detection, and experimentally demonstrated successful detection of concealed centimeter-sized targets using 10% sparse array, whereas all the other contemporary approaches failed at such a low sampling ratio. With the strong detection ability and order-of-magnitude cost reduction, we anticipate that this technique provides a practical way for large-scale single-shot MMW imaging.

Security check at public places such as airports and railway stations requires effective personnel surveillance techniques to prevent rising-concerned terrorist attacks[1]. However, conventional surveillance techniques have limited utility in practice, as metal detectors can only detect metallic weapons and explosives, X-ray machines expose individuals to harmful ionizing radiation[2], infrared imaging systems[3,4] are sensitive to environmental disturbance, and visible-light cameras cannot see through clothing. A practical security check system typically relies on the combination of these different techniques. Consequently, the security system suffers from a complex inspection workload and poor efficiency of passage. In this context, millimeter-wave (MMW) imaging has emerged as a promising alternative, due to its unique advantages of high-resolution and penetrable imaging ability while being safer for human exposure compared to X-ray machines[5]. MMW imaging provides the possibility of an all-in-one high-throughput security screening system with high resolution, high penetrability, and strong safety.

Existing MMW imaging systems are generally categorized into two types: passive and active. Passive systems[6] rely on detecting the naturally occurring MMW radiation emitted by targets. However, they

[1]State Key Laboratory of CNS/ATM & MIIT Key Laboratory of Complex-field Intelligent Sensing, Beijing Institute of Technology, 100081 Beijing, China. [2]Guangdong Province Key Laboratory of Intelligent Detection in Complex Environment of Aerospace, Land and Sea, Beijing Institute of Technology, Zhuhai 519088, China. [3]Beijing Key Laboratory of Millimeter Wave and Terahertz Technology, Beijing Institute of Technology, 100081 Beijing, China. [4]Academy for Network & Communications of CETC, Key Laboratory of Hebei Province on Unmanned System Intelligent Telemetry & Telecontrol Information Technology, 050050 Shijiazhuang, China. [5]Tangshan Research Institute of Beijing Institute of Technology, 063007 Tangshan, China. [6]These authors contributed equally: Liheng Bian, Daoyu Li, Shuoguang Wang. ✉e-mail: bian@bit.edu.cn; lisy_98@bit.edu.cn; zhjun@bit.edu.cn

often encounter limited imaging quality due to low radiation contrast between targets and the environment. On the other hand, active systems[5,7–11], including single-input-single-output (SISO) and multiple-input multiple-output (MIMO) configurations[12], leverage additional MMW illumination and capture scattered electromagnetic waves to achieve superior imaging resolution compared to passive systems. While a full SISO array can achieve high-quality imaging, it requires a large number of antenna elements that are prohibitively expensive. Consequently, a mechanically scanning linear array with range migration algorithm (RMA)[7] has emerged as a more economical alternative. However, the resulting images may suffer from blur even if the subjects jiggle during data acquisition[13], which degrades the subsequently concealed object detection accuracy[14]. The fully electronic MIMO arrays demonstrate the advantage of requiring fewer elements while enabling data collection through a single snapshot but need time-consuming high-dimensional data processing[12,15–19]. To sum up, the existing MMW imaging systems suffer from the inevitable tradeoff among imaging quality, cost, and running efficiency.

The sparse array synthesis (SAS)[20–25] technique is emerging as a cost-effective alternative to minimize manufacturing expenses of the fully sampled arrays. However, the state-of-the-art SAS methods struggle to synthesize large arrays[25], making them less applicable for security check scenarios where a large aperture is usually required. The next challenge lies in the reconstruction methods for sparse array-based systems. The existing compressive sensing (CS) techniques[26–29] may fail under low sampling ratios with large-scale arrays. In recent years, the deep learning (DL) approaches have been introduced to either enhance pre-reconstructed images[30,31], or directly deal with the inverse scattering problem[32–35], yielding successful reconstruction on specific datasets. However, DL methods heavily rely on training datasets, potentially leading to poor generalization for untrained or marginal distributions.

To tackle the above challenges, here we present a low-cost, large-scale single-shot MMW security inspection framework, which is capable of successfully detecting concealed centimetre-sized targets. First, we collected a set of real-captured echoes using a scanning-based full-sampled antenna array, and then analyzed the statistical importance ranking map of different array elements. Based on the ranking map, we experimentally derived a statistically optimized sampling strategy to sparsely select antenna elements at low sampling ratios (e.g., 10% and 25%). Second, we proposed an interpretable untrained learning approach, which ensures robust MMW reconstruction from sparsely sampled echoes. Unlike conventional DL imaging methods[30–35], this physics-informed learning strategy operates without the need for dataset training, ensuring case-specific optimization with strong robustness. Compared to the existing CS-based approaches, it yields high-fidelity reconstruction with up to 3.4 dB PSNR improvement at low sampling ratios. The combination of statistics-based sparse sampling and untrained learning results in a significant reduction in the cost of antenna array by up to an order of magnitude. Third, we developed a neural network that achieved successful centimeter-sized concealed target detection at a 10% sampling ratio, while other contemporary approaches failed. To conclude, this work leverages the statistical sparse prior and the emerging model-learning joint optimization framework, providing new insights into the development of low-cost and efficient MMW security inspection systems.

## Results

### Statistically optimized sparse array

In contrast to prevailing handicraft-designed arrays, our approach capitalizes on the statistical prior afforded by a large-scale dataset. To this end, we first gathered real echoes using a full-sampled antenna array. We have complied with all pertinent ethical regulations and secured informed consent from all fourteen volunteers. These volunteers carried five types of concealed objects, including knives, phones,

wrenches, pistols (metal replica model), and Explosive Powdered Material (EPM, employing silica gel with similar MMW absorption properties in our experiments). They were positioned around 0.4 meters in front of the array, as depicted in Fig. 1a. Throughout the data collection process, these volunteers were instructed to maintain a steady position. Under such an experiment configuration, we obtained a total of 1934 echo samples from ten volunteers, constituting the MMW imaging dataset. The dataset comprises samples with three dimensions: vertical, horizontal, and frequency, each containing $430 \times 186 \times 50$ voxels.

To analyze the distribution of 3D echo, we extracted a 2D slice of echo at the center frequency along the frequency dimension. This particular slice was chosen as it provides the most comprehensive representation of the echo characteristics within this frequency band. Then we calculated the average data of all the obtained 2D slices to generate a 2D average echo, which reflects the characteristics of the 3D averaged echo and facilitates investigation into the statistical importance ranking of elements in the 2D antenna array. Further, we extracted the average amplitude map $\bar{A}$ and inverse phase gradient map $\bar{P}$ (the reciprocal of phase's gradient) of the cross-section, as shown in Fig. 1b. A larger value in $\bar{A}$ indicates that this element has more significance in the antenna array, and vice versa. As for the phase item, a continuous target distribution can result in a flattened phase gradient distribution of the array ("Analysis on the statistical maps of MMW echoes" and Supplementary Note 4). Consequently, we suggest that attention should be focused on the portion of the array with a smaller phase gradient distribution (larger values in $\bar{P}$). A larger value in the product of $\bar{A}$ and $\bar{P}$, denoted as $\bar{M} = \bar{A} \times \bar{P}$, signifies greater importance of the element within the antenna array. We set $n = 1 \rightarrow N$ as orders of each sorted element in $\bar{M}$. A smaller $n$ means higher statistical importance. Given the statistical importance ranking $\bar{M}$ and a uniform random function $r(n)$ ranging from 0 to 1 (Fig. 1c), we selected the elements $n$ of $r(n) > S$, where $S$ is a hyperparameter to control the sparsity of sparse patterns ("Statistically sparse sampling").

We conducted a series of simulations to test the performance of the reported sparse sampling strategy. The root mean square error (RMSE), peak signal-to-noise ratio (PSNR), and structural similarity (SSIM)[36] were employed to quantify reconstruction accuracy. We compared both the 2D maximum value projections along the range direction and 3D results reconstructed by various imaging algorithms. The dynamic range of the reconstruction results was set to 20 dB. Images retrieved by RMA from full sampling echoes were regarded as references. Figure 1d demonstrates that there exists an optimal sparsity for a certain sampling ratio. For instance, the hyperparameter $S = 0.8$ corresponds to the optimal sparsity of 10% sampling ratio, while $S = 0.5$ corresponds to 25% sampling ratio. The statistics in Table 1 reveal that statistically sparse sampling yields average enhancements of 20% in RMSE, 2 dB in PSNR, and 0.22 in SSIM across all reconstruction methods compared to random sampling. Both visual and quantitative comparisons (Fig. 1e, f and Table 1) validate the superiority of the reported statistically sparse sampling on reconstruction accuracy.

### The untrained learning reconstruction

To ensure reconstruction fidelity, we employed a physical constraint to regulate the output of the neural network, following the interpretable untrained learning strategy[37,38]. By updating the parameters $\theta$ of the neural network $f_\theta$, the untrained reconstruction minimizes the loss between $\mathcal{H}(f_\theta(z))$ and the measurement $E_s$, where $\mathcal{H}$ is the physical model of MMW scattering, and $z$ is the input of the network. Considering the complex-valued characteristics of MMW echoes, we constructed a lightweight complex-valued convolutional network (CCN)[39–41] as shown in Fig. 2a. More details are referred to "Untrained reconstruction based on CCN". We applied the complex-valued total variation (CTV)[42] regularization to enhance the network's output. The

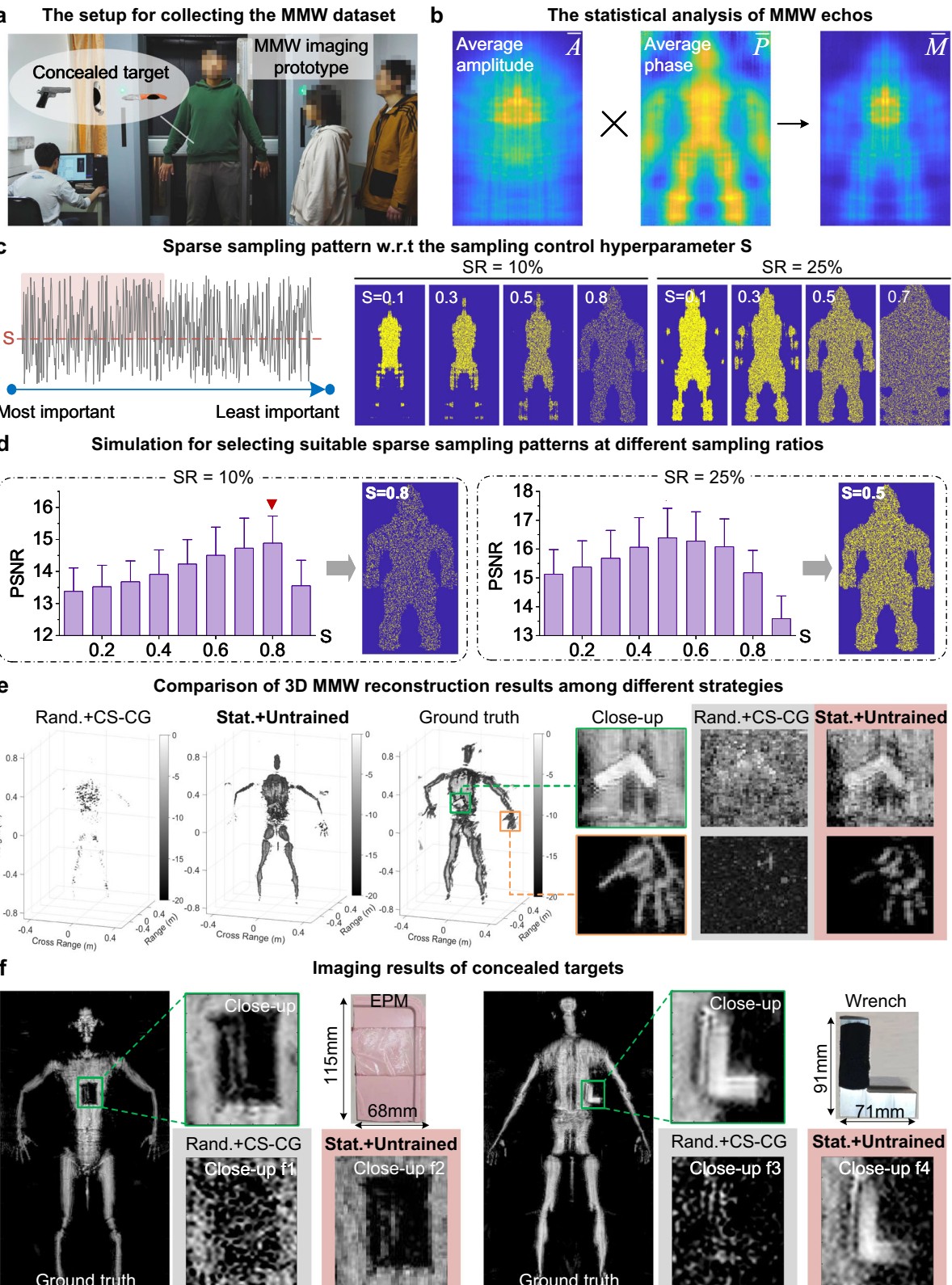

**Fig. 1 | The principle of the reported statistically sparse array design. a** We collected a full-sampled MMW concealed target inspection dataset containing 1934 echoes. **b** The statistical ranking $\bar{M}$ was obtained by multiplying the average amplitude $\bar{A}$ and inverse phase gradient $\bar{P}$. **c** With the preset sampling control hyperparameter $S$ and a uniform random function $r(n)$, we select the $n$-th point if $r(n) > S$ following the statistical ranking in **b**. **d** The reconstruction accuracy (PSNR) of 16 randomly selected echoes w.r.t. different $S$. The error bar represents the standard deviation. The optimal sparse patterns of Sampling Ratio(SR)=10% & 25% correspond to $S = 0.8$ & 0.5, respectively. More details are in Supplementary Note 3. **e** The reconstructed 3D scenes at 25% sampling ratio. **f** The reconstructed results of concealed target detection at 25% sampling ratio. We compared the random sampling with conventional CS-CG[43] reconstruction and the reported statistically sparse sampling with the untrained reconstruction in **e** and **f**.

**Table 1 | The quantitative comparison among different sampling strategies and reconstruction methods**

| Array | | Random | | | | | Statistically optimized | | | | |
|---|---|---|---|---|---|---|---|---|---|---|---|
| SR | Metrics | RMA | CS-CG | ADMM | DL | Untrained | RMA | CS-CG | ADMM | DL | Untrained |
| 10% | RMSE | 63.66/9.63 | 59.10/13.49 | 43.58/5.65 | 46.62/8.23 | 40.05/5.84 | 49.92/5.97 | 50.93/9.75 | 40.82/5.67 | 32.76/8.11 | 31.53/5.42 |
| | PSNR | 12.13/1.32 | 12.91/1.72 | 15.41/1.09 | 14.88/1.45 | 16.17/1.23 | 14.26/1.00 | 14.28/1.52 | 16.11/1.00 | 18.07/2.05 | 18.40/1.16 |
| | SSIM | 0.12/0.05 | 0.05/0.05 | 0.13/0.04 | 0.26/0.13 | 0.53/0.06 | 0.32/0.05 | 0.44/0.14 | 0.24/0.03 | 0.60/0.11 | 0.65/0.05 |
| 25% | RMSE | 53.33/8.49 | 44.90/7.39 | 47.15/7.96 | 35.65/10.09 | 28.23/4.20 | 33.89/4.86 | 35.94/3.28 | 40.91/7.22 | 28.05/8.86 | 23.90/3.31 |
| | PSNR | 13.75/1.36 | 15.22/1.25 | 14.90/1.45 | 17.40/2.28 | 19.30/1.24 | 17.65/1.25 | 17.07/0.77 | 16.09/1.54 | 19.55/2.50 | 20.72/1.15 |
| | SSIM | 0.19/0.06 | 0.16/0.07 | 0.56/0.06 | 0.49/0.16 | 0.59/0.06 | 0.45/0.07 | 0.60/0.07 | 0.61/0.05 | 0.64/0.15 | 0.72/0.05 |

We used all the 200 echoes of the test dataset for reconstruction. The evaluation metrics include the average/standard deviation of RMSE (↓), PSNR (↑), and SSIM (↑). More details are referred to Supplementary Note 7.
The bold font indicates the best RMSE/PSNR/SSIM value.

network does not require pre-training. Instead, it is the interplay between $\mathcal{H}$ and $f_\theta$ that causes the prior of $E_s$ to be captured by the neural network. When the optimization converges, the resulting $f_\theta$ corresponds to the inverse model of $\mathcal{H}$, which can be applied to reconstruct the target scene $I = f_\theta(z)$.

To evaluate the performance of the reported reconstruction method, the commonly used RMA, state-of-the-art CS-based methods including compressed sensing-complex gradient (CS-CG)[43], Alternating direction method of multipliers (ADMM)[44], and DL-based reconstruction networks were employed for comparison. We trained four complex-valued UNet-based[45] networks, corresponding to random array + 10% sampling ratio, statistically optimized array + 10% sampling ratio, random array + 25% sampling ratio, and statistically optimized array + 25% sampling ratio. Details of these DL networks can be found in Supplementary Note 5.1. As shown in Fig. 2, the reported untrained reconstruction can retrieve more clear textures of concealed targets, while RMA causes more artifacts and other CS-based methods cannot mitigate the clutter components and produce target details. Moreover, the comparison in Table 1 shows that the reported untrained learning outperforms existing algorithms under both random and statistics-based sampling. It shows an average 2.61 dB and 4.19 dB improvement in PSNR compared to existing CS-based approaches at 10% and 25% sampling ratios, respectively. Compared to the DL methods, untrained learning employed physics-based constraints, making the reconstruction quality more robust (with a smaller standard deviation). More details are referred to Supplementary Note 7.

Based on our experiment implementation, the average solution times of RMA, CS-CG, and ADMM methods running by Matlab on 2 × Intel E5-2687W CPUs are 2 s, 610 s, and 66 s, respectively. While the untrained method has a relatively high complexity, the testing shows that 100−200 iterations were sufficient for identifying concealed targets in most cases, which took <21 s on an NVIDIA RTX 4090 GPU. Further, the running time tends to be less than 1.6 s on an NVIDIA H100 GPU. Details about the analysis of complexity and running time are referred to Supplementary Note 5.4. The reported framework does not require a scanning process, so it can be adapted to most application scenarios, such as airports and key train stations, where it can safely and conveniently inspect hidden items through clothing and enhance the efficiency of security checks.

### Concealed target detection
To evaluate the concealed target detection performance of the reported technique, we adopted the commonly used object detection network YOLOv8[46]. Similar to multi-class detection works[47–49], the detection network was employed to distinguish five kinds of common hidden targets, including knives, wrenches, phones, guns, and EPM. We utilized the F1 score and mean Average Precision (mAP)[50] to evaluate the detection results and the imaging methods' performance.

More details can be found in Supplementary Note 5.5. The results are presented in Fig. 3. We can see that when using the detection network trained with the full-sampled echo reconstructions, both our statistically sparse array and untrained reconstruction exhibit superior accuracy over others. The combination of statistically sparse sampling and untrained reconstruction at 25% sampling ratio produced above 0.4 absolute numerical improvements on F1 and mAP50 compared to the existing imaging schemes (Fig. 3a). Figure 3b illustrates the detection results with bounding boxes at 10% sampling ratio. Under this extremely low sampling ratio, only the combination of the statistically sparse array and untrained reconstruction can yield an accurate location and classification of the concealed objects. The DL method with statistically sparse sampling can recover the body shape but failed to reconstruct the concealed target. If we train the detection network using corresponding sparsely sampled echo reconstructions, we can achieve higher accuracy compared to training with full-sampled echo reconstructions (Fig. 3c). To sum up, by employing statistically sparse array design and untrained learning for robust reconstruction, we can achieve successful single-shot detection of targets using only 10% of the original full antenna array elements. This indicates that the reported technique is promising to reduce the cost of antenna arrays for single-shot MMW security inspection by an order of magnitude.

### Applicability and generalization
We have conducted extensive experiments to assess our system's applicability and generalization for reconstruction and detection, encompassing different clothing, subject positioning, body shape, target positioning and status, etc. The exemplar reconstruction and detection results are presented in Fig. 4, while more detailed results can be found in Supplementary Note 8. Drawing from the experiment results, we summarize the main conclusions as follows:

1. Clothing: As shown in Fig. 4a and Supplementary Note 8.1, common materials such as cotton, synthetic fibers, and blended fabrics are penetrable by MMW, meaning that wearing clothes made of these materials does not affect the accuracy of reconstruction and detection. Woolen products, to some extent, may allow MMW penetration, but potentially affect the detection of MMW-absorbing targets. On the other hand, MMW is hard to penetrate leather items, which obstructs the reconstruction and detection of targets concealed under such cloth. Thicker garments made from materials penetrable by MMWs (like sweaters, down jackets, etc.), or layering multiple garments (e.g., T-shirt + sweater + down jacket), do not hinder reconstruction and detection processes. Hence, we advise operators to instruct individuals undergoing scans to remove clothing composed of leather, wool, or fur materials.
2. Body shape: The imaging coverage area of our system is set at around 185 cm in height and 90 cm in width. We tested multiple

## a    Untrained learning scheme for sparse MMW reconstruction

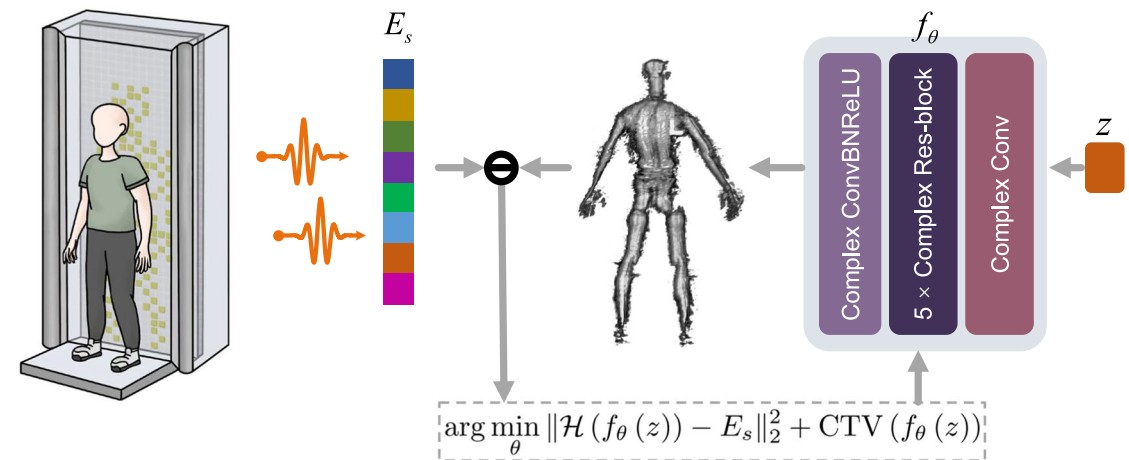

$$\arg\min_{\theta} \|\mathcal{H}(f_\theta(z)) - E_s\|_2^2 + \mathrm{CTV}(f_\theta(z))$$

## b    Reconstruction result comparison among different techniques

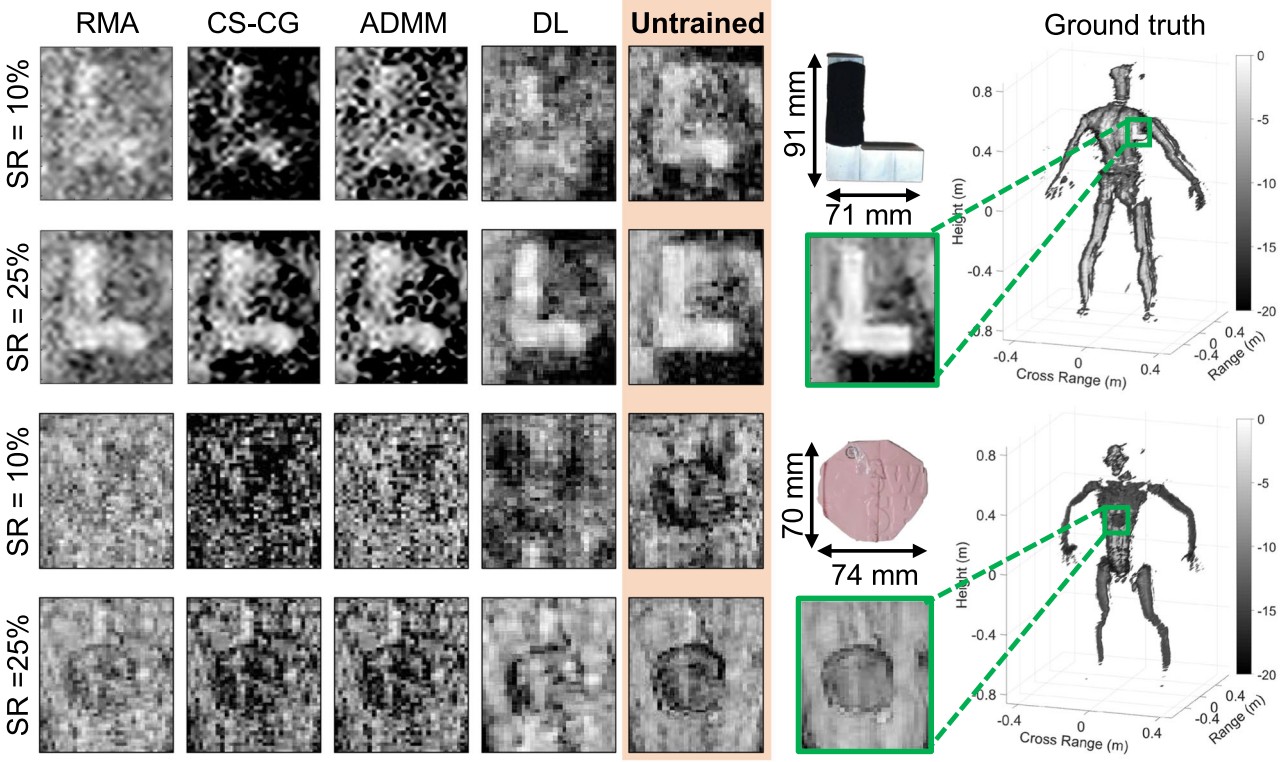

**Fig. 2 | The diagram of the reported lightweight complex-valued convolutional network (CCN). a** We optimize the CCN parameters following the physical constraint (Eq. (4)) in an untrained learning manner to reconstruct the target scene. The CCN network is composed of a complex-valued Conv-BN-ReLU layer, five complex-valued res-blocks, and a complex-valued Conv layer. **b** The reconstruction results of various algorithms with the statistically sparse sampling strategy.

subjects ranging from 160 cm to 189 cm in height and from 45 kg to 95 kg in weight. We experimentally validated that the system can successfully image and detect concealed targets of different people with common body shapes (Supplementary Note 8.2).

3.  Subject position: Orientation variation of the subject may impact detection accuracy. We adopt the orientation facing and parallel to the antenna array as a reference. When the targets are hidden on the subject's chest, the detectable range extends to a left and right rotation of 20° under both 10% and 25% sampling ratios (Fig. 4b, Supplementary Note 8.3). The detectable ranges for forward, backward, and lateral movements are as follows: 10 cm for forward movements, 20 cm for backward movements, and 40 cm (25% sampling ratio) / 15 cm (10% sampling ratio) for lateral movements, as depicted in Fig. 4c and Supplementary Note 8.3. In practice, we can position subjects within the detectable range to ensure successful detection.

4.  Target shape, position, and status: Our prototype was designed for detecting multiple classes of concealed targets, including EPM, knives, phones, guns, and wrenches (Fig. 3c). The length of these targets ranges from 7 to 15 cm, while their width varies from 2 to 11 cm. The employed YOLOv8 network has been validated in handling variations of targets' positions and status[46]. In addition, we have conducted a series of experiments (Supplementary Note 8.4) to validate that within the detectable range, regardless

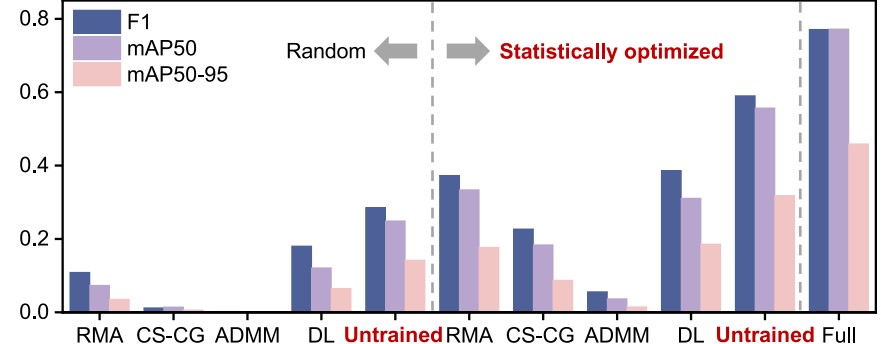

**a** Detection metrics (SR=25%) of the network trained on full-sampled echo reconstructions

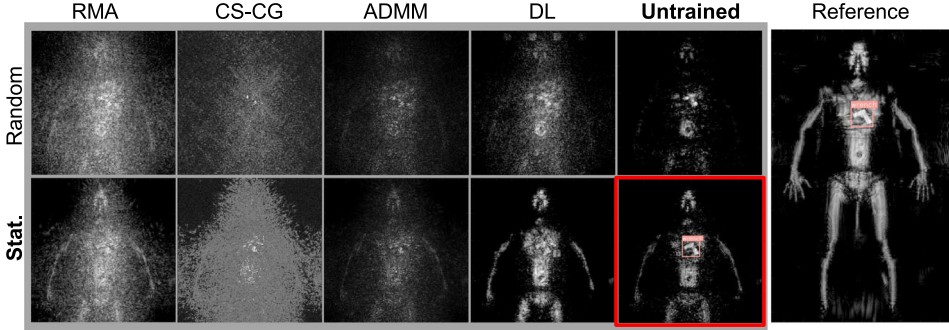

**b** Detection results (SR=10%) of the network trained on full-sampled echo reconstructions

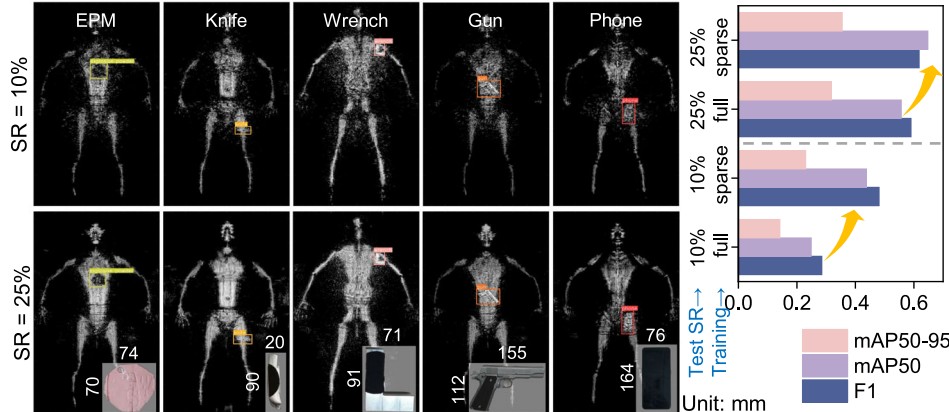

**c** Detection results of the networks trained on sparsely sampled echo reconstructions

**Fig. 3 | Detection results of various sampling strategies and reconstruction algorithms. a** The numerical comparison of detection performance at a 25% sampling ratio. The detection network was trained with full-sampled echo reconstructions. **b** The detection results (SR = 10%) using the network trained with full-sampled echo reconstructions. The combination of statistical sampling strategy and untrained reconstruction can stably recover clear textures from 10% sparsely sampled echoes for detecting concealed targets. **c** The detection results of the network trained with sparsely sampled echo reconstructions. According to the metrics and detection results, these networks exhibit higher accuracy compared to the networks trained with full-sampled echo reconstructions. 'EPM' is the abbreviation for explosive powdered material.

of the position, rotation angle, or open/closed status of the targets, the detection network is capable of successfully detecting hidden targets.

## Discussion

In this work, we developed a large-scale single-shot MMW imaging system for low-cost security inspection. We collected a large-scale MMW human security inspection dataset of real-captured MMW echoes. Based on the dataset, we proposed a statistical sampling approach for sparse MMW array design that maintains high-fidelity reconstruction quality while minimizing the number of elements. Besides, we report an untrained reconstruction approach based on a lightweight complex-valued neural network for sparsely sampled

measurements. By integrating iterative optimization and neural network techniques, this approach overcomes the black-box limitation of end-to-end neural networks, and offers superior reconstruction quality and high reliability for sensitive security inspection. Furthermore, we trained a concealed target detection network, achieving automatic high-precision detection from the reconstructed images. Experiments demonstrated that the reported framework achieved accurate detection of concealed centimetre-sized targets with a 10% sparse array, whereas other contemporary approaches failed at such a low sampling ratio. We conducted comprehensive experiments to validate the performance of the reported technique under variations in clothing, subject positioning, body shape, target positioning, and target status. These experiments validated that the reported system showcases

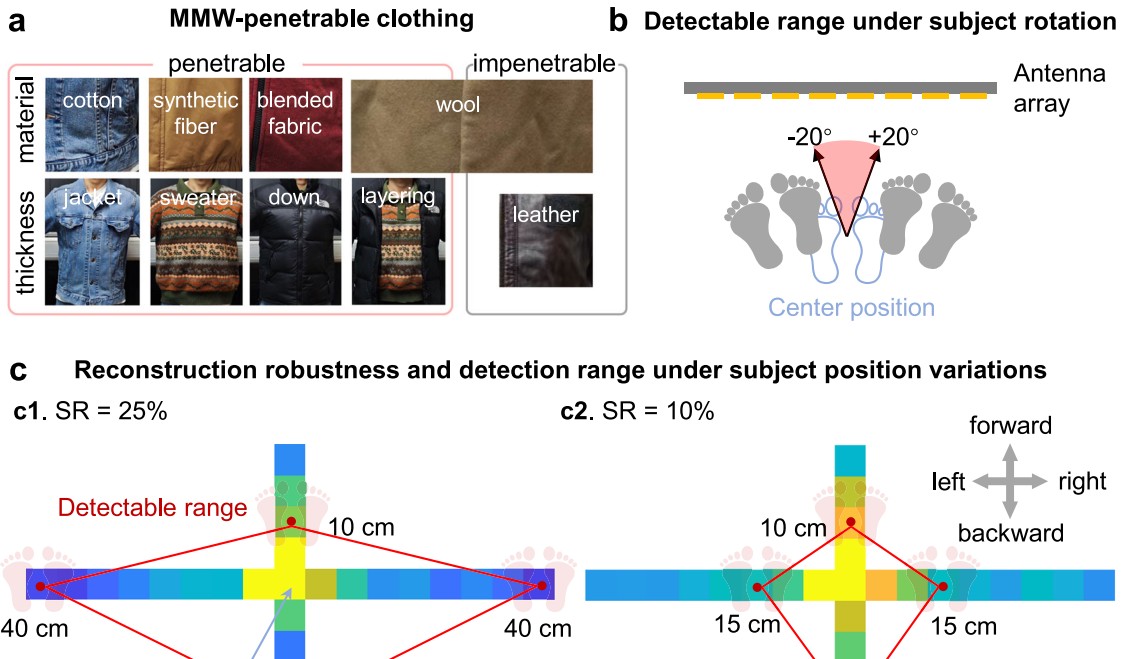

**Fig. 4 | Applicability and generalization of the system. a** MMW-penetrable clothing includes common materials such as cotton, synthetic fiber, and blended fabric. While wool does offer some degree of MMW penetration, it can potentially impact detection accuracy. MMW is hard to penetrate leather items. It does not affect the reconstructed image quality or detection accuracy in cases of common layering of multiple MMW-penetrated clothes. **b** The detectable range under subject rotation. The detectable range is between 20° to the left and right, with the human body facing the array as the reference. **c** The reconstruction robustness (PSNR/dB) and detectable range under varying subject positions. We show the difference between the PSNR values at offset positions and the PSNR value at the original position. The red box indicates the range within which hidden targets can be detected by the detection network. More details are presented in Supplementary Note 8.

effectiveness and robustness in common security check scenarios. Overall, the technique maintains the potential to decrease array cost by more than one order of magnitude for efficient MMW security check.

Aiming for practical applications, the generalization and running efficiency of the reported technique can be further enhanced. First, the dataset diversity can be improved by collecting various echo data corresponding to a variety of body shapes. Second, convolution-accelerate hardware such as high-FLOPs GPUs (see Supplementary Note 5.4), FPGAs[51], and specific SOCs[52,53] can be utilized to improve the speed of reconstruction in practice. We can also investigate such low-precision networks with increased efficiency and comparable performance[54,55]. Third, according to the subject's physique priori, defining a region of interest (ROI) and ignoring the reconstruction outside ROI can effectively reduce the number of network parameters, thereby further improving computational efficiency. Fourth, we can introduce joint learning for both the sampling array and reconstruction network, to further boost the efficiency and accuracy. When the training converges, the network inputs sparsely sampled signals to output the reconstruction results without the need for additional iterations.

The statistical sparse sampling strategy can be applied not only in SISO arrays but also in MIMO ones. For topology design, we can apply the statistical sampling strategy to the equivalent SISO array of a full MIMO array. Then, the statistically sampled equivalent SISO array is mapped to the MIMO array with fewer antenna elements. Besides the aforementioned synthetic aperture radar (SAR) imaging systems, the statistical sparse sampling strategy can also be extended to electrical scanning imaging systems such as electronically scanned antennas (ESAs). In the context of dynamic beamforming techniques, conventional ESAs rely on the phased array technique that enables high-fidelity beam control. However, the use of phased arrays is associated with significant system costs and complex hardware complexity, which limits their applicability in certain contexts. By employing the statistical sparse sampling strategy, it is possible to reduce array density with effective cost savings.

Further, we can introduce the end-to-end sensing strategy[8,56] to further enhance efficiency, which omits the imaging process and extracts high-dimensional semantic information directly from raw measurements. Such an image-free sensing strategy can effectively alleviate storage and bandwidth load originating from the image reconstruction process. Besides, it remains impervious to signal distortion and loss resulting from image reconstruction, thereby facilitating the enhancement of recognition accuracy. What's more, the joint training of both sampling matrix and sensing network can help achieve optimal sensing performance, further enhancing the accuracy of concealed target recognition.

## Method
### System implementation
We built an MMW imaging prototype for the following experiments (Fig. 5a). The prototype emits broadband linear frequency modulated (LFM) signals and receives echoes using the dechirp technique. The system diagram is shown in Fig. 5b. The first and second local oscillators (LO1 and LO2) output carrier signals $f_1$ and $f_2$, respectively, which are mixed with the LFM signal $f_d$ generated by the Direct Digital

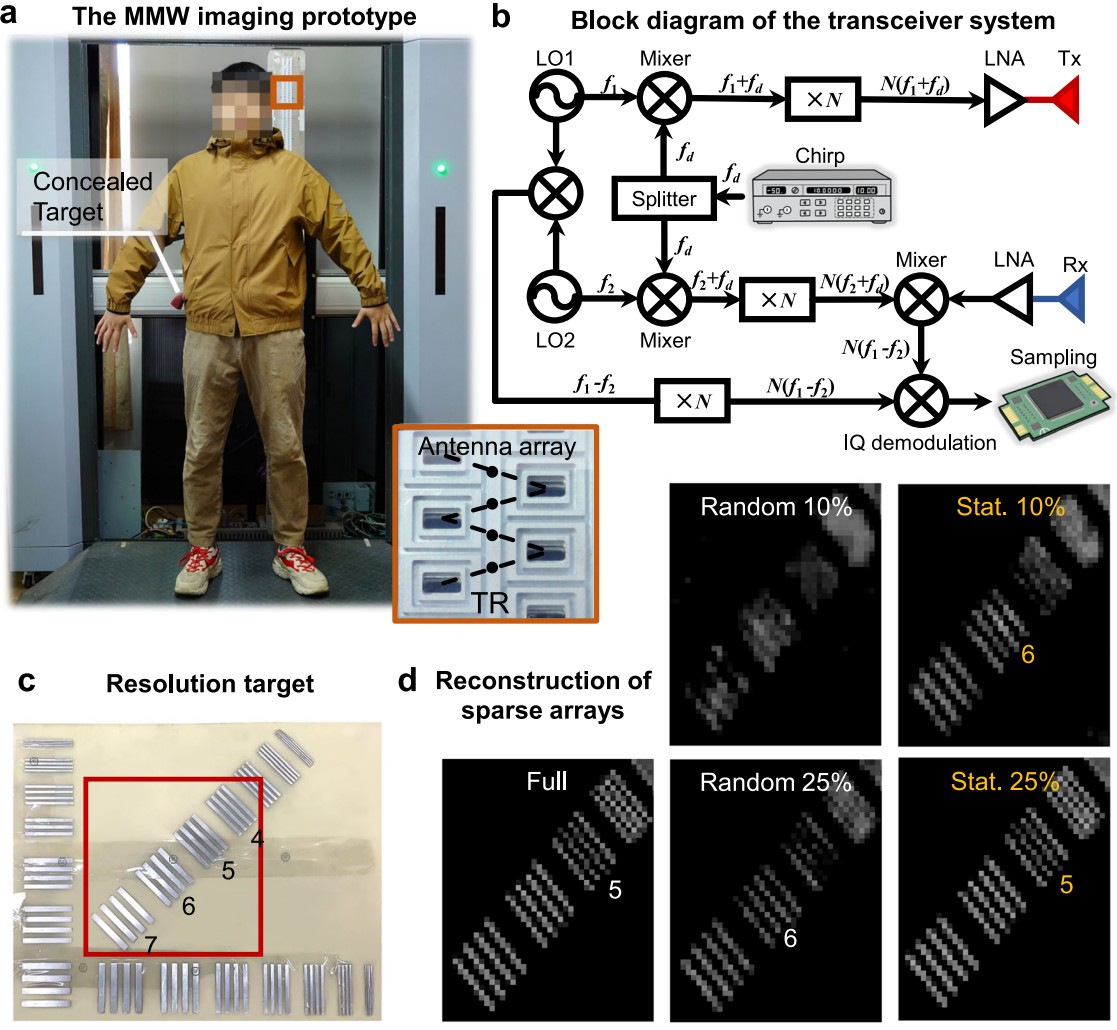

**Fig. 5 | The MMW imaging prototype. a** The working system for human security inspection. **b** The block diagram of the transceiver system. **c** A resolution target and corresponding full-sampled imaging results. The groups 4, 5, 6, and 7 correspond to the resolutions of 4 mm, 5 mm, 6 mm, and 7 mm. **d** The untrained reconstructions of random and statistically optimized sparse arrays. The resolutions of these arrays are 10% random >7 mm, 10% statistically optimized 6 mm, 25% random 6 mm, 25% statistically optimized 5 mm, and full array 5 mm.

Frequency Synthesis, to obtain $f_1 + f_d$ and $f_2 + f_d$, respectively. After $N$-fold frequency multiplication, the transmission signal $N(f_1 + f_d)$ and the reference signal $N(f_2 + f_d)$ are obtained, respectively. The echo $N(f_1 + f_d)$ is de-modulated with the reference signal to obtain the intermediate frequency signal $N(f_1 - f_2)$. The reference intermediate frequency signal $N(f_1 - f_2)$ is obtained by mixing and $N$-fold frequency multiplication of the LO signals. Finally, the analog IQ demodulation is employed to produce the baseband signal, with subsequent digital sampling.

The working frequency of the prototype ranges from 32 to 37 GHz. The prototype consists of a linear monostatic array under a vertical mechanical scanning structure. The antenna is in the waveguide slot form with vertical linear polarization. The transmit and receive arrays are arranged in a staggered manner, thus an equivalent sampling point is formed at the center between two neighboring transmit and receive elements. The sampling interval of the equivalent sampling point meets the system requirements[57]. The transmit antennas work in sequence, with the two neighboring receive antennas collecting the EM wave at the same time. The aperture size is 2 m × 1 m, providing the spatial resolutions both to be 5 mm (Fig. 5d). Statistically optimized arrays exhibit higher resolution than random arrays. With the reported statistically sparse sampling strategy, a 25% sparse array can achieve the same resolution as the original full array. Even using a 10% sparse array, the technique can achieve a resolution of 6 mm.

## Analysis on the statistical maps of MMW echoes

We conducted an analysis on the phase gradient map of a collection of real-captured echoes from a fully-sampled antenna array. This part aims to elucidate the underlying reasons why this map is capable of effectively reflecting the statistical importance ranking of phase in MMW echoes.

We consider a scenario where the target is distributed along the $x$ and $y$ axes. In this context, $\sigma(x,y)$ represents the scattering coefficient, $k_0$ corresponds to the wavenumber, and $x'$ and $y'$ denote the respective antenna locations along these axes. Additionally, $R_0$ denotes the distance between the target and the antenna array. The gradient of echo along the $x$-axis can be mathematically expressed as

$$\frac{\partial s(x',y')}{\partial x'} = -2C \int_{y'-y_{max}}^{y'+y_{max}} \int_{x'-x_{max}}^{x'+x_{max}} \sigma(x,y)e^{-jk_0\frac{(x'-x)^2+(y'-y)^2}{2R_0}}(x-x')\mathrm{d}x\mathrm{d}y,$$

(1)

where $C$ is a constant number, and

$$x_{max} = R_0 \tan\frac{\Theta_x}{2},$$
$$y_{max} = R_0 \tan\frac{\Theta_y}{2}.$$

(2)

Herein, $\Theta_x$ and $\Theta_y$ represent the antenna beamwidth along the azimuth and height directions, respectively. Due to the relatively uniform distribution of scattering coefficients $\sigma(x, y)$ of human targets, the derivatives $\frac{\partial s(x',y')}{\partial x'} \to 0$, $\frac{\partial s(x',y')}{\partial y'} \to 0$. In other words, selecting elements with small gradients can improve the quality of illumination for the target of interest. A more detailed analysis is referred to Supplementary Note 4.

Moreover, the averaged amplitude map reflects element significance, with higher amplitude usually indicating greater importance. By multiplying the averaged amplitude map with the inverse phase gradient, we obtain a statistical ranking of element importance. This ranking enables selectively choosing antenna elements of the highest importance at various sampling ratios.

**Statistically sparse sampling**

We developed a quantitative statistically sparse sampling strategy to obtain the sparse pattern $M$. Different from handcrafted-designed or other sparse sampling strategies[58], we selected the elements in the order of statistical importance in $\bar{M}$ with a fixed probability. Given the statistical prior $\bar{M}$ and a uniform random function $r(n)$ ranging from 0 to 1, we selected the element $n$ of $r(n) > S$, where $S$ is a hyperparameter to control the sparsity of the sampling pattern. The sparse pattern $M$ can be formulated as

$$M(n) = \begin{cases} 1 & r(n) > S, \\ 0 & r(n) \leq S, \end{cases} \tag{3}$$

where '1' means the element to be selected. When 1-$S$ is larger than the sampling ratio, the total number of selected elements is greater than the preset number determined by the sampling ratio, and the last out-of-range elements will be discarded. When 1-$S$ equals to the sampling ratio, the obtained array is a uniformly random array. There won't be enough elements in the resulting sparse array when 1-$S$ is less than the sampling ratio.

**Untrained reconstruction based on CCN**

The objective of the reported untrained reconstruction is formulated as

$$\arg\min_\theta \| \mathcal{H}(f_\theta(z)) - E_s \|_2^2 + CTV(f_\theta(z)), \tag{4}$$

where $\theta$ denotes the parameters of the CCN $f_\theta$, $E_s$ is the sparse MMW measurement, $\mathcal{H}$ is the physical model of MMW scattering, and $z$ is the input of the network. The scattering process $\mathcal{H}$ can be denoted as

$$\mathcal{H}[\cdot] = \mathcal{F}_{2D}^{-1}\{IN_k\{\mathcal{F}_{3D}[\cdot]\}e^{-jk_y R_0}\}, \tag{5}$$

where $\mathcal{F}_{3D}\{\cdot\}$ represents a 3D spatial Fourier transform for all the spatial dimensions of the imaging region. $\mathcal{F}_{2D}^{-1}\{\cdot\}$ denotes the 2D spatial inverse Fourier transform over the 2D array aperture. $IN_k$ indicates the interpolation with respect to the wavenumber $k$.

As shown in Fig. 2a, the reported lightweight CCN has 7 blocks, consisting of an input complex-valued Conv-BN-ReLU block, 5 complex-valued Res-blocks, and an output complex-valued Conv layer in sequential. We utilized the BFloat 16-bit (BF16) quantization to reduce computational workload while maintaining high reconstruction accuracy. All the complex-valued convolutional layers in the network have 256 complex-valued kernels (kernel size = 3, step = 1, padding = 1). What differentiates CCN from a real-valued convolutional network is the complex-valued convolutional layer. The complex-valued convolution takes the real and imaginary parts of a complex-valued feature as two-channel input and convolutes the input features with complex-valued kernels. We took the reconstructed 3D scene by RMA as the input $z$ and used the Adam[59] solver with a learning rate of 0.001 to update the parameter $\theta$. The detailed analysis of network

depth, quantization, gradient descent algorithm, and regularization can be found in Supplementary Note 5.2. The network was implemented on the Pytorch 2.0 platform.

1. Complex-valued convolutional layer: Given the input complex-valued feature map $F = F_R + iF_I$ and the complex-valued convolutional kernel $K = K_R + iK_I$, the complex-valued convolution is denoted as

$$\begin{aligned} F * K &= (F_R + iF_I) * (K_R + iK_I) \\ &= (F_R * K_R - F_I * K_I) + i(F_R * K_I + F_I * K_R), \end{aligned} \tag{6}$$

where $*$ denotes the convolution operation. In the implementation, we treat complex values as two-channel real values (real channel $R$ and imaginary channel $I$).

2. Complex-valued ReLU: The complex-valued ReLU is denoted as

$$\text{Complex ReLU}(x) = \text{ReLU}(x_R) + i\text{ReLU}(x_I), \tag{7}$$

where the ReLU function is

$$\text{ReLU}(x) = \begin{cases} x, & \text{if } x \geq 0, \\ 0, & \text{otherwise}. \end{cases} \tag{8}$$

3. Complex-valued batch normalization: The complex-valued batch normalization whitens the complex-valued features by multiplying the 0-centered data $(F - \mathbb{E}(F))$ by the inverse square root of the 2 × 2 covariance matrix $V$

$$\widetilde{F} = V^{-\frac{1}{2}}(F - \mathbb{E}(F)), \tag{9}$$

where $\mathbb{E}$ represents the mathematical expectation, and the covariance matrix $V$ is denoted as

$$V = \begin{bmatrix} V_{rr} & V_{ri} \\ V_{ir} & V_{ii} \end{bmatrix} = \begin{bmatrix} \text{Cov}(F_R, F_R) & \text{Cov}(F_R, F_I) \\ \text{Cov}(F_I, F_R) & \text{Cov}(F_I, F_I) \end{bmatrix}. \tag{10}$$

Same as the traditional BN, the complex-valued BN also has the learnable scaling parameter $\gamma$ and shift parameter $\beta$

$$\text{Complex BN}(F) = \gamma \widetilde{F} + \beta. \tag{11}$$

4. Complex-valued res-block: The complex-valued res-block contains complex-valued Conv-BN-ReLU-Conv-BN layers arranged sequentially, and the final BN layer's output feature is directly summed with the input feature. In our implementation, we employed a network architecture comprising five complex-valued res-blocks.

**Reporting summary**

Further information on research design is available in the Nature Portfolio Reporting Summary linked to this article.

## Data availability

The minimum data generated in this study have been deposited in the Zenodo database under accession code https://zenodo.org/doi/10.5281/zenodo.11091264[60]. The complete data is available under restricted access following the funded project requirements. Access can be obtained by reasonable request to the corresponding authors.

## Code availability

The demo code of the reported technique is available at https://github.com/bianlab/MMW.

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

## Acknowledgements

We appreciate the assistance from Beijing Zhongdun Anmin Fenxi Technology Limited Company in conducting certain measurements. This work was supported by the National Natural Science Foundation of China under Grants 62322502 (L.B.) and 62131003 (L.B.), and the Guangdong Province Key Laboratory of Intelligent Detection in Complex Environment of Aerospace, Land and Sea under Grant 2022KSYS016 (L.B.).

## Author contributions

L.B., D.L., S.W., and S.L. conceived the idea. D.L., S.W., H.L., J.W., C.T., H.X., G.Z., and X.C. conducted the experiments. D.L. and S.W. performed data analysis. L.B., S.L., and J.Z. supervised the project. All the authors participated in the analysis and discussion of the results.

## Competing interests

L.B. and D.L. hold patents on technologies related to the devices developed in this work (China patent numbers ZL202210778396.7 and ZL202010522279.5) and submitted related patent applications. The remaining authors declare no competing interests.
