## [Peer Review File · Nature Communications]

Towards Large-scale Single-shot Millimeter-wave Imaging for Low-cost Security InspectionREVIEWER COMMENTS

Reviewer #1 (Remarks to the Author):

Pros:

The proposed method leveraging the precaptured radar echo signals to design radar arrays is interesting and novel. I like this idea.

The proposed method has achieved significant improvement over existing methods regarding the reconstruction quality and recognition accuracy in applying the signals for object detection.

Major issues:

Is the learned sampling prior easily overfit to the echos collected from a certain distance, certain cloth, certain body shapes, certain objects, etc.? Does it generalize well to these cases (thickness of clothing, object size, body shapes, distances, orientation of the subjects, etc)? If not, what is the bound, and how the bound is related to the hardware and the algorithm? No study on these questions. Though the authors mention some of these questions the future work, I think without these studies, the current submission is too preliminary.

The results only compared with randomly sampled arrays (e.g. Table 1 Figure 2 and Figure 3). What are the results of a regular grid with different sparsity? Will this learned sampling strategy perform better with the regular grids?

How the ground truth is obtained?

In the reported metrics, how many times of repeated sampling experiments are carried out? What's the variance of different experiments?

Is the image for M in Fig. 1 the result of all the echos or an echo for a subject? Any insight for this M? Why does it look like this? How does it relate to the physical traits of radar signals?

No array or used signals from imaging resulting from the sampling is shown.

To summarize, the proposed method leveraging machine learning to help the radar array design is novel and inspiring. It has potential contributions to the community of MMwave imaging. However, the solidness of the work still requires further improvement regarding the study of the key design of the algorithms and the generalizations and insight of the learned results. Therefore, a major revision is suggested.

Reviewer #2 (Remarks to the Author):

This paper describes the development of an imaging system operating in the millimeter range. In the context of the development of a functional prototype, the authors highlight their contributions in three areas: The exploitation of a sparse antenna array, the development of a reconstruction technique adapted to the latter, and the adaptation of an automated target classification technique.

The images obtained using this system are highly convincing, demonstrating the team's ability to master all the engineering elements required for the successful operation of these complex systems.

However, there are a number of points which prevent me from recommending publication of this work as it stands in a journal such as Nature Communications.

Originality of the work and positioning in relation to the scientific literature:

According to my understanding of the paper, the authors first propose an approach inspired by compressive sensing to alleviate the hardware complexity inherent in the proper operation of these body scanners. The positioning in relation to the scientific literature seems to me incomplete on this first point, not allowing me to identify any real added value. From my understanding of the work presented, the aim is to restrict the application of this imaging system to a restricted dataset, considering only body scanner applications. The assumed correlation of the scenes to be imaged in the region of interest then enables the amount of information to be reduced by identifying the main structures associated. It then seems necessary to define a propagation operator involving a certain loss of information, implying that it is then possible to relate this to the spatial redundancy of the signals measured by the imaging system within the restricted framework of these body scanner applications.

The reconstruction algorithm seems to me the most relevant part of this work, considering the added value of a non-linear function capable of operating despite the undersampling of the antenna array exploited. There is, however, a substantial new literature in the field of microwave and millimeter wave imaging, exploiting recent advances in machine learning. The latter appears to be inadequately described, despite the fact that some of its processing is identical to that used in the field.

Finally, the authors highlight the exploitation of the automated detection and classification of various illicit or dangerous objects through the adaptation of a machine learning technique. Contrary to the authors' claims, the classification of a variety of objects in a security screening context has already been the subject of publications in recent years, including architectures that have been identified and simplified using compressive sensing and computational imaging techniques.

Reproducibility

This last point seems to me to be particularly critical for the publication of this work. The data presented in this paper, and in the supplementary materials, in no way ensure their reproducibility.

Conclusion

In conclusion, this work represents a fine engineering demonstration of the research team's ability to develop a functional system operating in a realistic situation. However, the added value put forward by the authors seems to me to be only incremental, and is insufficiently described and positioned in relation to the scientific literature, allowing neither a real means of comparison with the state of the art, nor ensuring its reproducibility.

Some additional points in reading order:

The proposed numerical approach, based on data multiplication derived from amplitude and phase information could benefit from a more complete illustration in supplementary materials. It could be interesting to show the raw data extracted at center frequency in amplitude and phase to justify the relevance of this method. An interesting link with existing support detection techniques in compressive sensing could also be useful.

The array sub-sampling strategy could be further developed. The link with physical considerations and rank analyses does not seem so obvious as it stands. The proposed description gives the impression that the authors have simply made a random selection of elements by reducing the number of antennas fed until they reach a threshold considered no longer acceptable under the experimental conditions studied.

The reconstruction technique studied is positioned in relation to the Range Migration Algorithm, applied to spatially truncated data. It seems necessary for the authors to propose a more detailed description of the steps considered for the adaptation of the RMA. Indeed, I have the impression that the results obtained will be particularly sensitive to the interpolation techniques used, as well as to the methods for merging samples in reciprocal space, given the multi-static nature of the imaging system under consideration.

Reviewer #3 (Remarks to the Author):

A sparse antenna array based large-scale single-shot millimeter-wave imaging framework is proposed in the manuscript to obtain low-cost high-fidelity security inspection. The untrained interpretable learning scheme is used to improve the robustness and the imaging accuracy of the imaging system. Based on the 10% sparse array, the concealed centimeter-sized targets can be effectively reconstructed. The experiment data are used to validate the proposed method under the given scene. The proposed method seems good. However, the contribution of the manuscript should be highlighted accurately. In addition, some issues should be addressed.

1, The cost of the proposed method is low, which is very important for the security check. Please give more description about the cost of the proposed method compared with the scanning method for example. How many elements are used for the 10% sparse array? The good reconstruction with low sampling ratio data is usually hard, which is depending on the array size. Please analyze the relationship between the array size and the performance of the proposed method.

2, The time cost should also be considered for the security check application. 30s is a very long time if the proposed method cannot be improved efficiently. There are several steps during the security check, such as detection, data processing, imaging and target recognition, etc. A time cost table is suggested to show how the time is used. The potential analysis of the improvement in the time cost for the proposed method is better given.

3, In Section 2.2, the verification platforms are different for the traditional method and the proposed method. Please analyze the computing power directly. It seems the proposed method need more computing power if the parallel computing is not used.

4, In Section 2.3, there are five kinds of common targets in the network training. Are the rotation direction, the opening and closing status, and the spatial attitude considered? How many validation samples are used? How about the generalization ability of the proposed method?

5, Is there any improvement of the untrained reconstruction technique in the proposed method? What is the contribution in addition to the usage of the untrained reconstruction technique?

6, How about the transmit and receive isolation of the detection system shown in Fig.4.

7, The resolution of the proposed imaging system is good. Please analyze how much position error can be tolerant during the data detection.

Dear Editor and Reviewers:

Thank you very much for your valuable comments on our manuscript NCOMMS-23-29557. We have carefully revised the manuscript according to the comments. For convenience, we highlight the revisions with blue font in the revised manuscript. The point-to-point responses to the comments are listed below.

Sincerely,

All the authors

Index

Reviewer 1	Reviewer 2	Reviewer 3
• Comment 1: page 1	• Comment 1: page 41	• Comment 1: page 102
• Comment 2: page 1	• Comment 2: page 41	• Comment 2: page 111
• Comment 3: page 19	• Comment 3: page 56	• Comment 3: page 113
• Comment 4: page 22	• Comment 4: page 61	• Comment 4: page 116
• Comment 5: page 22	• Comment 5: page 73	• Comment 5: page 131
• Comment 6: page 29	• Comment 6: page 76	• Comment 6: page 134
• Comment 7: page 36	• Comment 7: page 78	• Comment 7: page 135
• Comment 8: page 39	• Comment 8: page 84	
	• Comment 9: page 88	

[Reviewer 1]

<Comment 1>

The proposed method leveraging the precaptured radar echo signals to design radar arrays is interesting and novel. I like this idea. The proposed method has achieved significant improvement over existing methods regarding the reconstruction quality and recognition accuracy in applying the signals for object detection.

<Response>

Thanks for the reviewer's recognition of this manuscript and the positive comments.

<Comment 2>

Is the learned sampling prior easily overfit to the echos collected from a certain distance, certain cloth, certain body shapes, certain objects, etc.? Does it generalize well to these cases (thickness of clothing, object size, body shapes, distances, orientation of the subjects, etc.)? If not, what is the bound, and how the bound is related to the hardware and the algorithm? No study on these questions. Though the authors mention some of these questions the future work, I think without these studies, the current submission is too preliminary.

<Response>

Thank you for the reviewer's suggestions. We conducted extensive additional experiments to reveal the applicability and generalization of the reported system. These experiments explored reconstruction and detection accuracy under various conditions, including clothing material and thickness, subject position, body shape, target shape, position and status, etc. Based on our findings, we conclude that:

- **Clothing:** We tested various materials (cotton, synthetic fiber, blended fabric, feather, and wool) and thicknesses (jeans, jacket, sweater, down jacket, and down jacket + sweater + T-shirt) of clothing, as shown in Fig. S11 and Fig. S12, respectively. Our system performs well in all cases except when leather and wool clothes are involved. MMW is hard to penetrate leather items, which obstructs the reconstruction and detection of concealed targets. Wool is somewhat MMW-penetrable. However, it interferes with the detection of MMW-absorbing materials such as explosive powdered material (EPM). Therefore, we recommend operators

request subjects remove wool, leather, and fur clothing during MMW security screening.

- **Body shape:** According to the system settings and the reconstruction results, the imaging coverage area is around 185cm in height and 100cm in width. We tested several cases, including heights ranging from 160cm to 189cm and weights ranging from 45kg to 95kg. The reconstruction and detection results are shown in Fig. S13. The proposed system can successfully image and detect the concealed targets of different people with common body shapes.
- **Subject position:** We tested the subjects standing offset (forward, backward, left, right) and rotating from the central position. To establish a baseline for comparison, we utilized the imaging result obtained with the subject in the central position as the benchmark. The numerical results in Fig. S14 indicate that the reconstruction quality is degrading as these deviations become larger. As shown in Fig. S15, Fig. S16, and Fig. S17, the detectable ranges of the proposed statistically optimized arrays corresponding to these deviations are:
 - 25% sampling ratio: forward 10cm, backward 20cm, left and right 40cm, rotation 20°;
 - 10% sampling ratio: forward 10cm, backward 20cm, left and right 15cm, rotation 20°.

In practice, we can position subjects within the detectable range to ensure successful detection.

- **Target shape, position, and status:** Five different kinds of objects are involved as targets, including EPM, knife, phone, gun, and wrench (Fig. 5 c). The length of these targets ranges from 7 to 15 cm, while their width varies from 2 to 11 cm. We tested various target objects in different positions and statuses. The object can be successfully imaged and detected when hidden within the detectable range of the human body (Fig. S16). The detection network is robust to changes in object rotation and the open-closed status of knives (Fig. S18).
- **Spatial resolution:** We tested the spatial resolution of both random and statistically optimized sparse arrays. The reconstruction results are shown in Fig. 5, indicating that the statistically optimized sparse array exhibits a higher spatial resolution than the random array under the same sampling ratio. The spatial resolution of the full array is 5mm. The statistically optimized arrays, coupled with the untrained learning reconstruction, attain 6mm

resolution at a 10% sampling ratio and 5mm at a 25% sampling ratio. The system will face challenges in imaging and detecting targets with dimensions smaller than the spatial resolution.

We have added the detailed descriptions and relevant revisions as follows:

Location: Supplement Note 10

“Additional details of applicability and generalization

We conducted a series of experiments to reveal the applicability and generalization of the system. The experiments involved spatial resolution, clothing, body shape, subject position, target position and status, and element assembly error. The reference images are reconstructed from full-sampled echoes by RMA, while the sparsely sampled images are reconstructed by the untrained method.

Clothing material and thickness

As shown in Fig. S11, we can detect hidden objects under cotton, synthetic fiber, and blended fabric clothes. While other materials exhibit good reconstruction quality and facilitate concealed object identification, leather impedes MMW, making it hard to identify hidden targets. Wool, on the other hand, is somewhat MMW-penetrable. However, it interferes with the detection of MMW-absorbing materials such as EPM. So we recommend operators request the subject take off wool, leather, and fur clothing when performing MMW security checking. Further, we tested the reconstruction and detection results in cases of various clothing thicknesses, as shown in Fig. S12. For MMW-penetrated clothing materials, it does not affect the reconstructed image quality and detection accuracy whether the subject is wearing a single jacket or the common layering of multiple thick garments (Down jacket + sweater + T-shirt).

Fig. S11: Reconstructed images and detection results with different types of clothing material. The leather cloth impedes MMW, resulting in the inability or misleading to identify hidden targets. The wool cloth may affect the detection of MMW-absorbing targets such as EPM.

Fig. S12: Reconstructed images and detection results with different clothing thicknesses. The proposed scheme is not sensitive to the thicknesses of MMW-penetrable clothes.

Body shape

We tested several cases, including heights ranging from 160cm to 189cm and weights ranging from 45kg to 95kg. The reconstruction and detection results are shown in Fig. S13. The imaging coverage area of our system is set at around 185cm in height and 100cm in width, which can be validated by the reconstruction results. Even in the case where subjects have a height exceeding 185cm, the target hidden in the clothes can also be detected. The proposed system can successfully image and detect the concealed targets of different people with common body shapes. We aim to gather an even more expansive dataset that encompasses a greater variety of body shapes to further promote the applicability of the proposed system across a wider range of real-world use cases.

Fig. S13: Reconstructed images and detection results in cases of various body shapes with heights ranging from 160cm to 189cm and weights ranging from 45kg to 95kg. The proposed scheme can deal with subjects with various body shapes.

Subject position

We conducted a series of experiments to reveal the performance boundary in cases where the

subject is positioned off-center or rotates away from the central position. The space for the human movement was expanded to 115cm (azimuth) \times 80cm (range). Due to the inherent width and thickness of the human body, the feasible range for left and right movement of the subject is ± 40 cm, and the range for forward and backward movement is ± 20 cm. Besides, we tested situations with the subject facing forward (0°) and at angular deviations of 10° , 20° , 30° , and 45° . The numerical evaluations of offset forward/backward/to the left/to the right are shown in Fig. S14. The reconstruction and detection results are shown in Fig. S15 (offset to the right), Fig. S16 (offset forward & backward), and Fig. S17 (subject rotation). We can draw the main conclusions as follows:

- The image quality reconstructed by the random and statistically optimized arrays will decrease when the subject deviates from the central position (Fig. S14). In general, the larger the deviation, the lower the quality.
- However, a few examples may not follow the above rules. This is mainly because the bigger the deviation, the interference of the random array would also deviate, leading to greater sidelobe separation from the primary body. Consequently, some portions of the subject may appear dim, and PSNR may become better.
- As shown in Fig. S15 and Fig. S16, when the subject deviates from the central position, the detectable range is:
 - 25% sampling ratio: forward 10cm, backward 20cm, left and right 40cm;
 - 10% sampling ratio: forward 10cm, backward 20cm, left and right 15cm.
- When the subject's body rotates, the detectable range is within 20° for both 25% and 10% sampling ratios (Fig. S17). In actual use, when it is greater than 20° , the body will show an obvious rotation, and the operator should promptly remind the person being tested to adjust their posture.

Fig. S14: Reconstruction results of the random and statistically optimized sparse arrays in cases where the subject being tested is offset from the central position for inspection.

Fig. S15: Reconstructed images and detection results in cases where the subject is offset to the right from the central position.

Fig. S16: Reconstructed images and detection results in cases where the subject is offset forward or backward from the central position.

Fig. S17: Reconstructed images and detection results in cases where the subject faces different directions. We tested situations with the subject facing forward (0°) and at angular deviations of 10°, 20°, 30°, and 45°.

Target position and status

The utilized YOLO network has demonstrated its effectiveness in handling variations in targets' positions and statuses across natural images [46]. When we migrate it to the MMW image scenario, the network also exhibits robustness to these variations. As shown in Fig. S18, the object can be successfully detected when hidden within the detectable range of the human body. The detection network is robust to changes in object rotation and open-closed status of knives, as shown in Fig. S19.”

Fig. S18: Reconstructed images and detection results in cases of different object positions. The detection network is robust to changes in object positions.

Fig. S19: Reconstructed images and detection results in cases of different object orientations and open-closed statuses. The detection network is robust to changes in the rotation and open-closed status of objects.

Location: Main text, Section 4.1, Line 441 - 445.

“The resolution of the full sampling array is 5mm as illustrated in Fig. 5d. Statistically optimized arrays exhibit higher resolution than random arrays. With the reported statistically sparse design, a 25% sparse array can achieve the same resolution as the full array, and even a 10% sparse array can achieve a 6mm resolution.”

Fig. 5: The MMW imaging prototype. **a**, The working system for human security inspection. **b**, The block diagram of the transceiver system. **c**, The resolution target and full-sampled imaging results of a resolution target. Groups 4, 5, 6, and 7 correspond to 4mm, 5mm, 6mm, and 7mm resolutions. **d**, The untrained reconstructions of random and statistically optimized sparse arrays. The resolutions of these arrays are 10% random > 7mm, 10% statistically optimized 6mm, 25% random 6mm, 25% statistically optimized 5mm, and full array 5mm.

Location: Main text, Section 2.4, Line 274 - 317

“We have conducted extensive experiments to assess our system’s applicability and generalization in reconstruction and detection, encompassing clothing, subject positioning, body shape, target positioning and status, etc. The detailed reconstruction and detection results can be found in Supplement Note 10. We summarize the main conclusions as follows.

Clothing: As shown in Fig. 4 a and Supplement Note 10.1, common materials such as cotton, synthetic fibers, and blended fabrics are penetrable by MMW, meaning that wearing clothes made of these materials does not affect the accuracy of reconstruction and detection. Woolen products, to some extent, may allow MMW penetration, but potentially affect the detection of MMW-absorbing targets. MMW is hard to penetrate leather items, which obstructs the reconstruction and detection of concealed targets. Thicker garments made from materials penetrable by MMWs (like sweaters, down jackets, etc.), or layering multiple garments (e.g., T-shirt + sweater + down jacket), do not hinder reconstruction and detection processes. Hence, we advise operators to instruct individuals undergoing scans to remove clothing composed of leather, wool, or fur materials.

Body shape: The imaging coverage area of our system is set at around 185cm in height and 100cm in width. We tested multiple subjects ranging from 160cm to 189cm in height and from 45kg to 95kg in weight. We experimentally validated that the proposed system can successfully image and detect the concealed targets of different people with common body shapes (Supplement Note 10.2).

Subject position: Variations in the orientation of the subject have an impact on the detection accuracy. We adopt the orientation of the subject facing and parallel to the antenna array as the reference. When the targets are hidden on the subject's chest, the detectable range extends to a left and right rotation of 20° under both 10% and 25% sampling ratios (Fig.4 b, Supplement Note 10.3). The detectable ranges for forward, backward, and lateral movements are as follows: 10cm for forward movements, 20cm for backward movements, and 40cm (25% sampling ratio) / 15cm (10% sampling ratio) for lateral movements, as depicted in Fig. 4 c and Supplement Note 10.3. In practice, we can position subjects within the detectable range to ensure successful detection.

Target shape, position and status: Our prototype is designed for detecting multiple classes of concealed targets, including EPM, knives, phones, guns, and wrenches (Fig. 3 c). The length of these targets ranges from 7 to 15 cm, while their width varies from 2 to 11 cm. The employed YOLOv8 network has proven its effectiveness in handling variations in targets' positions and statuses across natural images [46]. In addition, we have conducted a series of experiments (Supplement Note 10.4) to verify that within the detectable range, regardless of the position, rotation angle, or open/closed status of the targets, the detection network is capable of successfully detecting hidden targets.”

Fig. 4: Applicability and generalization of the system. **a**, MMW-penetrable clothing includes commonly used materials such as cotton, synthetic fiber, and blended fabric. While wool does offer some degree of MMW penetration, it can potentially impact detection accuracy. MMW is hard to penetrate leather items. It does not affect the reconstructed image quality or detection accuracy in cases of common layering of multiple MMW-penetrated clothes. **b**, The detectable range under subject rotation. The detectable range is between 20° to the left and right, with the human body facing the array as the reference. **c**, The reconstruction robustness (PSNR/dB) and detectable range under varying subject positions. We show the difference between the PSNR values at offset positions and the PSNR value at the original position. The red box indicates the range within which hidden targets can be detected by the detection network. More details are in Supplement Note 10.

Fig. 3: Detection results of various sampling strategies and reconstruction algorithms. **a**, The numerical comparison of detection performance at a 25% sampling ratio. The detection network was trained with full-sampled reconstructions. **b**, The detection results of the network trained with full-sampled reconstructions for 10% sampling ratio reconstruction. The combination of statistical sampling and untrained reconstruction can stably recover image textures from 10% sparsely sampled echoes and detect the concealed targets. **c**, The detection results of the network trained with sparsely sampled reconstructions. According to the metrics and detection results, these networks exhibit higher accuracy compared to the networks trained with full-sampled reconstruction. ‘EPM’ is the abbreviation for explosive powdered material.

References:

- [46] Jocher, G., Chaurasia, A., Qiu, J.: YOLO by Ultralytics. <https://github.com/ultralytics>
[S81] Feko, A.: Altair engineering. Inc., www.altairhyperworks.com/feko (2018)

<Comment 3>

The results only compared with randomly sampled arrays (e.g. Table 1 Figure 2 and Figure 3). What are the results of a regular grid with different sparsity? Will this learned sampling strategy perform better with the regular grids?

<Response>

Thanks for the reviewer’s suggestion. Employing regular grids with consistent sparsity introduces the challenge of distorted imaging results due to grating lobes. We have added a detailed analysis of the regular sparse array. In addition, we experimentally compared the reconstruction accuracy of random, regular, and statistically optimized sparse arrays to verify the advancement of the reported statistically sparse array. The relevant revisions in the supplement are provided as follows:

Location: Supplement Note 6

“Comparison with commonly used sparse arrays

This section conducts a comparative analysis between the proposed statistically optimized sparse array and other prevalent sparse array configurations, including both random and regular topologies. The random sparse array is generated through direct sampling from a full array scheme [S39], [25], albeit this can introduce distortions in the images.

Regular grids, characterized by uniform element spacing, hold inherent characteristics that warrant exploration. One crucial factor to consider is that uniform element spacing, when not aligned with the Nyquist sampling criterion, often leads to the emergence of grating lobe artifacts within the reconstructed images. These artifacts manifest as ghost targets and can potentially compromise the overall image quality. This phenomenon is particularly evident when the element interval restricts the imaging region, contributing to the presence of grating lobes. The azimuth and height arrays encompass transmitting and receiving apertures, respectively, and their unambiguous distance in the azimuth direction is influenced by the array’s design parameters.

As articulated in ref. [19], when $\frac{\lambda_c}{\Delta_x} \leq \Theta$, the azimuth unambiguous distance D_x is given by

the formula:

$$D_x = \frac{\lambda_c \cdot R_0}{\Delta_x}, \quad (\text{S26})$$

where D_x represents the azimuth unambiguous distance, λ_c signifies the wavelength of the center operating frequency, R_0 pertains to the imaging distance, Δ_x denotes the element interval, and Θ corresponds to the antenna's beamwidth. Conversely, when $\frac{\lambda_c}{\Delta_x} > \Theta$, the presence of ghost targets becomes less conspicuous; however, it comes at the cost of reduced azimuth resolution. The manifestation of ghost targets is intricately tied to the interplay between λ_c and Δ_x relative to the antenna beamwidth Θ .

To illustrate this phenomenon, we can consider our experimental parameters. We find that for the parameters used in our experiments, the azimuth unambiguous distance, D_x , exceeds 0.75 meters. This is sufficient for effective security checks in near-field imaging scenarios. However, when half of the azimuth elements are removed (sampling rate of 25%), reducing the element spacing, D_x drops to 0.375 meters. This reduction signifies a severe aliasing phenomenon, wherein ghost effects become prominent. Therefore, the ambiguity phenomenon is intricately linked to the specific parameter settings.

Furthermore, we have conducted a comprehensive analysis that includes the assessment of sparse arrays arranged in a regular grid. Our evaluation encompasses sparsity levels of 25% and 10% relative to the full array configuration, as shown in Fig. S6. The results of this examination provide valuable insights into the performance of regular grids with uniform element spacing. In the regular arrays of both sparse rates, it is important to note that the presence of grating lobes introduces interference with the primary body regions, leading to image distortion. Such distortion has the potential to adversely impact concealed target detection procedures. To mitigate this, we have also considered random grid arrays for comparison. Random arrays demonstrate the ability to suppress the ghost effect to some extent, albeit at the potential cost of higher sidelobe effects [S39]. The reported statistically optimized array achieves superior reconstruction performance compared to random and regular arrays.”

Fig. S6: Reconstruction results by CS-CG and the proposed untrained technique under 25% (a) and 10% (b) sampling ratios of different sparse arrays including random, regular, and statistically optimized arrays. The reconstructions of the regular sparse array contain severe artifacts. The reported statistically optimized array achieves superior reconstruction performance compared to random and regular arrays.

References:

- [S39] Wang, S., Li, S., Hoorfar, A., Miao, K., Zhao, G., Sun, H.: Compressive sensing based sparse MIMO array synthesis for wideband near-field millimeter-wave imaging. *IEEE Trans. Aerosp. Electron. Syst.* (2023)
- [25] Wang, S., Li, S., Ren, B., Miao, K., Zhao, G., Sun, H.: Convex optimization-based design of sparse arrays for 3-D near-field imaging. *IEEE Sensors J.* (2023)
- [19] Li, S., Wang, S., Amin, M.G., Zhao, G.: Efficient near-field imaging using cylindrical mimo arrays. *IEEE Trans. Aerosp. Electron. Syst.* 57(6), 3648–3660 (2021)

<Comment 4>

How the ground truth is obtained?

<Response>

Thank you for your inquiry. In our study, the establishment of ground truth involves two essential components:

- Ground truth for imaging results: The ground truth of the imaging results is derived from the application of the Range Migration Algorithm (RMA) to the data obtained from the full-sampled array. These images are characterized by their high-quality representation and serve as the benchmark for our evaluations.
- Ground truth for concealed object detection: The second facet of ground truth involves the determination of the positions and classifications of concealed objects within the MMW images. This ground truth is acquired by directly annotating and marking the positions and classifications of concealed objects on the MMW images using ‘labelImg’ software.

By combining these two distinct sources of ground truth, we ensure the accuracy and reliability of our evaluation metrics, facilitating a comprehensive assessment of the proposed methodology’s performance in detecting concealed objects.

<Comment 5>

In the reported metrics, how many times of repeated sampling experiments are carried out? What’s the variance of different experiments?

<Response>

Thanks for your thoughtful comment. We measured the reconstruction accuracy (RMSE, PSNR, and SSIM) and detection metrics (F1, mAP50, and mAP50-95) on the test dataset which contains 200 randomly selected MMW echoes. The detection metrics (F1, mAP50, and mAP50-95) are statistical measures on the dataset, and typically, studies do not involve the variance of these measures. However, we can provide statistical information such as the mean and standard deviation of reconstruction accuracy (RMSE, PSNR, SSIM), as shown in Tab. 1 and Fig. S9. These numerical results indicate that our untrained learning reconstruction has superior and robust performance with the best mean values of the accuracy metrics and relatively small standard deviations. We have included statements about the statistical information and a detailed description of these experiments in the revised main text and supplement.

Location: Main text, Section 2.2, Line 210 – 228.

“To evaluate the performance of the reported reconstruction, the common-used RMA, state-of-the-art CS-based methods including Compressed Sensing-Complex Gradient (CS-CG) [36], Alternating Direction Method of Multipliers (ADMM) [44], and DL-based reconstruction networks are employed in the following comparisons. We trained four complex-valued UNet-like [45] networks, corresponding to random array + 10% sampling ratio, statistically optimized array + 10% sampling ratio, random array + 25% sampling ratio, and statistically optimized array + 25% sampling ratio. Details about these DL networks are in Supplement Note 7. As shown in Fig. 2, the reported untrained reconstruction can retrieve more clear textures of concealed targets while RMA causes more artifacts, and other CS-based methods cannot mitigate the clutter components and maintain the details of the target. Moreover, the comparison in Tab. 1 leads to the conclusion that the reported untrained learning outperforms existing algorithms in accuracy under both random and statistics-based sampling. It shows an average 2.61dB and 4.19dB improvement in PSNR compared to existing CS-based approaches at 10% and 25% sampling ratios, respectively. Compared to the DL method, untrained learning has constraints of physical priors, which makes the reconstruction quality more robust (with a smaller standard deviation). More details are in Supplement Note 8.”

Table 1 The quantitative comparisons of reconstructed data with different sampling strategies and reconstruction methods. We used all the 200 echoes of the test dataset for reconstruction. The evaluation metrics include the average/standard deviation of RMSE (\downarrow), PSNR (\uparrow), and SSIM (\uparrow). More details are in Supplement Note 8.

Array		Random					Statistically optimized				
SR	Metrics	RMA	CS-CG	ADMM	DL	Untrained	RMA	CS-CG	ADMM	DL	Untrained
10%	RMSE	63.66/9.63	59.10/13.49	43.58/5.65	46.62/8.23	40.05/5.84	49.92/5.97	50.93/9.75	40.82/5.67	32.76/8.11	31.53/5.42
	PSNR	12.13/1.32	12.91/1.72	15.41/1.09	14.88/1.45	16.17/1.23	14.26/1.00	14.28/1.52	16.11/1.00	18.07/2.05	18.40/1.16
	SSIM	0.12/0.05	0.05/0.05	0.13/0.04	0.26/0.13	0.53/0.06	0.32/0.05	0.44/0.14	0.24/0.03	0.60/0.11	0.65/0.05
25%	RMSE	53.33/8.49	44.90/7.39	47.15/7.96	35.65/10.09	28.23/4.20	33.89/4.86	35.94/3.28	40.91/7.22	28.05/8.86	23.90/3.31
	PSNR	13.75/1.36	15.22/1.25	14.90/1.45	17.40/2.28	19.30/1.24	17.65/1.25	17.07/0.77	16.09/1.54	19.55/2.50	20.72/1.15
	SSIM	0.19/0.06	0.16/0.07	0.56/0.06	0.49/0.16	0.59/0.06	0.45/0.07	0.60/0.07	0.61/0.05	0.64/0.15	0.72/0.05

Fig. 2: The diagram of the reported lightweight complex-valued convolutional network (CCN). **a**, We optimize the parameter of CCN following the physical constraint (Eq. (4)) in an untrained manner to reconstruct the target scene. The CCN network is composed of a complex ConvBNReLU layer, five complex Res-blocks, and a complex Conv layer. **b**, The reconstruction results of various algorithms with the statistically sparse sampling.

Location: Supplement Note 8

“Reconstruction results of the test dataset

We tested the performance of the proposed method on the test set under various sampling ratios. As mentioned in the main text, the test set consists of 200 randomly selected MMW echo signals. We compared the proposed untrained learning method with RMA, CS-CG, ADMM, and deep learning techniques. The visual and numerical comparisons are shown in Fig. S8 and Fig. S9, respectively. We can conclude that:

- The statistically sparse array outperforms the randomly sparse array.
- The deep learning method, while performing well in certain scenarios, lacks robustness and may perform worse in certain cases.
- The untrained learning method outperforms the other comparative methods in cases of average reconstruction accuracy and robustness.”

Fig. S8: Visual comparisons of various algorithms. We tested the reconstructions from both random and statistically optimized arrays at 10% and 25% sampling ratios.

Fig. S9: The numerical comparisons between different reconstruction techniques including RMA, CS-CG, ADMM, Deep Learning (DL), and the proposed untrained learning. The test dataset contains 200 different echoes. The comparisons involve the evaluated RMSE(\downarrow), PSNR(\uparrow), and SSIM(\uparrow) values of 10% randomly sampled (a), 10% statistically sparse sampled (b), 25% randomly sampled (c), and 25% statistically sparse sampled (d) echoes.

References:

[36] Li, S., Zhao, G., Sun, H., Amin, M.: Compressive sensing imaging of 3-d object by a holo-

graphic algorithm. IEEE Trans. Antennas Propag. 66(12), 7295–7304 (2018)

[44] Boyd, S., Parikh, N., Chu, E., Peleato, B., Eckstein, J., et al.: Distributed optimization and statistical learning via the alternating direction method of multipliers. Found. Trends Mach. Learn. 3(1), 1–122 (2011)

[45] Chang, X., Zhao, R., Jiang, S., Shen, C., Zheng, G., Yang, C., Bian, L.: Complex-domain-enhancing neural network for large-scale coherent imaging. Adv. Photon. Nexus 2(4), 046006–046006 (2023)

<Comment 6>

Is the image for M in Fig. 1 the result of all the echos or an echo for a subject? Any insight for this M ? Why does it look like this? How does it relate to the physical traits of radar signals?

<Response>

We appreciate your valuable inquiry. The image for M is obtained from the result of all the echoes. The workflow of obtaining M can be summarized as 3 steps (see Supplement Note 2): collecting the echo dataset, obtaining the statistical importance map, and generating the sparse array design. We can further experimentally select a suitable M of best performance (see Supplement Note 3).

The echo can be represented by a convolution between the antenna beam and the target’s distribution. The amplitude of the echo indicates the average strength of the target’s reflection within the beam coverage. Meanwhile, the gradient of the echo’s phase conveys information about the smoothness of targets being illuminated. Thus, we combine both the amplitude and phase information of a large-scale human security echo dataset to formulate the statistical importance map \bar{M} . The magnitude order of the values on \bar{M} represents the importance of the corresponding antenna element at that position. Based on the statistical importance map, the sparse array pattern M is derived by Monte Carlo sampling. Therefore, M reflects the statistical information of the entire dataset. After numerical comparisons, the M with the best reconstruction performance corresponding to a suitable hyperparameter of Monte Carlo sampling is derived. A suitable M places a limited number of antenna elements in the most crucial positions, thereby reducing the cost of the antenna array while maintaining acceptable reconstruction and detection accuracy. We have included a detailed description in the revised supplement.

Location: Supplement Note 2

“The workflow of generating a statistically sparse array design

We summarize the overall workflow to obtain a statistically sparse array design M in Fig. S1 and as follows.

- **Collecting the echo dataset:** This workflow starts by collecting the 3D echo dataset. We have collected a large-scale 3D echo dataset which contains 1934 human security inspection echoes. Then we extract the amplitude and phase from the center-frequency cross-sections of 3D echoes.
- **Obtaining the statistical importance map:** The statistical importance map \bar{M} is the production of the averaged amplitude (\bar{A}) and inverse phase gradient (\bar{P}) of 2D cross-sections. This approach was adopted as an intuitive means of statistically integrating the significance of both amplitude and phase components. The averaged representations of these three distributions, (\bar{M} , \bar{A} , and \bar{P}) as depicted in Fig. S1, offer a more comprehensive portrayal of the outline and structural characteristics of all the subjects being tested. A higher value in \bar{M} denotes higher statistical importance of the element in the array, which is theoretically proofed in Sec. 4.2 and Supplement Note 3.
- **Generating the sparse array design:** Next, we perform Monte Carlo sampling to generate the sparse array design M . All element positions are assigned an importance order, where this importance order is the sorting of the element positions on \bar{M} from the highest value to the lowest value. Generate a uniformly random function $r(n)$ (seed = 100), where the horizontal axis is the importance order ($n = 1 \rightarrow N$, with 1 being most important and N being least important), and the vertical axis takes values from 0 to 1. Given a threshold S , search from 1 to N for n where $r(n) > S$, until the number meets the preset sampling ratio. These corresponding element positions of $r(n) > S$ are the positions retained in the sparse array design M . Different hyperparameter S correspond to different M , so we evaluated the performance of different M under 10% and 25% sampling ratios, and obtained the optimized sparse array designs as shown in Fig. 1 d in the main text.”

Fig. S1: The workflow for generating a statistically sparse array design. We start by collecting the 3D echo dataset and then extracting the center-frequency cross-sections of these echoes. The statistical importance map \bar{M} is the production of averaged amplitude map \bar{A} and inverse phase gradient map \bar{P} . The sparse array design is derived M by Monte Carlo sampling.

Location: Supplement Note 3

“Physical traits of the statistical importance map

Here, we provide a theoretical analysis to elucidate the principle of statistical importance ranking of MMW echoes. The bandwidth of the system is less than 20%. The scattering characteristics of targets do not vary significantly across different frequencies, as shown in Fig. S2. Therefore, we opt for the center-frequency cross-sections to reflect the overall echoes. The following analysis involves the physical traits behind the reported sparse array design method: higher values in amplitude \bar{A} , inverse phase gradient \bar{P} , and importance map \bar{M} correspond to heightened significance.

Amplitude \bar{A}

The amplitude of the echo distribution plays a pivotal role as an indicator of significance within the array. The MMW imaging system’s echo can be mathematically expressed as:

$$s(x', z', k) = \int \int \int \sigma(x, y, z) e^{-jk\sqrt{(R_0-y_0)^2+(x'-x)^2+(z'-z)^2}} dx dy dz. \quad (\text{S1})$$

Here, $\sigma(x, y, z)$ represents the scattering coefficient, and k denotes the wavenumber. The variables x' and z' denote the antenna location along the azimuth and height, respectively, and R_0 represents the distance from the target to the array.

Thus, a larger amplitude $\|s(x', z', k)\|$ indicates a larger $\sigma(x, y, z)$ within the beam coverage, corresponding to a higher average reflection. Consequently, a higher value in \bar{A} denotes a higher importance ranking for the corresponding array element.

Inverse phase gradient \bar{P}

To simplify the problem, we first discuss the one-dimensional case where the target is distributed along the x axis. Assuming that the Born approximation [S73] is satisfied during the scattering process with omitted propagation attenuation, the echo can be obtained as:

$$s(x') = \int \sigma(x) e^{-jk_0\sqrt{R_0^2+(x'-x)^2}} dx, \quad (\text{S2})$$

where k_0 denotes the wavenumber of the working single-frequency signal. Considering narrow antenna beam pattern, we have $\|x' - x\| \ll R_0$, then keep the first two terms of the Taylor expansions

of the expressions under root, Eq. (S2) can be rewritten as:

$$\begin{aligned} s(x') &\approx \int \sigma(x) e^{-jk_0 \left[R_0 + \frac{(x'-x)^2}{2R_0} \right]} dx \\ &= e^{-jk_0 R_0} \int \sigma(x) e^{-jk_0 \frac{(x'-x)^2}{2R_0}} dx. \end{aligned} \quad (\text{S3})$$

The gradient along the x -axis can be expressed as:

$$\frac{ds(x')}{dx'} = C \int \sigma(x) e^{-jk_0 \frac{(x'-x)^2}{2R_0}} (2x' - 2x) dx, \quad (\text{S4})$$

where

$$C = j \frac{k_0}{2R_0} e^{-jk_0 R_0}. \quad (\text{S5})$$

Let us focus on the gradient around the element where only the target within its beamwidth can be illuminated,

$$\frac{ds(x')}{dx'} = -2C \int_{x'-x_{\max}}^{x'+x_{\max}} \sigma(x) e^{-jk_0 \frac{(x'-x)^2}{2R_0}} (x - x') dx, \quad (\text{S6})$$

where

$$x_{\max} = R_0 \tan \frac{\Theta_h}{2}. \quad (\text{S7})$$

Herein, Θ_h represents the antenna beamwidth. By employing the technique of integration by parts, we can derive the following expression:

$$\begin{aligned} \frac{ds(x')}{dx'} &= \frac{2R_0 C}{k_0} \left\{ e^{-jk_0 \frac{x_{\max}^2}{2R_0}} [\sigma(x' - x_{\max}) - \sigma(x' + x_{\max})] + \right. \\ &\quad \left. \int_{x'-x_{\max}}^{x'+x_{\max}} \sigma'(x) e^{-jk_0 \frac{(x'-x)^2}{2R_0}} dx \right\}, \end{aligned} \quad (\text{S8})$$

The first row in Eq. (S8) elucidates the influence of the target amplitude near the boundary of the antenna beamwidth. The second row in the same equation delineates the impact of target fluctuations. When the imaging target exhibits a relatively planar structure within the antenna beamwidth, as is often the case in human imaging scenarios, both of these terms tend to approach zero, specifically $\frac{ds(x')}{dx'} = 0$. Conversely, in situations characterized by fluctuating imaging targets or a void in the imaging area where noise dominates the echo, both terms in Eq. (S8) deviate from zero. Under certain conditions, the accumulation of terms in the second row of Eq. (S8) can result

in a significant $\frac{ds(x')}{dx'}$.

In summation, when the target's spatial distribution is relatively uniform, the gradient between adjacent elements tends to be small. Conversely, for targets characterized by fluctuating scattering coefficients or those resembling noise-like patterns, the gradient becomes notably larger. The analysis presented above leads to the conclusion that the construction of a sparse array with small element gradients can enhance the quality of the illumination for the specific target of interest. That is, a larger value in \bar{P} indicates that the corresponding element has a higher importance ranking.

Importance map $\bar{M} = \bar{A} \times \bar{P}$

The statistical importance map $\bar{M} = \bar{A} \times \bar{P}$ reflects both the amplitude and phase distribution of the echo dataset. Similarly, a larger value in \bar{M} indicates a higher importance of the corresponding element in the array. Following the workflow in Fig. S1, the sparse array design is generated. In the third step of this workflow, we can vary the hyperparameter S to find a suitable coverage range and sparsity of the sampling pattern. By numerical comparisons of randomly selected echoes (Fig. S3), the optimized statistical sparse arrays at 10% and 25% sampling ratios correspond to S equals 0.8 and 0.5, respectively.”

Fig. S2: The exemplar 3D echo amplitudes and phases, and the corresponding center-frequency amplitude and inverse phase gradient cross-sections.

Fig. S3: Numerical comparisons of reconstruction accuracy w.r.t. different S when the sampling ratio equals 10% (**a**) and 25% (**b**). The error bar represents the standard deviation. The optimal sparse patterns of SR=10%&25% correspond to $S = 0.8$ & 0.5 , respectively.

References:

[S73] Gubernatis, J., Domany, E., Krumhansl, J., Huberman, M.: The born approximation in the theory of the scattering of elastic waves by flaws. J. Appl. Phys. 48(7), 2812–2819 (1977)

<Comment 7>

No array or used signals from imaging resulting from the sampling are shown.

<Response>

Thank you for your thorough review. The prototype emits Linear Frequency Modulated (LFM) signal and receives echoes with the dechirp technique. We have included the whole MMW security check workflow and intermediate results of each procedure, including the sparse array pattern, MMW echo signals, and reconstruction and detection results, in the revised supplement.

Location: Supplement Note 9

“The workflow of the reported MMW security check framework

We summarize the workflow of the reported MMW security check framework as shown in Fig. S10. The framework consists of three main techniques:

1. Sparse echo acquisition: We first acquire the sparse echo of the subject using the statistically optimized sparse antenna array.
2. Untrained reconstruction: We reconstruct the 3D MMW image from the sparse echo using the untrained learning method.
3. Detection: We finally apply the YOLO network to detect the concealed targets with the reconstructed image.

Above all, we have established a complete MMW security check workflow encompassing sparse echo acquisition, reconstruction, and detection. Based on this framework, it is promising to achieve a highly reliable, low-cost, and high-throughput MMW security check system.”

Fig. S10: The workflow of the reported MMW security check framework (a) and the results of each procedure under 10% and 25% sampling ratios (b).

<Comment 8>

To summarize, the proposed method leveraging machine learning to help the radar array design is novel and inspiring. It has potential contributions to the community of MMW imaging. However, the solidness of the work still requires further improvement regarding the study of the key design of the algorithms and the generalizations and insight of the learned results. Therefore, a major revision is suggested.

<Response>

Thank you for your thoughtful feedback. We greatly appreciate your positive assessment of our work. To further improve the solidity of the work, we have conducted extensive additional experiments for a more comprehensive description of the reported sparse array design, untrained learning reconstruction, the applicability and generalization of our system, etc. We have included the following revisions in the main text and supplement:

- Section 2.2 and Table 1: Add the reconstruction results of deep learning methods.
- Section 2.3 and Figure 3: Revise the detection results of the networks trained with full-sampled reconstructions. Add the detection results of the networks trained with sparsely sampled reconstructions.
- Section 2.4 and Figure 4: Add the applicability and generalization of the reported system.
- Figure 5: Add the comparison of spatial resolutions of random and statistically optimized arrays.
- Supplement Note 1: A review of MMW sparse array synthesis and imaging.
- Supplement Note 2: The workflow of generating a statistically sparse array design
- Supplement Note 3: Physical traits of the statistical importance map
- Supplement Note 4: Relationship between the sparse array topology and imaging results
- Supplement Note 5: Comparison with mechanical scanning arrays
- Supplement Note 6: Comparison with commonly used sparse arrays

- Supplement Note 7: Details about the learning-based reconstruction network
- Supplement Note 8: Reconstruction results of the test dataset
- Supplement Note 9: The workflow of the reported MMW security check framework
- Supplement Note 10: Additional details of applicability and generalization
- Supplement Note 11: Additional details of the untrained learning reconstruction
- Supplement Note 12: The influence of interpolation in the scattering process
- Supplement Note 13: Complexity of CCN
- Supplement Note 14: Additional details about the detection network
- Supplement Note 15: Discussion about transmit and receive antenna isolation and the calibration procedure

[Reviewer 2]

<Comment 1>

This paper describes the development of an imaging system operating in the millimeter range. The images obtained using this system are highly convincing, demonstrating the team's ability to master all the engineering elements required for the successful operation of these complex systems.

<Response>

We appreciate the reviewer's recognition of our efforts and the positive evaluation of our work.

<Comment 2>

According to my understanding of the paper, the authors first propose an approach inspired by compressive sensing to alleviate the hardware complexity inherent in the proper operation of these body scanners. The positioning in relation to the scientific literature seems to me incomplete on this first point, not allowing me to identify any real added value.

<Response>

We appreciate your inquiry. We have revised the introduction section and added a brief literature review in the supplement about recent relevant works to clarify the innovations of our reported framework.

1) **Sparse array synthesis (SAS):** State-of-the-art SAS methods based on optimization struggle to synthesize large arrays (up to 121×121 as the original full array)[S42]. Besides, the imaging regions of these methods are relatively limited. However, our work, for the first time, proposes to incorporate statistical sparse design into MMW imaging, leveraging statistical insights to prioritize significant elements. In this statistically sparse design, there is no need for large matrix calculations. Thus, it is more suitable for designing large sparse arrays. To our knowledge, this work reports the largest sparse antenna array design (430×186 as the full array) and achieves successful MMW security inspection with a sampling ratio as low as 10% with the target scale of $1.85\text{m} \times 1.00\text{m} \times 0.5\text{m}$.

2) **Reconstruction:** In contrast to existing CS methods that might fail under low sampling ratios, our study, for the first time, introduces the untrained learning approach for MMW imaging, which enables accurate and robust reconstruction under low sampling ratios. To accommodate the

complex characteristics inherent to MMW imaging modalities, we have integrated the Complex-valued Convolutional Network (CCN) into the untrained learning framework for reconstruction. The inherent regularization within the network structure contributes to an accurate and robust reconstruction [38,39]. In our experiments, the untrained learning method outperforms the existing RMA, CS-CG, ADMM, and deep learning algorithms, and showcases relatively strong robustness among these methods.

3) **Imaging and detection scheme:** We provide a complete MMW security check framework, including statistically sparse array design, untrained learning reconstruction and detection, which is promising to realize a high-fidelity, low-cost, and high-throughput MMW security check system. Empirical results substantiate the successful detection of concealed centimeter-sized targets using a sparse array with a sampling ratio as low as 10%.

We present these revisions in a structured manner as follows:

Location: Main text, Section 1, Line 64 - 142

“The Sparse Array Synthesis (SAS) [20-25] technique is emerging as a cost-effective alternative to minimize manufacturing expenses of the fully sampled arrays. However, state-of-the-art SAS methods struggle to synthesize large arrays [25], making them hard to apply to security check scenarios where a large aperture is usually required. The next challenge lies in the reconstruction methods for sparse array-based systems. Existing Compressive Sensing (CS) methodologies [26-29] report their effectiveness in many scenarios. However, they might fail under low sampling ratios with large-scale arrays. Deep Learning (DL) approaches have flourished in recent years. Some are designed to enhance pre-reconstructed images [30, 31] while others directly deal with the inverse scattering problem [32-35], yielding commendable reconstruction results in specific datasets. However, these methodologies heavily rely on training datasets, potentially leading to poor generalization for untrained or marginal distributions.

Considering these challenges, we present a low-cost, large-scale single-shot MMW security inspection framework capable of successfully detecting concealed centimeter-sized targets. This framework encompasses the entire inspection workflow, incorporating statistically sparse array design, untrained learning reconstruction, and detection processes.

Instead of regularly arranged or optimization-based approaches, we introduce a statistically sparse array design method that fully exploits the echoes of existing full-sampled arrays and easily

scales to any size free from heavy computation. Specifically, we first collected a set of real-captured echoes using a scanning-based full-sampled antenna array, and then analyzed the statistical importance ranking map of array elements. We theoretically proved that a larger value in the map indicates a more significant element in the antenna array. Based on the ranking map, we experimentally derived a statistically optimized sampling strategy to sparsely select antenna elements at various sampling ratios. Leveraging this approach, we developed a large-scale sparse array (430×186 as the full array) designed for MMW near-field imaging. At low sampling ratios (e.g., 10% and 25%), the technique shows an average 20% RMSE improvement, 2dB higher for PSNR, and 0.22 SSIM improvement compared to random sampling. This statistical approach allows for a more efficient selection of the most significant elements, thereby reducing the number of inefficient elements in antenna array design.

Moreover, to attain high-quality images using the statistically optimized array, we propose an interpretable untrained learning approach. This approach ensures robust MMW reconstruction from sparsely sampled echoes. We optimize a lightweight complex-valued network based on the objective derived from the physical model of MMW scattering. Unlike conventional DL imaging methods [30-35], this physics-informed learning strategy operates without the need for training, ensuring case-specific optimization for robustness. Compared to the existing CS-based approaches, it yields high-fidelity reconstruction with up to 3.4 dB PSNR improvement at 10% and 25% sampling ratios. Additionally, the technique shows prominent priority over existing techniques for detecting concealed targets from sparsely sampled echoes. The combination of statistics-based sparse sampling and untrained learning results in a significant reduction in the cost of the antenna array by up to an order of magnitude, without compromising the system's ability to detect centimeter-sized concealed targets.

Furthermore, we employ a neural network for the automatic detection of concealed targets. The reported framework combining statistical sparse array and untrained reconstruction shows closer detection accuracy to the full-sampled detection results than existing approaches, which indicates superior reconstruction accuracy. Experiments demonstrated that the reported imaging framework achieved successful concealed target detection at a 10% sampling ratio, while other contemporary approaches failed. This work leverages the statistical prior and emerging fusion frameworks of neural networks and optimization, providing new insights into the development of low-cost and

efficient MMW security inspection systems.”

Location: Supplement Note 1

“Sparse array synthesis for near-field imaging

Imaging with full arrays demands large arrays, resulting in unaffordable manufacturing costs. Sparse Array Synthesis (SAS) serves as a solution to this challenge. We delve into existing SAS literature to provide a brief overview, with relevant references in Tab. S2.

Some works [S31, S33, S40] proposed sparse arrays with fixed array topologies. However, it is imperative to acknowledge that these topological configurations are bound by practical constraints and exhibit limited flexibility. Some works have resorted to optimization techniques, such as the Simulated Annealing algorithm (SA) [S41] and the Particle Swarm Optimization (PSO) method [S30], to achieve SAS for near-field imaging.

To address the challenges posed by the stochastic optimization methods, efforts have been made to design sparse array topologies that capitalize on principles of uniformity and reduced element shadowing [S34, S35]. While these designs have yielded arrays with favorable focusing and sidelobe-suppression properties, it is pertinent to note that stochastic optimization approaches are computationally intensive and susceptible to issues of local convergence, particularly when executed over limited random trials. On the contrary, convex optimization methods [S39], [25] offer a promising avenue for SAS. However, it’s hard to apply the optimization-based design methods for very large arrays (such as the reported 430×186 -element array), given the substantial computational burden they impose in terms of both time and memory requirements. Moreover, while the previously mentioned methods demonstrate satisfactory performance with relatively limited target scales, as illustrated in Table S2, they may not yield favorable reconstructed images for larger targets.

In contrast to the existing paradigms of artisanally crafted and optimized arrays, our research reports a novel statistically sparse array design method. This approach hinges upon the collection of full-sampled real echoes and leverages statistical insights to prioritize significant elements while simultaneously reducing the number of inefficient ones. This innovative statistical methodology does not rely on handcraft design or resource-consuming optimization which are hard to employ on large-scale array design, achieving advanced imaging quality for a large-scale (430×186 -element) array.

Table S2: Recent advancements in sparse array synthesis methods for MMW near-field imaging

Ref	Scale(m)	Elements	Array Form	Distance(cm)	Band(GHz)	Step(MHz)	Sparsity(%)	PSL(dB)	Target scale(m)	Year
[S30]	0.5	20	linear	100	10.0-18.0	-	-	-	0.3(ID)	2008
[S31]	0.425×0.425	20	planar	75	5-25	-	-	-	0.07 × 0.02(2D)	2009
[S26]	0.5	12	linear (SAR)	50	2.8-19.5	-	23.5	-	0.2 × 0.2 × 0.1(3D)	2010
[S32]	0.4 × 0.4	25	planar	50	3-19.5	-	-	-18	0.2 × 0.3 × 0.1(3D)	2010
[S28]	0.65	58	linear(SAR)	60	75-90	117	20	-	0.6 × 0.5 (2D)	2011
[S33]	0.5×0.5	25	planar	50	9.5-12.5	-	-	-22	0.3 × 0.2 × 0.1(3D)	2012
[21]	0.707	26	curvilinear(SAR)	31	56-62	400	-	-	0.3 × 0.3 × 0.2(3D)	2013
[S34]	0.48×0.48	20	planar	50	2.14-9.04	-	-	-19.04	0.54 × 0.54 (2D)	2015
[S35]	0.5×0.5	20	planar	50	2.55-7.89	-	-	-	0.54 × 0.54(2D)	2016
[S36]	0.4×0.4	6	planar	28	77.38-80.93	-	-	-	0.35 × 0.2(2D)	2019
[S37]	0.425×0.425	24	spiral	60	3.5-8.5	-	-	-17.7	0.35 × 0.2(2D)	2020
[S38]	3	6	linear	4	1-3	17.9	-	-	2.0 × 1.0(2D)	2021
[S39]	0.5×0.5	68	T-shaped	120	90-96	68	33.7	-	0.15 × 0.15 × 0.1(3D)	2023
[25]	0.6×0.6	3732	planar	50	30-35	50	25.5	-	0.2 × 0.1 × 0.1(3D)	2023
Our	1.85 × 1.00	7998	planar	30-50	32-37	100	10.0	-	1.85 × 1.00 × 0.5(3D)	2023

We depart from the convention of employing regularly arranged arrays or scanning arrays by introducing a sparsely optimized antenna array tailored for single-shot MMW imaging. We commence with the collection of real-captured echoes through a full-sampled antenna array and subsequently analyze the statistical importance ranking map of array elements (see Supplement Note 2). It is theoretically substantiated that a higher value within this ranking map signifies the significance of an element within the antenna array (see Supplement Note 3). Guided by this ranking map, we experimentally derive a statistically optimized sampling strategy to selectively include antenna elements at various sampling ratios. Our experimental results, achieved at low sampling ratios, such as 25%, reveal a substantial enhancement in Root Mean Square Error (RMSE) by 20.6%, an increase in Peak Signal-to-Noise Ratio (PSNR) by 2 dB, and a boost in Structural Similarity index (SSIM) [37] by 0.22 when compared to conventional random sampling strategies.

Compressive Sensing(CS)-based reconstruction algorithms for sparse arrays

Sparse arrays employ specific algorithms to achieve effective image reconstruction. Several works have explored the application of various CS-based techniques to acquire 2D images [S42-S44], [26, 27, 28]. While CS imaging has found application in sparse Inverse Synthetic Aperture Radar (ISAR) imaging [27, 28] and two-dimensional scenarios [S43], it is essential to acknowledge that persisting challenges concerning high memory consumption and computational complexity have remained unaddressed. Furthermore, these methodologies typically prove inadequate when it comes to reconstructing meaningful 3D images. The endeavors to develop 3D image reconstruction techniques grounded in the principles of CS are presented in [29], [36], [S45-S47].

Specifically, one avenue of exploration involves the utilization of the Diffraction Tomography (DT) method [29] and the Two-Level Block Matching Pursuit (TLBMP) algorithm [S45] in conjunction with CS for 3D Through-the-Wall Radar Imaging (TWRI). Unfortunately, these approaches have been associated with challenges such as low resolution and protracted reconstruction times. Furthermore, an interpolation-free holographic imaging algorithm has been developed as part of the CS iterative optimization process [36]. Nevertheless, comparative evaluations have revealed that this algorithm fails to achieve comparable reconstruction results under similar conditions.

In contrast to these methodologies, our study introduces the interpretable untrained learning approach, which enables accurate and robust MMW reconstruction from sparsely sampled echoes.

Instead of training-based techniques, it optimizes a lightweight complex-valued network using an objective function rooted in the physical model of MMW scattering. Notably, this physics-informed learning strategy obviates the need for traditional training procedures, effectively overcoming the black-box limitations and underperforming in edge cases associated with training-based techniques. Comparative assessments against existing CS-based approaches showcase the superior performance of our approach, manifesting in a 2.7 dB improvement in PSNR at low sampling ratios. Moreover, our technique exhibits pronounced advantages in the context of detecting concealed targets from sparsely sampled echoes.

MMW near-field imaging based on full arrays

Here we present a review of recent MMW near-field imaging methods, with a specific focus on full arrays. The references related to this discussion are provided in Tab. S1. The referenced techniques in this section build the foundation of MMW security check imaging methods.

It is imperative to introduce the concept of monostatic arrays. The utilization of a full Single-Input Single-Output (SISO) array entails a substantial number of elements, thereby resulting in prohibitively high manufacturing costs. Consequently, mechanically scanning linear array systems have found widespread application in conjunction with the Range Migration Algorithm (RMA) [7], [S16] to facilitate real-time imaging. Key works in this domain include ref. [S17-S22].

However, it is imperative to note that mechanically scanning linear array systems are challenged by the time-consuming scanning procedure, rendering them unsuitable for one-shot imaging. In practice, it is challenging to ensure complete steadiness, especially in cases involving slight movements, such as body swaying or shaking, particularly among elderly individuals. Such movements can significantly degrade image quality (see Supplement Note 5). Given that state-of-the-art detection and classification methods, including deep neural networks, often rely on precise object edge and feature detection within images [14], blurred images with distorted object features can substantially compromise detection accuracy. Additionally, even slight subject movements during data acquisition lead to blurred images [13], thus compromising the accuracy of subsequent concealed object detection algorithms [14], [S23].

In contrast, MIMO arrays offer the advantage of data collection through a single snapshot, necessitating fewer elements [19]. A typical MIMO imaging prototype involves square clutters [S24], wherein the computationally intensive back-projection algorithm [15] is employed for image recon-

Table S1: Recent advancements in active near-field imaging algorithms. The gray background indicates the algorithm used for Ka-band imaging.

Ref	Imaging System	Distance(cm)	Band(GHz)	Step(MHz)	Elements	Complexity	Resolution ^a (mm)	Time(s)	Year
[15]	2D SISO	-	33.2-33.8	-	-	$O(N^3)$	-	-	1992
[S1]	SISO-SAR	230	15.5-17.5	5	2	-	75/20/20	103	2000
[7]	SISO-SAR	-	27-33	-	128	-	-	-	2001
[S2]	SISO-SAR	230	15.5-17.5	-	2	-	75/20/20	65	2003
[S3]	2D MIMO	30	2.66	-	-	-	-	19.2	2008
[S4] ^b	2D SISO	2000	34.5-35.5	14.5	-	$O(JM) + O(2N \log_2 N)^c$	-	1.13	2010
[12]	2D MIMO	50	1-26	100	25	-	9.1/12.4/12.4	12	2012
[S5]	SISO SAR	3.4	35.04-44.64	640	-	-	-	218/340	2014
[17]	2D MIMO	35	60-66	600	-	-	25/15/19	9000	2014
[S6]	MIMO-SAR	32	130-150	100	45	$O(N^4 \log_2 N)$	-	187.52	2017
[18]	MIMO-SAR	-	27-31	80	64	-	-	0.13	2018
[S7]	MIMO-CSAR	30	30-36	200	58	$O(N^3 \log_2 N)$	-	123.6	2018
[S8]	MIMO-SAR	30	30-36	200	52	$O(N^4 \log_2 N)$	-	34.7	2018
						$O(N^4)$	-	9.1	
[16]	2D MIMO	100	18-26	-	142	-	12.8/7.6/7.6	27.1	2019
						-	17.8/7.6/7.6	24.8	
[S9]	2D MIMO	50	5-18	100	25	$O(N^5 \log N)$	-	79	2020
						-	-	125	
[S10]	2D MIMO	50	15-30	300	32	$O(N^5 \log_2 N)$	-	17.3	2020
[S11]	MIMO-SAR	50	27-33	150	50	$O(N^4 \log_2 N)$	6.96/6.52/-	0.22	2020
[S12]	MIMO-SAR	50	30-36	120	28	$O(N^6)$	-	228.38	2020
[S13]	MIMO-SAR	100	92.125-107.875	525	45	$O(N^4)$	4.5/4.4/8.4	12	2020
[S14]	MIMO-CSAR	-	10	-	38	-	-	0.057	2021
[S15]	MIMO-CSAR	25	27-33	100	56	$O(N^4 \log_2 N)$	-	303	2021
Our	2D SISO	30-50	32-37	100	7998	$O(N^5)$	30/5/5	10.4	2023

^a Range/Azimuth/Height resolution

^b Despite the imaging distance spanning 20 meters, the target remains well within the near-field region of the array, primarily due to the considerable size of the array itself.

^c $O(JM)$ represents the “gridding” step in the algorithm, where J is the number of the nearest neighbors to the nonuniform digital frequency in radians.

struction. Several Fourier-based imaging algorithms have been developed for acceleration [12,16-19]. However, these methods struggle to achieve real-time imaging due to the higher-dimensional processing when compared to SISO arrays. Phase calibration methods can be employed to attain real-time imaging but may introduce image distortions [2], [S25]. Consequently, the cost-efficient MIMO arrays often confront challenges in simultaneously achieving rapid and high-quality imaging.

A parallel avenue of exploration involves the amalgamation of linear or arc MIMO arrays with mechanical scanning structures (MIMO-SAR) for enhanced cost-effectiveness. MIMO linear arrays can be configured to scan along straight [S8, S26, S27] or circular tracks [S7, S13, S18]. In a notable departure from this, the transformation from arc arrays to equivalent linear arrays has been proposed [S14] to enable azimuthal imaging for circular-arc MIMO arrays. Furthermore, a full wavenumber domain 3D imaging algorithm [S29] has been devised within this framework. However, akin to MIMO arrays, MIMO-SAR schemes grapple with protracted imaging times attributed to the higher-dimensional processing. Moreover, they are unable to achieve one-shot imaging through mechanical scanning.

Our study introduces an innovative scheme for low-cost, large-scale MMW imaging tailored for single-shot human security inspection. This approach involves a complete workflow of statistically sparse echo acquisition, untrained learning reconstruction, and detection. The reported statistically sparse array design contributes to constructing a $430 \times 186 \times 10\%$ sparse array based on the statistics of human bodies' echoes. We incorporate the interpretable untrained learning strategy for robust reconstruction from sparsely sampled echoes. By physics-informed learning without a conventional training process, we realize efficient and high-fidelity reconstruction, culminating in state-of-the-art reconstruction performance. The confluence of statistics-based sparse sampling and untrained learning serves to substantially reduce the cost of antenna arrays by an order of magnitude while preserving the capability to detect concealed centimeter-sized targets. This research amalgamates the merits of statistical priors and the burgeoning fusion framework of neural networks and optimization, thus opening novel vistas in the realm of low-cost and efficient MMW imaging.”

References:

- [5] Li, L., Ruan, H., Liu, C., Li, Y., Shuang, Y., Al'u, A., Qiu, C.-W., Cui, T.J.: Machine-learning reprogrammable metasurface imager. *Nat. Commun.* 10(1), 1082 (2019)
- [6] Lynch, J.J., Moyer, H.P., Schaffner, J.H., Royter, Y., Sokolich, M., Hughes, B., Yoon, Y.J., Schulman, J.N.: Passive millimeter-wave imaging module with preamplified zero-bias detection. *IEEE Trans. Microw. Theory Techn.* 56(7), 1592–1600 (2008)
- [7] Sheen, D.M., McMakin, D.L., Hall, T.E.: Three-dimensional millimeter-wave imaging for concealed weapon detection. *IEEE Trans. Microw. Theory Techn.* 49(9), 1581–1592 (2001)
- [8] Liu, C., Ma, Q., Luo, Z.J., Hong, Q.R., Xiao, Q., Zhang, H.C., Miao, L., Yu, W.M., Cheng, Q., Li, L., et al.: A programmable diffractive deep neural network based on a digital-coding metasurface array. *Nat. Electron.* 5(2), 113–122 (2022)
- [9] Hunt, J., Driscoll, T., Mrozack, A., Lipworth, G., Reynolds, M., Brady, D., Smith, D.R.: Metamaterial apertures for computational imaging. *Science* 339(6117), 310–313 (2013)
- [10] Cui, T.J., Qi, M.Q., Wan, X., Zhao, J., Cheng, Q.: Coding metamaterials, digital metamaterials and programmable metamaterials. *Light Sci. Appl.* 3(10), 218–218 (2014)
- [11] Li, L., Jun Cui, T., Ji, W., Liu, S., Ding, J., Wan, X., Bo Li, Y., Jiang, M., Qiu, C.-W., Zhang, S.: Electromagnetic reprogrammable coding-metasurface holograms. *Nat. Commun.* 8(1), 197 (2017)
- [12] Zhuge, X., Yarovoy, A.G.: Three-dimensional near-field MIMO array imaging using range migration techniques. *IEEE Trans. Image Process.* 21(6), 3026–3033 (2012)
- [13] Liu, H., Wang, S., Jing, H., Li, S., Zhao, G., Sun, H.: Millimeter-wave image deblurring via cycle-consistent adversarial network. *Electronics* 12(3), 741 (2023)
- [14] Kupyn, O., Budzan, V., Mykhailych, M., Mishkin, D., Matas, J.: DeblurGAN: Blind motion deblurring using conditional adversarial networks. In: *IEEE/CVF Conf. Comput. Vis. Pattern Recogn.*, pp. 8183–8192 (2018)
- [15] Desai, M.D., Jenkins, W.K.: Convolution backprojection image reconstruction for spotlight mode synthetic aperture radar. *IEEE Trans. Image Process.* 1(4), 505–517 (1992)
- [16] Fromenteze, T., Yurduseven, O., Berland, F., Decroze, C., Smith, D.R., Yarovoy, A.G.: A transverse spectrum deconvolution technique for MIMO short-range fourier imaging. *IEEE Trans. Geosci. Remote Sens.* 57(9), 6311–6324 (2019)
- [17] Alvarez, Y., Rodriguez-Vaqueiro, Y., Gonzalez-Valdes, B., Mantzavinos, S., Rappaport, C.M.,

- Las-Heras, F., Martínez-Lorenzo, J. A.: Fourier-based imaging for multistatic radar systems. *IEEE Trans. Microw. Theory Techn.* 62(8), 1798–1810 (2014)
- [18] Abbasi, M., Shayei, A., Shabany, M., Kavehvash, Z.: Fast fourier-based implementation of synthetic aperture radar algorithm for multistatic imaging system. *IEEE Trans. Instrum. Meas.* 68(9), 3339–3349 (2018)
- [19] Li, S., Wang, S., Amin, M.G., Zhao, G.: Efficient near-field imaging using cylindrical MIMO arrays. *IEEE Trans. Aerosp. Electron. Syst.* 57(6), 3648–3660 (2021)
- [20] Yang, B., Zhuge, X., Yarovoy, A., Ligthart, L.: UWB MIMO antenna array topology design using PSO for through dress near-field imaging. In: *Eur. Microw. Conf.*, pp. 1620–1623 (2008). IEEE
- [21] Gonzalez-Valdes, B., Allan, G., Rodriguez-Vaqueiro, Y., Alvarez, Y., Mantzavinos, S., Nickerson, M., Berkowitz, B., Marti, J., Las-Heras, F., Rappaport, C.M., et al.: Sparse array optimization using simulated annealing and compressed sensing for near-field millimeter wave imaging. *IEEE Trans. Antennas Propag.* 62(4), 1716–1722 (2013)
- [22] Tan, K., Wu, S., Wang, Y., Ye, S., Chen, J., Fang, G.: A novel two-dimensional sparse MIMO array topology for UWB short-range imaging. *IEEE Antennas Wireless Propag. Lett.* 15, 702–705 (2015)
- [23] Tan, K., Wu, S., Wang, Y., Ye, S., Chen, J., Liu, X., Fang, G., Yan, S.: On sparse MIMO planar array topology optimization for uwb near-field high-resolution imaging. *IEEE Trans. Antennas Propag.* 65(2), 989–994 (2016)
- [24] Wang, S., Li, S., Hoorfar, A., Miao, K., Zhao, G., Sun, H.: Compressive sensing based sparse MIMO array synthesis for wideband near-field millimeter-wave imaging. *IEEE Trans. Aerosp. Electron. Syst.* (2023)
- [25] Wang, S., Li, S., Ren, B., Miao, K., Zhao, G., Sun, H.: Convex optimization-based design of sparse arrays for 3-D near-field imaging. *IEEE Sensors J.* (2023)
- [26] Coker, J.D., Tewfik, A.H.: Compressed sensing and multistatic SAR. In: *Int. Conf. Acoust. Speech Signal Process.*, pp. 1097–1100 (2009). IEEE
- [27] Li, S., Zhao, G., Zhang, W., Qiu, Q., Sun, H.: ISAR imaging by two-dimensional convex optimization-based compressive sensing. *IEEE Sensors J.* 16(19), 7088–7093 (2016)
- [28] Li, S., Zhao, G., Li, H., Ren, B., Hu, W., Liu, Y., Yu, W., Sun, H.: Near-field radar imaging

- via compressive sensing. *IEEE Trans. Antennas Propag.* 63(2), 828–833 (2014)
- [29] Barzegar, A.S., Cheldavi, A., Sedighy, S.H., Nayyeri, V.: 3-D through-the-wall radar imaging using compressed sensing. *IEEE Geosci. Remote Sens. Lett.* 19, 1–5 (2021)
- [30] Ichikawa, K., Hirose, A.: Singular unit restoration in InSAR using complex-valued neural networks in the spectral domain. *IEEE Trans. Geosci. Remote Sens.* 55(3), 1717–1723 (2016)
- [31] Hu, C., Wang, L., Li, Z., Zhu, D.: Inverse synthetic aperture radar imaging using a fully convolutional neural network. *IEEE Geosci. Remote Sens. Lett.* 17(7), 1203–1207 (2019)
- [32] Li, L., Wang, L.G., Teixeira, F.L., Liu, C., Nehorai, A., Cui, T.J.: DeepNIS: Deep neural network for nonlinear electromagnetic inverse scattering. *IEEE Trans. Antennas Propag.* 67(3), 1819–1825 (2018)
- [33] Wei, Z., Chen, X.: Deep-learning schemes for full-wave nonlinear inverse scattering problems. *IEEE Trans. Geosci. Remote Sens.* 57(4), 1849–1860 (2018)
- [34] Rostami, P., Zamani, H., Fakharzadeh, M., Amini, A., Marvasti, F.: A deep learning approach for reconstruction in millimeter-wave imaging systems. *IEEE Trans. Antennas Propag.* 71(1), 1180–1184 (2022)
- [35] Bao, J., Li, D., Li, S., Zhao, G., Sun, H., Zhang, Y.: Fine-grained image generation network with radar range profiles using cross-modal visual supervision. *IEEE Trans. Microw. Theory Techn.* (2023)
- [S1] Lopez-Sanchez, J.M., Fortuny-Guasch, J.: 3-D radar imaging using range migration techniques. *IEEE Trans. Antennas Propag.* 48(5), 728–737 (2000)
- [S2] Gimeno-Nieves, E., Lopez-Sanchez, J., Pascual-Villalobos, C.: Extension of the chirp scaling algorithm to 3-D near-field wideband radar imaging. *IEE Proc. Radar Sonar Navigation* 150(3), 152–157 (2003)
- [S3] Zhuge, X., Savelyev, T., Yarovoy, A.G., Ligthart, L.: UWB array-based radar imaging using modified kirchhoff migration. In: *IEEE Int. Conf. Ultra-Wideband*, vol. 3, pp. 175–178 (2008). IEEE
- [S4] Li, S., Sun, H., Zhu, B., Liu, R.: Two-dimensional nufft-based algorithm for fast near-field imaging. *IEEE Antennas Wireless Propag. Lett.* 9, 814–817 (2010)
- [S5] Yang, Z., Zheng, Y.R.: A comparative study of compressed sensing approaches for 3-D synthetic aperture radar image reconstruction. *Digit. Signal Process.* 32, 24–33 (2014)

- [S6] Zhu, R., Zhou, J., Jiang, G., Fu, Q.: Range migration algorithm for near-field MIMO-SAR imaging. *IEEE Geosci. Remote Sens. Lett.* 14(12), 2280–2284 (2017)
- [S7] Gao, J., Deng, B., Qin, Y., Wang, H., Li, X.: An efficient algorithm for MIMO cylindrical millimeter-wave holographic 3-D imaging. *IEEE Trans. Microw. Theory Techn.* 66(11), 5065–5074 (2018)
- [S8] Gao, J., Qin, Y., Deng, B., Wang, H., Li, X.: Novel efficient 3D short-range imaging algorithms for a scanning 1D-MIMO array. *IEEE Trans. Image Process.* 27(7), 3631–3643 (2018)
- [S9] Wang, J., Aubry, P., Yarovoy, A.: 3-D short-range imaging with irregular MIMO arrays using NUFFT-based range migration algorithm. *IEEE Trans. Geosci. Remote Sens.* 58(7), 4730–4742 (2020)
- [S10] Tan, K., Chen, X.: Fast 3-D image reconstruction on nonregular UWB sparse MIMO planar array using scaling techniques. *IEEE Trans. Microw. Theory Techn.* 69(1), 222–234 (2020)
- [S11] Tan, K.: A fast Omega-K algorithm for near-field 3-D imaging of MIMO synthetic aperture radar data. *IEEE Geosci. Remote Sens. Lett.* 18(8), 1431–1435 (2020)
- [S12] Wang, F., Deng, B., Yang, Q., Wang, H., Zhang, Y.: A fast 3D near range imaging algorithm for a scanning sparse MIMO array in the millimeter band. *Sensors* 20(17), 4701 (2020)
- [S13] Gao, H., Li, C., Wu, S., Geng, H., Zheng, S., Qu, X., Fang, G.: Study of the extended phase shift migration for three-dimensional MIMO-SAR imaging in terahertz band. *IEEE Access* 8, 24773–24783 (2020)
- [S14] Wu, S., Wang, H., Li, C., Liu, X., Fang, G.: A modified Omega-K algorithm for near-field single-frequency MIMO-arc-array-based azimuth imaging. *IEEE Trans. Antennas Propag.* 69(8), 4909–4922 (2021)
- [S15] Tan, K., Chen, X.: Precise near-range 3-D image reconstruction based on MIMO circular synthetic aperture radar. *IEEE Trans. Microw. Theory Techn.* 69(5), 2651–2661 (2021)
- [S16] Sheen, D.M., McMakin, D.L., Collins, H.D., Hall, T.E.: Near-field millimeter-wave imaging for weapons detection. In: *Appl. Signal Image Process. Explosives Detection Syst.*, vol. 1824, pp. 223–233 (1993). SPIE
- [S17] Keller, P.E., McMakin, D.L., Sheen, D.M., McKinnon, A.D., Summet, J.W.: Privacy algorithm for airport passenger screening portal. In: *Appl. Sci. Computat. Intell. III*, vol. 4055, pp. 476–483 (2000). SPIE

- [S18] Keller, P.E., McMakin, D.L., Sheen, D.M., McKinnon, A.D., Summet, J.W.: Privacy algorithm for cylindrical holographic weapons surveillance system. *IEEE Aerosp. Electron. Syst. Mag.* 15(2), 17–24 (2000)
- [S19] McMakin, D.L., Sheen, D.M., Hall, T.E.: Millimeter-wave imaging for concealed weapon detection. In: *Nondestruct. Detection Meas. Homeland Secur.*, vol. 5048, pp. 52–62 (2003). SPIE
- [S20] Sheen, D.M., McMakin, D.L., Hall, T.E.: Near field imaging at microwave and millimeter wave frequencies. In: *IEEE/MTT-S Int. Microw. Symp.*, pp. 1693–1696 (2007). IEEE
- [S21] McMakin, D.L., Sheen, D.M., Hall, T.E., Kennedy, M.O., Foote, H.P.: Biometric identification using holographic radar imaging techniques. In: *Sensors, and Command, Control, Commun., and Intell. (C3I) Technol. for Homeland Secur. and Homeland Defense VI*, vol. 6538, pp. 80–91 (2007). SPIE
- [S22] McMakin, D.L., Sheen, D.M., Griffin, J.W., Lechelt, W.M.: Extremely high-frequency holographic radar imaging of personnel and mail. In: *Sensors, and Command, Control, Commun., and Intell. (C3I) Technol. for Homeland Secur. and Homeland Defense V*, vol. 6201, pp. 580–591 (2006). SPIE
- [S23] Tao, X., Gao, H., Shen, X., Wang, J., Jia, J.: Scale-recurrent network for deep image deblurring. In: *IEEE/CVF Int. Conf. Pattern Recognit.*, pp. 8174–8182 (2018)
- [S24] Ahmed, S.S., Schiessl, A., Schmidt, L.-P.: A novel fully electronic active real-time imager based on a planar multistatic sparse array. *IEEE Trans. Microw. Theory Techn.* 59(12), 3567–3576 (2011)
- [S25] Moulder, W.F., Krieger, J.D., Majewski, J.J., Coldwell, C.M., Nguyen, H.T., Maurais-Galejs, D.T., Anderson, T.L., Dufilie, P., Herd, J.S.: Development of a high-throughput microwave imaging system for concealed weapons detection. In: *IEEE Int. Symp. Phased Array Syst. Technol.*, pp. 1–6 (2016). IEEE
- [S26] Zhuge, X., Yarovoy, A.G.: A sparse aperture MIMO-SAR-based UWB imaging system for concealed weapon detection. *IEEE Trans. Geosci. Remote Sens.* 49(1), 509–518 (2010)
- [S27] Zhu, R., Zhou, J., Jiang, G., Cheng, B., Fu, Q.: Grating lobe suppression in near range mimo array imaging using zero migration. *IEEE Trans. Microw. Theory Techn.* 68(1), 387–397 (2019)
- [S28] Gumbmann, F., Schmidt, L.-P.: Millimeter-wave imaging with optimized sparse periodic array for short-range applications. *IEEE Trans. Geosci. Remote Sens.* 49(10), 3629–3638 (2011)

- [S29] Li, S., Wang, S., Wu, S., Hoorfar, A., An, Q., Xing, G., Zhao, M., Zhao, G.: Millimeter-wave imaging via circular-arc mimo arrays. *IEEE Trans. Microw. Theory Techn.* 71(7), 3156–3172 (2023)
- [S30] Yang, B., Zhuge, X., Yarovoy, A., Ligthart, L.: UWB MIMO antenna array topology design using PSO for through dress near-field imaging. In: *Eur. Microw. Conf.*, pp. 1620–1623 (2008). IEEE
- [S31] Yang, B., Yarovoy, A., Aubry, P., Zhuge, X.: Experimental verification of 2D UWB mimo antenna array for near-field imaging radar. In: *Eur. Microw. Conf.*, pp. 97–100 (2009). IEEE
- [S32] Zhuge, X., Yarovoy, A.: Near-field ultra-wideband imaging with two-dimensional sparse MIMO array. In: *Eur. Conf. Antennas Propag.*, pp. 1–4 (2010). IEEE
- [S33] Zhuge, X., Yarovoy, A.G.: Study on two-dimensional sparse MIMO UWB arrays for high resolution near-field imaging. *IEEE Trans. Antennas Propag.* 60(9), 4173–4182 (2012)
- [S34] Tan, K., Wu, S., Wang, Y., Ye, S., Chen, J., Fang, G.: A novel two-dimensional sparse MIMO array topology for UWB short-range imaging. *IEEE Antennas Wireless Propag. Lett.* 15, 702–705 (2015)
- [S35] Tan, K., Wu, S., Wang, Y., Ye, S., Chen, J., Liu, X., Fang, G., Yan, S.: On sparse MIMO planar array topology optimization for UWB near-field high-resolution imaging. *IEEE Trans. Antennas Propag.* 65(2), 989–994 (2016)
- [S36] Yanik, M.E., Torlak, M.: Near-field MIMO-SAR millimeter-wave imaging with sparsely sampled aperture data. *IEEE Access* 7, 31801–31819(2019)
- [S37] Cheng, Q., Liu, Y., Zhang, H., Hao, Y.: A generic spiral MIMO array design method for short-range UWB imaging. *IEEE Antennas Wireless Propag. Lett.* 19(5), 851–855 (2020)
- [S38] An, Q., Hoorfar, A., Lv, H., Wang, J.: Task-specific sparse MIMO array design for twri using multi-objective CMA-ES. In: *General Assem. Scientific Symp. Int. Union Radio Sci.*, pp. 1–4 (2021). IEEE
- [S39] Wang, S., Li, S., Hoorfar, A., Miao, K., Zhao, G., Sun, H.: Compressive sensing based sparse MIMO array synthesis for wideband near-field millimeter-wave imaging. *IEEE Trans. Aerosp. Electron. Syst.* (2023)
- [S40] Ahmed, S.S., Schiess, A., Schmidt, L.-P.: Near field mm-wave imaging with multistatic sparse 2D-arrays. In: *Eur. Radar Conf.*, pp. 180–183 (2009). IEEE

- [S41] Gonzalez-Valdes, B., Allan, G., Rodriguez-Vaqueiro, Y., Alvarez, Y., Mantzavinos, S., Nickerson, M., Berkowitz, B., Marti, J., Las-Heras, F., Rappaport, C.M., et al.: Sparse array optimization using simulated annealing and compressed sensing for near-field millimeter wave imaging. *IEEE Trans. Antennas Propag.* 62(4), 1716–1722 (2013)
- [S42] Huang, C., Wang, S., Li, S.: Modified markov random fields-based variational bayesian imaging approach for cluster structured faint scattered targets. In: *Int. Conf. Microw. Millimeter Wave Technol.*, pp. 1–3 (2022).IEEE
- [S43] Lv, M., Chen, H., Ma, J., Chen, L., Yang, J., Ma, X.: 2D high-resolution ISAR imaging by joint using matrix completion and compressed sensing. In: *CIE Int. Conf. Radar*, pp. 107–110 (2021)
- [S44] Liu, J., Xu, S., Gao, X., Li, X.: Compressive radar imaging methods based on fast smoothed l0 algorithm. *Procedia Eng.* 29, 2209–2213 (2012)
- [S45] Wang, X., Li, G., Liu, Y., Amin, M.G.: Two-level block matching pursuit for polarimetric through-wall radar imaging. *IEEE Trans. Geosci. Remote Sens.* 56(3), 1533–1545 (2017)
- [S46] Fang, Y., Wang, B., Sun, C., Wang, S., Hu, J., Song, Z.: Joint sparsity constraint interferometric ISAR imaging for 3-D geometry of near-field targets with sub-apertures. *Sensors* 18(11), 3750 (2018)
- [S47] Barzegar, A.S., Cheldavi, A., Sedighy, S.H., Nayyeri, V.: 3-D through-the-wall radar imaging using compressed sensing. *IEEE Geosci. Remote Sens. Lett.* 19, 1–5 (2021)

<Comment 3>

From my understanding of the work presented, the aim is to restrict the application of this imaging system to a restricted dataset, considering only body scanner applications. The assumed correlation of the scenes to be imaged in the region of interest then enables the amount of information to be reduced by identifying the main structures associated. It then seems necessary to define a propagation operator involving a certain loss of information, implying that it is then possible to relate this to the spatial redundancy of the signals measured by the imaging system within the restricted framework of these body scanner applications.

<Response>

Thank you for your inquiry. To clarify, our proposed scheme is primarily intended for human secu-

rity checks. Thus, we have focused our efforts on body scanner applications, as these applications have the potential to reduce the cost of efficient MMW surveillance by more than one order of magnitude. Also, the techniques can be applied in various other contexts, including Synthetic Aperture Radar (SAR) imaging systems electronically scanned antennas (ESAs), and etc. Regarding the definition of a propagation operator that involves a certain loss of information, we have established a connection between the sparse array topology and the images in the revised supplement:

Location: Supplement Note 4

“Relationship between the sparse array topology and imaging results

Fig. S4: Configuration of the planar sparse array for wideband near-field imaging.

This section establishes a connection between the sparse array topology and the resulting images. We first elucidate the relationship between array size and its influence on imaging results, drawing inspiration from the work of Wang et al. [25]. Consider a wideband monostatic sparse array characterized by arbitrary element positions, as illustrated in Fig. S4. These positions are sampled from a 2D dense (full) array comprising N elements. In the context of an imaging scenario featuring Q scatterers and relying on the Born approximation [S74], the expression for the received scattered field of the dense array can be represented as per Kay’s fundamentals [S75]:

$$s(k, \mathbf{r}') = \sum_{q=1}^Q \sigma_q \frac{e^{-j2k\|\mathbf{r}' - \mathbf{r}_q\|}}{16\pi^2 \|\mathbf{r}' - \mathbf{r}_q\|^2}, \quad (\text{S9})$$

where \mathbf{r}' signifies the element position within the dense array, \mathbf{r}_q and σ_q denote the position and scattering coefficient, respectively, of the q th target. The wavenumber k is expressed as $k = \frac{2\pi f}{c}$.

The matrix representation of Eq. (S9) is articulated as follows:

$$\mathbf{s}_k = \mathbf{A}_k \boldsymbol{\sigma}, \quad (\text{S10})$$

where

$$\mathbf{s}_k = [s(k, \mathbf{r}'_1), s(k, \mathbf{r}'_2), \dots, s(k, \mathbf{r}'_N)]^T, \quad (\text{S11})$$

$$\mathbf{A}_k = \begin{bmatrix} \frac{e^{-j2k\|\mathbf{r}'_1-\mathbf{r}_1\|}}{16\pi^2\|\mathbf{r}'_1-\mathbf{r}_1\|^2} & \cdots & \frac{e^{-j2k\|\mathbf{r}'_1-\mathbf{r}_Q\|}}{16\pi^2\|\mathbf{r}'_1-\mathbf{r}_Q\|^2} \\ \vdots & \ddots & \vdots \\ \frac{e^{-j2k\|\mathbf{r}'_N-\mathbf{r}_1\|}}{16\pi^2\|\mathbf{r}'_N-\mathbf{r}_1\|^2} & \cdots & \frac{e^{-j2k\|\mathbf{r}'_N-\mathbf{r}_Q\|}}{16\pi^2\|\mathbf{r}'_N-\mathbf{r}_Q\|^2} \end{bmatrix}_{N \times N_Q}, \quad (\text{S12})$$

$$\boldsymbol{\sigma} = [\sigma_1, \sigma_2, \dots, \sigma_Q]^T. \quad (\text{S13})$$

The sparse array can be derived through subsampling from \mathbf{s}_k and can be represented as follows:

$$\mathbf{s}'_k = \boldsymbol{\Psi} \mathbf{s}_k, \quad (\text{S14})$$

where

$$\mathbf{s}'_k = [s'(k, \mathbf{r}'_1), s'(k, \mathbf{r}'_2), \dots, s'(k, \mathbf{r}'_N)]^T, \quad (\text{S15})$$

$$\boldsymbol{\Psi} = \begin{bmatrix} w_1 & & & & \\ & w_2 & & & \\ & & \ddots & & \\ & & & \ddots & \\ & & & & w_N \end{bmatrix}_{N \times N}, \quad (\text{S16})$$

The matrix $\boldsymbol{\Psi}$ is predominantly a diagonal matrix with the majority of its diagonal elements set to zero. Assuming that there are a total of N' sampling points distributed across the imaging region, the imaging result corresponding to k due to the array aperture can be expressed as follows:

$$\mathbf{I}_k = \boldsymbol{\Phi} \mathbf{s}'_k, \quad (\text{S17})$$

where Φ represents the imaging operator, which can encompass methods such as Back Projection (BP) [15], RMA, Compressive Sensing with Conjugate Gradient (CS-CG) [36], and the proposed untrained learning technique, among others. The vector \mathbf{I}_k is defined as:

$$\mathbf{I}_k = [I(k, \mathbf{r}''_1), I(k, \mathbf{r}''_2), \dots, I(k, \mathbf{r}''_{N'})]^T, \quad (\text{S18})$$

Here, $\mathbf{r}''_{N'}$ represents the position of the N' th imaging pixel. Given that Ψ is a diagonal matrix, and \mathbf{s}_k is a vector, we can rewrite Eq. (S17) as:

$$\mathbf{I}_k = \Phi \mathbf{S}_k \mathbf{w}, \quad (\text{S19})$$

where

$$\mathbf{I}_k = \begin{bmatrix} s(k, \mathbf{r}'_1) & & & & \\ & s(k, \mathbf{r}'_2) & & & \\ & & \ddots & & \\ & & & \ddots & \\ & & & & s(k, \mathbf{r}'_N) \end{bmatrix}_{N \times N}, \quad (\text{S20})$$

$$\mathbf{w} = [w_1, w_2, \dots, w_N]^T. \quad (\text{S21})$$

The sub-imaging results across all wavenumbers in the bandwidth need to be integrated to generate the final image:

$$\mathbf{I} = \mathbf{B} \mathbf{w}, \quad (\text{S22})$$

where

$$\mathbf{I} = \sum_k \mathbf{I}_k, \quad (\text{S23})$$

$$\mathbf{B} = \sum_k \Phi \mathbf{S}_k. \quad (\text{S24})$$

To summarize, regardless of the specific imaging method employed, it is evident that the design of sparse arrays has a direct impact on the resulting images, as explicitly demonstrated in Eq. (S22).

By integrating RMA with Eq. (S22), and using a reference array (which can be a full array), we can derive the averaged imaging result \mathbf{I}_{ref} , serving as the baseline by consolidating all the echoes. Subsequently, we can determine the propagation operator that quantifies the loss incurred by arbitrary imaging methods:

$$L_{\text{info}} = \chi \cdot \|\mathbf{I}_{\text{ref}} - \mathbf{B}\mathbf{w}\|_2 \quad (\text{S25})$$

where χ signifies the image quality indicator, which can encompass metrics like RMSE, PSNR, SSIM, and others. This allows us to readily obtain the averaged propagation matrix or operator L_{info} , encapsulating the statistical significance for the specific subject under investigation.

In practice, we utilize the imaging result of a particular method \mathbf{I} for quantitative assessment of information loss by comparing it with the referenced result \mathbf{I}_{ref} . To facilitate this assessment, we employ straightforward metrics, specifically the RMSE, PSNR, and SSIM. These criteria provide a robust foundation for establishing a relationship between the quality of reconstruction results and spatial redundancy, a pivotal aspect of our research. Table 1, Supplement Note 6, and Supplement Note 7 demonstrate the superior performance of the reported statistically optimized sparse array.

Our full array comprises 79980 elements, and existing methods [S10, S30, S34, S35, S38, S39, S76–S78] struggle to address the challenges posed by arrays with an enormous number of elements. Hence, we propose a statistically sparse array design strategy, rooted in the principle of ranking the combination of magnitude and inverse phase gradient of echoes. This strategy effectively tailors the sparse array to meet the requirements of security-check scenarios.”

References:

- [25] Wang, S., Li, S., Ren, B., Miao, K., Zhao, G., Sun, H.: Convex optimization-based design of sparse arrays for 3-d near-field imaging. *IEEE Sensors Journal* (2023)
- [S74] Morse, P.M., Feshbach, H.: *Methods of theoretical physics*. *Amer. J. Phys.* 22(6), 410–413 (1954)
- [S75] Kay, S.M.: *Fundamentals of Statistical Signal Processing: Estimation Theory*. Prentice-Hall, Inc., New Jersey (1993)
- [15] Desai, M.D., Jenkins, W.K.: Convolution backprojection image reconstruction for spotlight mode synthetic aperture radar. *IEEE Trans. Image Process* 1(4), 505–517 (1992)

- [36] Li, S., Zhao, G., Sun, H., Amin, M.: Compressive sensing imaging of 3-d object by a holographic algorithm. *IEEE Trans. Antennas Propag.* 66(12), 7295–7304 (2018)
- [S10] Tan, K., Chen, X.: Fast 3-D image reconstruction on nonregular UWB sparse MIMO planar array using scaling techniques. *IEEE Trans. Microw. Theory Techn.* 69(1), 222–234 (2020)
- [S30] Yang, B., Zhuge, X., Yarovoy, A., Ligthart, L.: UWB MIMO antenna array topology design using PSO for through dress near-field imaging. In: *Eur. Microw. Conf.*, pp. 1620–1623 (2008). IEEE
- [S34] Tan, K., Wu, S., Wang, Y., Ye, S., Chen, J., Fang, G.: A novel two-dimensional sparse MIMO array topology for UWB short-range imaging. *IEEE Antennas Wireless Propag. Lett.* 15, 702–705 (2015)
- [S35] Tan, K., Wu, S., Wang, Y., Ye, S., Chen, J., Liu, X., Fang, G., Yan, S.: On sparse MIMO planar array topology optimization for UWB near-field high-resolution imaging. *IEEE Trans. Antennas Propag.* 65(2), 989–994 (2016)
- [S38] An, Q., Hoorfar, A., Lv, H., Wang, J.: Task-specific sparse MIMO array design for twri using multi-objective CMA-ES. In: *General Assem. Scientific Symp. Int. Union Radio Sci.*, pp. 1–4 (2021). IEEE
- [S39] Wang, S., Li, S., Hoorfar, A., Miao, K., Zhao, G., Sun, H.: Compressive sensing based sparse MIMO array synthesis for wideband near-field millimeter-wave imaging. *IEEE Trans. Aerosp. Electron. Syst.* (2023)
- [S76] Valdes, B., Allan, G., YR-Vaqueiro, Y.A., Mantzavinos, S., Nickerson, M., et al.: Sparse array optimization using simulated annealing and 1065 compressed sensing for near-field millimeter wave imaging. *IEEE Trans. Antennas Propag.* 62(4), 1716–1722 (2014)
- [S77] Napier, P.J., Thompson, A.R., Ekers, R.D.: The very large array: Design and performance of a modern synthesis radio telescope. *Proc. IEEE* 71(11), 1295–1320 (1983)
- [S78] Schwartz, J.L., Steinberg, B.D.: Ultrasparse, ultrawideband arrays. *IEEE Trans. Ultrason. Ferroelect. Freq. Control* 45(2), 376–393 (1998)

<Comment 4>

The reconstruction algorithm seems to me the most relevant part of this work, considering the added value of a non-linear function capable of operating despite the undersampling of the antenna

array exploited. There is, however, a substantial new literature in the field of microwave and millimeter wave imaging, exploiting recent advances in machine learning. The latter appears to be inadequately described, despite the fact that some of its processing is identical to that used in the field.

<Response>

Thank you for your insightful comments on our manuscript. We have conducted an extensive review of the recent literature concerning the application of machine learning techniques. In summary, these methods rely on training datasets, which may result in overfitting issues and a lack of generalization and robustness. In contrast, our proposed untrained learning scheme eliminates conventional training procedures, achieving accurate and robust imaging of concealed targets. To evaluate the performance of the Deep Learning (DL) method, we have trained four commonly used UNet reconstruction networks (see Sec. 2.2 in the main text and Supplement Note 7) corresponding to different combinations of sampling strategies and ratios. Both numerical and visual results (Tab. 1, Fig. S8, and Fig. S9) indicate the superior performance of the reported untrained learning method compared to DL and existing compressive sensing methods. Besides, untrained learning has the advantage of being more robust than conventional DL methods (Tab. 1 and Fig. S9). Our approach offers superior reconstruction accuracy and high reliability, particularly for sensitive security inspection tasks.

Location: Main text, Section 2.2, Line 210 - 228.

“To evaluate the performance of the reported reconstruction, the common-used RMA, state-of-the-art CS-based methods including Compressed Sensing-Complex Gradient (CS-CG) [36], Alternating Direction Method of Multipliers (ADMM) [44], and DL-based reconstruction networks are employed in the following comparisons. We trained four complex-valued UNet-like [45] networks, corresponding to random array + 10% sampling ratio, statistically optimized array + 10% sampling ratio, random array + 25% sampling ratio, and statistically optimized array + 25% sampling ratio. Details about these DL networks are in Supplement Note 7. As shown in Fig. 2, the reported untrained reconstruction can retrieve more clear textures of concealed targets while RMA causes more artifacts, and other CS-based methods cannot mitigate the clutter components and maintain the details of the target. Moreover, the comparison in Tab. 1 leads to the conclusion that the reported untrained learning outperforms existing algorithms in accuracy under both random and

statistics-based sampling. It shows an average 2.61dB and 4.19dB improvement in PSNR compared to existing CS-based approaches at 10% and 25% sampling ratios, respectively. Compared to the DL method, untrained learning has constraints of physical priors, which makes the reconstruction quality more robust (with a smaller standard deviation). More details are in Supplement Note 8.

Fig. 2 The diagram of the reported lightweight complex-valued convolutional network (CCN). **a**, We optimize the parameter of CCN following the physical constraint (Eq. (4)) in an untrained manner to reconstruct the target scene. The CCN network is composed of a complex ConvBNReLU layer, five complex Res-blocks, and a complex Conv layer. **b**, The reconstruction results of various algorithms with the statistically sparse sampling.

Table 1 The quantitative comparisons of reconstructed data with different sampling strategies and reconstruction methods. We used all the 200 echoes of the test dataset for reconstruction. The evaluation metrics include the average/standard deviation of RMSE (\downarrow), PSNR (\uparrow), and SSIM (\uparrow). More details are in Supplement Note 8.

Array		Random					Statistically optimized				
SR	Metrics	RMA	CS-CG	ADMM	DL	Untrained	RMA	CS-CG	ADMM	DL	Untrained
10%	RMSE	63.66/9.63	59.10/13.49	43.58/5.65	46.62/8.23	40.05/5.84	49.92/5.97	50.93/9.75	40.82/5.67	32.76/8.11	31.53/5.42
	PSNR	12.13/1.32	12.91/1.72	15.41/1.09	14.88/1.45	16.17/1.23	14.26/1.00	14.28/1.52	16.11/1.00	18.07/2.05	18.40/1.16
	SSIM	0.12/0.05	0.05/0.05	0.13/0.04	0.26/0.13	0.53/0.06	0.32/0.05	0.44/0.14	0.24/0.03	0.60/0.11	0.65/0.05
25%	RMSE	53.33/8.49	44.90/7.39	47.15/7.96	35.65/10.09	28.23/4.20	33.89/4.86	35.94/3.28	40.91/7.22	28.05/8.86	23.90/3.31
	PSNR	13.75/1.36	15.22/1.25	14.90/1.45	17.40/2.28	19.30/1.24	17.65/1.25	17.07/0.77	16.09/1.54	19.55/2.50	20.72/1.15
	SSIM	0.19/0.06	0.16/0.07	0.56/0.06	0.49/0.16	0.59/0.06	0.45/0.07	0.60/0.07	0.61/0.05	0.64/0.15	0.72/0.05

Location: Supplement Note 1.4

“Deep learning-based reconstruction algorithms

With the bloom of Deep Learning (DL) theory, various DL-based reconstruction methods (listed in Tab. S3) have been proposed to yield high-quality images within specific datasets. It is important to note that some metrics may vary greatly in different imaging scenarios and datasets, due to the varying complexity of imaging different scenes. The effectiveness and reliability of these approaches should be interpreted in specific imaging conditions and dataset characteristics.

In this domain, many researchers have employed various neural networks [S48], [30, 31] to enhance the imaging results of pre-reconstruction algorithms. However, these methods do not fully utilize the anisotropic scattering characteristics of targets and the physical process of scattering. Moreover, conventional networks are often considered black boxes. To enhance interpretability and robustness, some works combined model-based CS methods and neural networks for reconstruction [S49, S50, S52].

On the other hand, deep learning techniques have been directly employed for dealing with the Inverse Scattering Problem (ISP) end-to-end, where these methods unfold the nonlinear electromagnetic inverse scattering process using deep neural networks, thereby reducing computational costs [S58, S60, S63], [32, 33, 34, 35]. Besides, CS methods can also be integrated into deep learning networks to achieve ISP [S65, S67, S68, S72]. However, some adopt transfer learning by employing optical data as a reference to enhance the reconstruction results [35], but this may hinder learning the penetrability characteristics of electromagnetic waves, which are vital for concealed

Table S3: Recent advancement of learning-based MMW imaging techniques. The light yellow background represents the enhancement methods, where the input is a pre-reconstructed image and the output is the enhanced image. The pink background represents the end-to-end methods, where the input is an EM wave signal and the output is the reconstructed image.

Ref	Train/Test	Network	2D/3D	Scenario	MSE	PSNR	SSIM	FA	TCR	ENT	IC	Year
[30]	-/-	CVNN	2D	Terrain	377	16.05	-	-	-	-	-	2016
[S48]	50000/-	CV-CNN	2D	Point scatters	-	-	-	-	-	-	-	2018
[31]	600/-	UNet	2D	Airplane scatters	-	-	-	30	67.37	4.76	12.54	2019
[S49]	-/-	ADMM-CNN	2D	Airplane scatters	-	23.87	-	-	-	-	-	2020
[S50]	5184/576	CS-CNN	2D	Point scatters	-	-	-	-	-	-	-	2020
[S51]	420/60	SARNet	2D	Deer	-	22.98	0.84	-	-	-	-	2021
[S52]	10000/-	GAN	2D	Point scatters	-	26.64	0.84	-	-	-	-	2022
[S53]	1520/80	CVPHD	2D	Yak-42 aircraft	-	-	0.7989	-	-	-	-	2022
[S54]	5000/-	AGAN	2D	Point scatters	-	44.77	-	-	-	8.750	-	2022
[S55]	2082/520	UFGAN	2D	Yak-42 aircraft	-	-	0.8163	-	91.20	-	-	2022
[S56]	1000/100	MF-ADMM-Net	2D	Sea surface	-	-	0.91	-	-	6.33	-	2022
[S57]	8920/1000	PMANN	2D	Characters	-	34.79	0.996	-	-	-	-	2023
[32]	7000/2000	CNN	2D	Letter-shaped objects	0.0447	-	0.8685	-	-	-	-	2018
[33]	475/25	UNet-CNN	2D	Circular-cylinders	-	-	-	-	-	-	-	2018
[S58]	475/25	UNet	2D	Cylinders	-	-	-	-	-	-	-	2020
[S59]	-/-	CIST	2D	Yak-42 aircraft	-	-	-	-	22.57	0.2418	-	2020
[S60]	800/200	2D-ADMM-Net	2D	Yak-42 aircraft	-	45.58	0.99	-	-	0.3194	-	2021
[S61]	1000/-	RMIST-Net	3D	Point scatters	-	45.14	-	-	-	-	-	2021
[S62]	-/-	AF-AMPNet	2D	Airplane scatters	0.61	-	0.978	-	-	6.563	63.52	2021
[S63]	2223/-	U-ADMMNet	2D	Satellite model	-	38.22	-	-	-	-	-	2021
[S64]	10000/-	TPSSI-Net	3D	A cone scatter model	-	-	0.84	-	-	-	-	2021
[S65]	36000/4000	MDLI-Net	2D	Point scatters	-	-	0.9998	-	-	-	-	2021
[S66]	1000/-	LFIST-Net	3D	Fighter point cloud model	-	41.71	0.997	-	-	-	-	2021
[S67]	300/70	LRSR-ADMM-Net	2D	Complex surface features	-	23.99	0.8261	-	-	-	-	2022
[34]	4537/1134	DNN	2D	Car model	-	25.34	0.9607	-	-	-	-	2022
[S68]	12000/-	ATResCS	2D	Wheat	-	43.82	0.957	-	-	-	-	2022
[S69]	-/-	SAF-3DNet	3D	Rabbit point cloud model	-	47.78	0.9924	-	-	-	-	2022
[S70]	2000/-	SISR-Net	3D	Vehicle point cloud model	-	35.74	0.704	-	-	-	-	2022
[S71]	1000/-	SFH-ADMM-Net	3D	Fighter point cloud model	-	38.70	0.972	-	-	-	-	2022
[S72]	600/400	2D-IADIANet	2D	Point scatters	-	34.75	0.969	-	-	-	-	2023
[35]	-/-	GAN	2D	Airplane scatters	-	28.60	0.86	-	-	-	-	2023

object detection.

In the realm of 3D imaging based on deep learning, scholars have proposed dedicated learning-based networks [S64, S69, S70] or integrated CS methods into the frameworks [S61, S66]. However, all the aforementioned methods often heavily rely on training datasets, which can result in overfitting issues and a lack of generalization and robustness. Therefore, further research is needed to address these challenges and improve the performance of 3D imaging algorithms in real-world scenarios.

In contrast, our proposed untrained learning scheme eliminates the need for conventional training procedures and avoids the pattern transfer process, preserving the inherent penetrability properties of MMW. Furthermore, our approach overcomes the black-box limitations associated with end-to-end neural networks and offers superior reconstruction accuracy and high reliability, particularly for sensitive security inspection tasks.

In summary, we present a large-scale, single-shot MMW imaging framework leveraging a sparse antenna array. This framework facilitates cost-effective yet high-fidelity security inspections while incorporating an interpretable learning scheme. Furthermore, the scheme delivers robust and precise image reconstruction from sparsely sampled echoes. Finally, our research culminates in the development of a neural network for automatic object detection. Empirical results substantiate the successful detection of concealed centimeter-sized targets using a sparse array with a sampling ratio as low as 10%, a feat unattainable by contemporary approaches.”

Location: Supplement Note 8

“Reconstruction results of the test dataset

We tested the performance of the proposed method on the test set under various sampling ratios. As mentioned in the main text, the test set consists of 200 randomly selected MMW echo signals. We compared the proposed untrained learning method with RMA, CS-CG, ADMM, and deep learning techniques. The visual and numerical comparisons are shown in Fig. S8 and Fig. S9, respectively. We can conclude that:

- The statistically sparse array outperforms the randomly sparse array.
- The deep learning method, while performing well in certain scenarios, lacks robustness and may perform worse in certain cases.

- The untrained learning method outperforms the other comparative methods in cases of average reconstruction accuracy and robustness.”

Fig. S8: Visual comparisons of various algorithms. We tested the reconstructions from both random and statistically optimized arrays at 10% and 25% sampling ratios.

Fig. S9: The numerical comparisons between different reconstruction techniques including RMA, CS-CG, ADMM, Deep Learning (DL), and the proposed untrained learning. The test dataset contains 200 different echoes. The comparisons involve the evaluated RMSE(\downarrow), PSNR(\uparrow), and SSIM(\uparrow) values of 10% randomly sampled (a), 10% statistically sparse sampled (b), 25% randomly sampled (c), and 25% statistically sparse sampled (d) echoes.

References:

[30] Ichikawa, K., Hirose, A.: Singular unit restoration in InSAR using complex-valued neural

- networks in the spectral domain. *IEEE Trans. Geosci. Remote Sens.* 55(3), 1717–1723 (2016)
- [31] Hu, C., Wang, L., Li, Z., Zhu, D.: Inverse synthetic aperture radar imaging using a fully convolutional neural network. *IEEE Geosci. Remote Sens. Lett.* 17(7), 1203–1207 (2019)
- [32] Li, L., Wang, L.G., Teixeira, F.L., Liu, C., Nehorai, A., Cui, T.J.: DeepNIS: Deep neural network for nonlinear electromagnetic inverse scattering. *IEEE Trans. Antennas Propag.* 67(3), 1819–1825 (2018)
- [33] Wei, Z., Chen, X.: Deep-learning schemes for full-wave nonlinear inverse scattering problems. *IEEE Trans. Geosci. Remote Sens.* 57(4), 1849–1860(2018)
- [34] Rostami, P., Zamani, H., Fakharzadeh, M., Amini, A., Marvasti, F.: A deep learning approach for reconstruction in millimeter-wave imaging systems. *IEEE Trans. Antennas Propag.* 71(1), 1180–1184 (2022)
- [35] Bao, J., Li, D., Li, S., Zhao, G., Sun, H., Zhang, Y.: Fine-grained image generation network with radar range profiles using cross-modal visual supervision. *IEEE Trans. Microw. Theory Techn.* (2023)
- [36] Li, S., Zhao, G., Sun, H., Amin, M.: Compressive sensing imaging of 3-d object by a holographic algorithm. *IEEE Trans. Antennas Propag.* 66(12), 7295–7304 (2018)
- [44] Boyd, S., Parikh, N., Chu, E., Peleato, B., Eckstein, J., et al.: Distributed optimization and statistical learning via the alternating direction method of multipliers. *Found. Trends Mach. Learn.* 3(1), 1–122 (2011)
- [45] Chang, X., Zhao, R., Jiang, S., Shen, C., Zheng, G., Yang, C., Bian, L.: Complex-domain-enhancing neural network for large-scale coherent imaging. *Adv. Photon. Nexus* 2(4), 046006–046006 (2023)
- [S48] Gao, J., Deng, B., Qin, Y., Wang, H., Li, X.: Enhanced radar imaging using a complex-valued convolutional neural network. *IEEE Geosci. Remote Sens. Lett.* 16(1), 35–39 (2018)
- [S49] Li, R., Zhang, S., Zhang, C., Liu, Y., Li, X.: Deep learning approach for sparse aperture isar imaging and autofocusing based on complex-valued admm-net. *IEEE Sensors J.* 21(3), 3437–3451 (2020)
- [S50] Cheng, Q., Ihalage, A.A., Liu, Y., Hao, Y.: Compressive sensing radar imaging with convolutional neural networks. *IEEE Access* 8, 212917–212926 (2020)
- [S51] Su, W.-t., Hung, Y.-C., Chao, T.-H., Yu, P.-J., Yang, S.-H., Lin, C.-W.: Seeing through a

black box: Toward high-quality terahertz tomographic imaging via multi-scale spatio-spectral image fusion. arXiv preprint arXiv:2103.16932 (2021)

[S52] Wang, H., Li, K., Lu, X., Zhang, Q., Luo, Y., Kang, L.: Isar resolution enhancement method exploiting generative adversarial network. *Remote Sens.* 14(5), 1291 (2022)

[S53] Yuan, H., Li, H., Zhang, Y., Wang, Y., Liu, Z., Wei, C., Yao, C.: High-resolution refocusing for defocused ISAR images by complex-valued Pix2pixHD network. *IEEE Geosci. Remote Sens. Lett.* 19, 1–5 (2022)

[S54] Yuan, Y., Luo, Y., Ni, J., Zhang, Q.: Inverse synthetic aperture radar imaging using an attention generative adversarial network. *Remote Sens.* 14(15), 3509 (2022)

[S55] Li, W., Yuan, Y., Zhang, Y., Luo, Y.: Unblurring ISAR imaging for maneuvering target based on UFGAN. *Remote Sens.* 14(20), 5270 (2022)

[S56] Li, M., Wu, J., Huo, W., Jiang, R., Li, Z., Yang, J., Li, H.: Target-oriented SAR imaging for SCR improvement via deep MF-ADMM-Net. *IEEE Trans. Geosci. Remote Sens.* 60, 1–14 (2022)

[S57] Su, Z., Zhang, Y., Zhou, J., Shi, J., Qi, F.: Data-driven based terahertz image restoration. *IEEE Sensors J.* (2023)

[S58] Chen, X., Wei, Z., Maokun, L., Rocca, P., et al.: A review of deep learning approaches for inverse scattering problems (invited review). *Electromagn. Waves* 167, 67–81 (2020)

[S59] Wei, S., Liang, J., Wang, M., Zeng, X., Shi, J., Zhang, X.: CIST: An improved ISAR imaging method using convolution neural network. *Remote Sens.* 12(16), 2641 (2020)

[S60] Li, X., Bai, X., Zhou, F.: High-resolution ISAR imaging and autofocusing via 2D-ADMM-Net. *Remote Sens.* 13(12), 2326 (2021)

[S61] Wang, M., Wei, S., Liang, J., Zeng, X., Wang, C., Shi, J., Zhang, X.: Rmist-net: Joint range migration and sparse reconstruction network for 3-D mmW imaging. *IEEE Trans. Geosci. Remote Sens.* 60, 1–17 (2021)

[S62] Wei, S., Liang, J., Wang, M., Shi, J., Zhang, X., Ran, J.: Af-ampnet: A deep learning approach for sparse aperture ISAR imaging and autofocusing. *IEEE Trans. Geosci. Remote Sens.* 60, 1–14 (2021)

[S63] Xiao, C., Gao, X., Zhang, C.: U-ADMMNet: A model-based deep learning method for sparse aperture isar imaging. In: *Int. Congr. Image Signal Process. BioMed. Eng. Inform.*, pp. 1–7 (2021). IEEE

- [S64] Wang, M., Wei, S., Liang, J., Zhou, Z., Qu, Q., Shi, J., Zhang, X.: Tpsinet: Fast and enhanced two-path iterative network for 3D SAR sparse imaging. *IEEE Trans. Image Process.* 30, 7317–7332 (2021)
- [S65] Hu, X., Xu, F., Guo, Y., Feng, W., Jin, Y.-Q.: Mdli-net: Model-driven learning imaging network for high-resolution microwave imaging with large rotating angle and sparse sampling. *IEEE Trans. Geosci. Remote Sens.* 60, 1–17 (2021)
- [S66] Wang, M., Wei, S., Liang, J., Liu, S., Shi, J., Zhang, X.: Lightweight FISTA-inspired sparse reconstruction network for mmw 3-D holography. *IEEE Trans. Geosci. Remote Sens.* 60, 1–20 (2021)
- [S67] An, H., Jiang, R., Wu, J., Teh, K.C., Sun, Z., Li, Z., Yang, J.: LRSR-ADMM-Net: A joint low-rank and sparse recovery network for SAR imaging. *IEEE Trans. Geosci. Remote Sens.* 60, 1–14 (2022)
- [S68] Jiang, Y., Li, G., Ge, H., Wang, F., Li, L., Chen, X., Lv, M., Zhang, Y.: Adaptive compressed sensing algorithm for terahertz spectral image reconstruction based on residual learning. *Spectrochimica Acta Part A: Mol. Biomolecular Spectrosc.* 281, 121586 (2022)
- [S69] Zhou, Z., Wei, S., Zhang, H., Shen, R., Wang, M., Shi, J., Zhang, X.: SAF-3DNet: Unsupervised AMP-inspired network for 3-D MMW SAR imaging and autofocusing. *IEEE Trans. Geosci. Remote Sens.* 60, 1–15(2022)
- [S70] Wei, S., Zhou, Z., Wang, M., Zhang, H., Shi, J., Zhang, X., Fan, L.: Learning-based split unfolding framework for 3-D mmW radar sparse imaging. *IEEE Trans. Geosci. Remote Sens.* 60, 1–17 (2022)
- [S71] Wang, M., Wei, S., Zhou, Z., Shi, J., Zhang, X.: Efficient ADMM framework based on functional measurement model for mmW 3-D SAR imaging. *IEEE Trans. Geosci. Remote Sens.* 60, 1–17 (2022)
- [S72] Lv, M., Chen, W., Yang, J., Wang, D., Wu, X., Ma, X.: Joint 2D sparse ISAR imaging and autofocusing by using 2D-IADIANet. *IEEE Sensors J.* (2023)

<Comment 5>

Finally, the authors highlight the exploitation of the automated detection and classification of various illicit or dangerous objects through the adaptation of a machine learning technique. Contrary

to the authors' claims, the classification of a variety of objects in a security screening context has already been the subject of publications in recent years, including architectures that have been identified and simplified using compressive sensing and computational imaging techniques.

<Response>

Thank you for your careful review and suggestions. As mentioned by the reviewer, in recent years, Wang et al. [47], Yuan et al. [48], and Su et al. [49] have proposed several multi-class MMW human security object detection methods. We acknowledge that the novelty of our work does not reside in the multi-class classification. Instead, it encompasses the statistically sparse array design and untrained learning-based reconstruction. Following the reviewer's suggestion, we list the revisions in the manuscript and supplementary material as follows:

Location: Main text, Section 1, Line 97 - 142

“Considering these challenges, we present a low-cost, large-scale single-shot MMW security inspection framework capable of successfully detecting concealed centimeter-sized targets. This framework encompasses the entire inspection workflow, incorporating statistically sparse array design, untrained learning reconstruction, and detection processes.

Instead of regularly arranged or optimization-based approaches, we introduce a statistically sparse array design method that fully exploits the echoes of existing full-sampled arrays and easily scales to any size free from heavy computation. Specifically, we first collected a set of real-captured echoes using a scanning-based full-sampled antenna array, and then analyzed the statistical importance ranking map of array elements. We theoretically proved that a larger value in the map indicates a more significant element in the antenna array. Based on the ranking map, we experimentally derived a statistically optimized sampling strategy to sparsely select antenna elements at various sampling ratios. Leveraging this approach, we developed a large-scale sparse array (430×186 as the full array) designed for MMW near-field imaging. At low sampling ratios (e.g., 10% and 25%), the technique shows an average 20% RMSE improvement, 2dB higher for PSNR, and 0.22 SSIM improvement compared to random sampling. This statistical approach allows for a more efficient selection of the most significant elements, thereby reducing the number of inefficient elements in antenna array design.

Moreover, to attain high-quality images using the statistically optimized array, we propose an interpretable untrained learning approach. This approach ensures robust MMW reconstruction

from sparsely sampled echoes. We optimize a lightweight complex-valued network based on the objective derived from the physical model of MMW scattering. Unlike conventional DL imaging methods [30-35], this physics-informed learning strategy operates without the need for training, ensuring case-specific optimization for robustness. Compared to the existing CS-based approaches, it yields high-fidelity reconstruction with up to 3.4 dB PSNR improvement at 10% and 25% sampling ratios. Additionally, the technique shows prominent priority over existing techniques for detecting concealed targets from sparsely sampled echoes. The combination of statistics-based sparse sampling and untrained learning techniques results in a significant reduction in the cost of the antenna array by up to an order of magnitude, without compromising the system's ability to detect centimeter-sized concealed targets.

Furthermore, we employ a neural network for the automatic detection of concealed targets. The reported framework combining statistical sparse array and untrained reconstruction shows closer detection accuracy to the full-sampled detection results than existing approaches, which indicates superior reconstruction accuracy. Experiments demonstrated that the reported imaging framework achieved successful concealed target detection at a 10% sampling ratio, while other contemporary approaches failed. This work leverages the statistical prior and emerging fusion frameworks of neural networks and optimization, providing new insights into the development of low-cost and efficient MMW security inspection systems.”

Location: Main text, Section 2.3, Line 247 - 251

“Similar to several multi-class detection works [47-49], the detection network is employed to distinguish five kinds of common hidden targets, such as knives, wrenches, phones, guns, and Explosive Powdered Material (EPM, using silica gel instead in experiments) ”

References:

- [30] Ichikawa, K., Hirose, A.: Singular unit restoration in InSAR using complex-valued neural networks in the spectral domain. *IEEE Trans. Geosci. Remote Sens.* 55(3), 1717–1723 (2016)
- [31] Hu, C., Wang, L., Li, Z., Zhu, D.: Inverse synthetic aperture radar imaging using a fully convolutional neural network. *IEEE Geosci. Remote Sens. Lett.* 17(7), 1203–1207 (2019)
- [32] Li, L., Wang, L.G., Teixeira, F.L., Liu, C., Nehorai, A., Cui, T.J.: Deep-NIS: Deep neural network for nonlinear electromagnetic inverse scattering. *IEEE Trans. Antennas Propag.* 67(3),

1819–1825 (2018)

[33] Wei, Z., Chen, X.: Deep-learning schemes for full-wave nonlinear inverse scattering problems. *IEEE Trans. Geosci. Remote Sens.* 57(4), 1849–1860 (2018)

[34] Rostami, P., Zamani, H., Fakharzadeh, M., Amini, A., Marvasti, F.: A deep learning approach for reconstruction in millimeter-wave imaging systems. *IEEE Trans. Antennas Propag.* 71(1), 1180–1184 (2022)

[35] Bao, J., Li, D., Li, S., Zhao, G., Sun, H., Zhang, Y.: Fine-grained image generation network with radar range profiles using cross-modal visual supervision. *IEEE Trans. Microw. Theory Techn.* (2023)

[47] Wang, C., Shi, J., Zhou, Z., Li, L., Zhou, Y., Yang, X.: Concealed object detection for millimeter-wave images with normalized accumulation map. *IEEE Sensors J.* 21(5), 6468–6475 (2020)

[48] Yuan, M., Zhang, Q., Li, Y., Yan, Y., Zhu, Y.: A suspicious multi-object detection and recognition method for millimeter wave SAR security inspection images based on multi-path extraction network. *Remote Sens.* 13(24), 4978 (2021)

[49] Su, B., Yuan, M.: Object recognition for millimeter wave MIMO-SAR images based on high-resolution feature recursive alignment fusion network. *IEEE Sensors J.* (2023)

<Comment 6>

This last point seems to me to be particularly critical for the publication of this work. The data presented in this paper, and in the supplementary materials, in no way ensure their reproducibility.

<Response>

We have made the project publicly available (<https://github.com/bianlab/MMW>), including the sparse patterns, the reconstruction algorithm, the detection code, and exemplar data. We have verified the reproducibility of the aforementioned data and code under certain hardware conditions. To further improve the solidity of the work, we have conducted extensive additional experiments for a more comprehensive description of the reported sparse array design, untrained learning reconstruction, the applicability and generalization of our system, etc. We have included the following revisions in the main text and supplement:

- Section 2.2 and Table 1: Add the reconstruction results of deep learning methods.

- Section 2.3 and Figure 3: Revise the detection results of the networks trained with full-sampled reconstructions. Add the detection results of the networks trained with sparsely sampled reconstructions.
- Section 2.4 and Figure 4: Add the applicability and generalization of the reported system.
- Figure 5: Add the comparison of spatial resolutions of random and statistically optimized arrays.
- Supplement Note 1: A review of MMW sparse array synthesis and imaging.
- Supplement Note 2: The workflow of generating a statistically sparse array design
- Supplement Note 3: Physical traits of the statistical importance map
- Supplement Note 4: Relationship between the sparse array topology and imaging results
- Supplement Note 5: Comparison with mechanical scanning arrays
- Supplement Note 6: Comparison with commonly used sparse arrays
- Supplement Note 7: Details about the learning-based reconstruction network
- Supplement Note 8: Reconstruction results of the test dataset
- Supplement Note 9: The workflow of the reported MMW security check framework
- Supplement Note 10: Additional details of applicability and generalization
- Supplement Note 11: Additional details of the untrained learning reconstruction
- Supplement Note 12: The influence of interpolation in the scattering process
- Supplement Note 13: Complexity of CCN
- Supplement Note 14: Additional details about the detection network
- Supplement Note 15: Discussion about transmit and receive antenna isolation and the calibration procedure

<Comment 7>

The proposed numerical approach, based on data multiplication derived from amplitude and phase information could benefit from a more complete illustration in supplementary materials. It could be interesting to show the raw data extracted at center frequency in amplitude and phase to justify the relevance of this method. An interesting link with existing support detection techniques in compressive sensing could also be useful.

<Response>

Thank you for your valuable suggestions. We have provided a more complete illustration in supplementary materials about the sparse array design and the physical traits of statistically sparse sampling. With the system bandwidth less than 20%, the scattering characteristics of targets do not vary significantly across different frequencies, as demonstrated in Fig. S2. Thus, we extracted the center-frequency cross-section of each 3D echo to analyze the statistical characteristics of these echoes. We have included a comprehensive illustration of the statistically sparse array design (Supplement Note 2) with its physical traits (Supplement Note 3) in the revised supplement. We have also explored linkages with compressive sensing detection techniques later in this response.

Location: Supplement Note 2

“The workflow of generating a statistically sparse array design

We summarize the overall workflow to obtain a statistically sparse array design M in Fig. S1 and as follows.

- **Collecting the echo dataset:** This workflow starts by collecting the 3D echo dataset. We have collected a large-scale 3D echo dataset which contains 1934 human security inspection echoes. Then we extract the amplitude and phase from the center-frequency cross-sections of 3D echoes.
- **Obtaining the statistical importance map:** The statistical importance map \bar{M} is the production of the averaged amplitude (\bar{A}) and inverse phase gradient (\bar{P}) of 2D cross-sections. This approach was adopted as an intuitive means of statistically integrating the significance of both amplitude and phase components. The averaged representations of these three distributions, (\bar{M} , \bar{A} , and \bar{P}) as depicted in Fig. S1, offer a more comprehensive portrayal of the outline and structural characteristics of all the subjects being tested. A higher value in

\bar{M} denotes higher statistical importance of the element in the array, which is theoretically proofed in Sec. 4.2 and Supplement Note 3.

- **Generating the sparse array design:** Next, we perform Monte Carlo sampling to generate the sparse array design M . All element positions are assigned an importance order, where this importance order is the sorting of the element positions on \bar{M} from the highest value to the lowest value. Generate a uniformly random function $r(n)$ (seed = 100), where the horizontal axis is the importance order ($n = 1 \rightarrow N$, with 1 being most important and N being least important), and the vertical axis takes values from 0 to 1. Given a threshold S , search from 1 to N for n where $r(n) > S$, until the number meets the preset sampling ratio. These corresponding element positions of $r(n) > S$ are the positions retained in the sparse array design M . Different hyperparameter S correspond to different M , so we evaluated the performance of different M under 10% and 25% sampling ratios, and obtained the optimized sparse array designs as shown in Fig. 1 **d** in the main text.”

① Collecting the echo dataset

② Obtaining statistical importance map

③ Generating the sparse array design

Fig. S1: The workflow for generating a statistically sparse array design. We start by collecting the 3D echo dataset and then extracting the center-frequency cross-sections of these echoes. The statistical importance map \bar{M} is the production of averaged amplitude map \bar{A} and inverse phase gradient map \bar{P} . The sparse array design is derived from \bar{M} by Monte Carlo sampling.

Location: Supplement Note 3

“The bandwidth of the system is less than 20%. The scattering characteristics of targets do not vary significantly across different frequencies, as shown in Fig. S2. Therefore, we opt for the center-frequency cross-sections to reflect the overall echoes.”

Fig. S2: The exemplar 3D echo amplitudes and phases, and the corresponding center-frequency amplitude and inverse phase gradient cross-sections.

Exploring Linkages with Existing Support Detection Techniques in Compressive Sensing (CS): We appreciate your recommendation to establish a connection with existing support detection techniques in CS. In the realm of linking our research to existing support detection techniques in CS, the available references are relatively limited, which are utilized in optical remote sensing (ORS) scenes and may not be easily transferred to radar imaging.

Firstly, Xiao et al. [X1] have introduced a pipeline based on CS measurements designed for ship detection tasks. They have devised an algorithm called CS-CenterNet, which harnesses convolutional neural networks (CNNs) and jointly optimizes the scene compression sampling phase and the ship detection phase within the measurements. Besides, Liao et al. [X2] have introduced a low-cost image CS approach referred to as MRCS for object detection. It is important to note that while these approach prove highly effective, they pertain to CS measurements obtained from ORS scenes rather than radar imaging. Additionally, its avoidance of scene reconstruction may limit its applicability in certain specialized scenarios.

It is important to emphasize that the primary focus of our manuscript does not revolve around detection but rather aims to corroborate the efficiency of the statistically optimized sparse array and the untrained interpretable learning scheme. Therefore, for our security inspection purposes, we have opted to directly employ YOLOv8 [29], a state-of-the-art object detection network, for post-processing. The detection results conclusively affirm the potential of this architecture in substantially reducing the cost associated with antenna arrays for single-shot MMW security inspections by an order of magnitude.

In addition, we have included the description about CS reconstruction techniques in the revised Supplement Note 1 (CS-based reconstruction algorithms for sparse arrays).

References:

[X1] Xiao, S., Zhang, Y., Chang, X.: Ship detection based on compressive sensing measurements of optical remote sensing scenes. *IEEE Journal of Selected Topics in Applied Earth Observations and Remote Sensing* 15, 8632–8649 (2022)

[X2] Liao, L., Li, K., Yang, C., Liu, J.: Low-cost image compressive sensing with multiple measurement rates for object detection. *Sensors*, 19(9), 2079 (2019)

[46] Jocher, G., Chaurasia, A., Qiu, J.: YOLO by Ultralytics. <https://github.com/ultralytics>

<Comment 8>

The array sub-sampling strategy could be further developed. The link with physical considerations and rank analyses does not seem so obvious as it stands. The proposed description gives the impression that the authors have simply made a random selection of elements by reducing the number of antennas fed until they reach a threshold considered no longer acceptable under the experimental conditions studied.

<Response>

Thank you for your question. Our statistically sparse array design method is not a purely random selection, but rather involves selection based on the importance ranking of each element in the antenna array. The higher-importance elements are sampled first in descending order. We have provided a more comprehensive demonstration of the workflow of the statistically sparse array design (please see Supplement Note 2 in **Comment 7**) and the underlying physical principles behind it. The relevant revision is provided below.

Location: Supplement Note 3

“Physical traits of the statistical importance map

Here, we provide a theoretical analysis to elucidate the principle of statistical importance ranking of MMW echoes. The bandwidth of the system is less than 20%. The scattering characteristics of targets do not vary significantly across different frequencies, as shown in Fig. S2. Therefore, we opt for the enter-frequency cross-sections to reflect the overall echoes. The following analysis involves the physical traits behind the reported sparse array design method: higher values in amplitude \bar{A} , inverse phase gradient \bar{P} , and importance map \bar{M} correspond to heightened significance.

Amplitude \bar{A}

The amplitude of the echo distribution plays a pivotal role as an indicator of significance within the array. The MMW imaging system’s echo can be mathematically expressed as:

$$s(x', z', k) = \int \int \int \sigma(x, y, z) e^{-jk\sqrt{(R_0-y_0)^2+(x'-x)^2+(z'-z)^2}} dx dy dz. \quad (\text{S1})$$

Here, $\sigma(x, y, z)$ represents the scattering coefficient, and k denotes the wavenumber. The variables x' and z' denote the antenna location along the azimuth and height, respectively, and R_0 represents the distance from the target to the array.

It is evident that a larger $\sigma(x, y, z)$ results in a larger $\|s(x', z', k)\|$. In simpler terms, a heightened amplitude value received by an array element indicates that the corresponding position can capture more information compared to positions with lower values, and vice versa. Consequently, a higher value in \bar{A} denotes a higher importance ranking for the corresponding array element.

Inverse phase gradient \bar{P}

To simplify the problem, we first discuss the one-dimensional case where the target is distributed along the x axis. Assuming that the Born approximation [S74] is satisfied during the scattering process with omitted propagation attenuation, the echo can be obtained as:

$$s(x') = \int \sigma(x) e^{-jk_0 \sqrt{R_0^2 + (x' - x)^2}} dx, \quad (\text{S2})$$

where k_0 denotes the wavenumber of the working single-frequency signal. Considering narrow antenna beam pattern, we have $\|x' - x\| \ll R_0$, then keep the first two terms of the Taylor expansions of the expressions under root, Eq. (S2) can be rewritten as: The gradient along the x -axis can be expressed as:

$$\frac{ds(x')}{dx'} = C \int \sigma(x) e^{-jk_0 \frac{(x' - x)^2}{2R_0}} (2x' - 2x) dx, \quad (\text{S4})$$

where

$$C = j \frac{k_0}{2R_0} e^{-jk_0 R_0}. \quad (\text{S5})$$

Let us focus on the gradient around the element where only the target within its beamwidth can be illuminated,

$$\frac{ds(x')}{dx'} = -2C \int_{x' - x_{\max}}^{x' + x_{\max}} \sigma(x) e^{-jk_0 \frac{(x' - x)^2}{2R_0}} (x - x') dx, \quad (\text{S6})$$

where

$$x_{\max} = R_0 \tan \frac{\Theta_h}{2}. \quad (\text{S7})$$

Herein, Θ_h represents the antenna beamwidth. By employing the technique of integration by parts, we can derive the following expression:

$$\begin{aligned} \frac{ds(x')}{dx'} = \frac{2R_0 C}{k_0} & \left\{ e^{-jk_0 \frac{x_{\max}^2}{2R_0}} [\sigma(x' - x_{\max}) - \sigma(x' + x_{\max})] + \right. \\ & \left. \int_{x' - x_{\max}}^{x' + x_{\max}} \sigma'(x) e^{-jk_0 \frac{(x' - x)^2}{2R_0}} dx \right\}, \end{aligned} \quad (\text{S8})$$

The first row in Eq. (S8) elucidates the influence of the target amplitude near the boundary of the antenna beamwidth. The second row in the same equation delineates the impact of target fluctuations. When the imaging target exhibits a relatively planar structure within the antenna beamwidth, as is often the case in human imaging scenarios, both of these terms tend to approach zero, specifically $\frac{ds(x')}{dx'} = 0$. Conversely, in situations characterized by fluctuating imaging targets or a void in the imaging area where noise dominates the echo, both terms in Eq. (S8) deviate from zero. Under certain conditions, the accumulation of terms in the second row of Eq. (S8) can result in a significant $\frac{ds(x')}{dx'}$.

In summation, when the target's spatial distribution is relatively uniform, the gradient between adjacent elements tends to be small. Conversely, for targets characterized by fluctuating scattering coefficients or those resembling noise-like patterns, the gradient becomes notably larger. The analysis presented above leads to the conclusion that the construction of a sparse array with small element gradients can enhance the quality of the illumination for the specific target of interest. That is, a larger value in \bar{P} indicates that the corresponding element has a higher importance ranking.

Importance map $\bar{M} = \bar{A} \times \bar{P}$

The statistical importance map $\bar{M} = \bar{A} \times \bar{P}$ reflects both the amplitude and phase distribution of the echo dataset. Similarly, a larger value in \bar{M} indicates a higher importance of the corresponding element in the array. Following the workflow in Fig. (S1), the sparse array design is generated. In the third step of this workflow, we can vary the hyperparameter S to find a suitable coverage range and sparsity of the sampling pattern. By numerical comparisons of randomly selected echoes (Fig. (S3), the optimized statistical sparse arrays at 10% and 25% sampling ratios correspond to S equals 0.8 and 0.5, respectively.”

Fig. S3: Numerical comparisons of reconstruction accuracy w.r.t. different S when the sampling ratio equals 10% (**a**) and 25% (**b**). The error bar represents the standard deviation. The optimal sparse patterns of SR=10%&25% correspond to $S = 0.8$ & 0.5 , respectively.

Besides, We have illustrated that the design of a sparse array directly influences the resulting images (please see Supplement Note 4 in **Comment 3**). Recognizing the constraints of contemporary state-of-the-art sparse array synthesis methods, particularly when dealing with large arrays of approximately 80,000 elements, we have introduced an innovative statistical sampling approach for the design of sparse MMW arrays. This approach leverages a data-driven methodology to optimize

the sparse array statistically. Remarkably, our statistically optimized sparse array has demonstrated its capacity to achieve high-fidelity reconstruction quality, even with a significantly reduced number of elements (7998).

References:

[S74] Gubernatis, J., Domany, E., Krumhansl, J., Huberman, M.:The born approximation in the theory of the scattering of elastic waves by flaws. J. Appl. Phys. 48(7), 2812–2819 (1977)

<Comment 9>

The reconstruction technique studied is positioned in relation to the Range Migration Algorithm, applied to spatially truncated data. It seems necessary for the authors to propose a more detailed description of the steps considered for the adaptation of the RMA. Indeed, I have the impression that the results obtained will be particularly sensitive to the interpolation techniques used, as well as to the methods for merging samples in reciprocal space, given the multi-static nature of the imaging system under consideration.

<Response>

Thank you for your insightful question. We have added a comparison between the untrained learning scheme with and without the interpolations. Besides, we have added comprehensive experiments with various conditions to verify the robustness of the statistically array optimization method, which merges echoes in reciprocal space (Supplement Note 2).

1. For untrained learning reconstruction, the results obtained using either interpolations or not are essentially consistent. Thus, we can conclude that our untrained learning technique is robust to the interference caused by interpolations.

Location: Supplement Note 12

“The influence of interpolation in the scattering process

The objective of untrained learning involves the scattering process, which is denoted as

$$\mathcal{H} [\cdot] = \mathcal{F}_{2D}^{-1} \left\{ \text{IN}_k \left\{ \mathcal{F}_{3D} [\cdot] \right\} e^{-jk_y R_0} \right\},$$

where $\mathcal{F}_{3D} \{ \cdot \}$ represents a 3D spatial Fourier transform for all the spatial dimensions of the imaging region. $\mathcal{F}_{2D}^{-1} \{ \cdot \}$ denotes the 2D spatial inverse Fourier transform over the 2D array aperture.

IN_k indicates the interpolation with respect to the wavenumber k . To estimate the influence of interpolation, we adopt the interpolation-free scattering process which is inspired from [S82]:

$$\mathcal{H}[\cdot] = \mathcal{F}_{2D}^{-1} \left\{ \int \mathcal{F}_{2D}[\cdot] e^{-jk_y(R_0+y)} dy \right\}.$$

The reconstruction results of the untrained learning with and without interpolation can be found in Fig. S23. Notably, the 2D and 3D images generated with or without interpolation exhibit a remarkable similarity. This similarity arises because in near-field imaging scenarios, the wavenumber variation is gradual, and interpolation is applied to slowly changing signals. In this way, simple linear interpolation can promise high-quality images. Both the imaging results and analysis have proven the robustness of the interpolation in the scattering process of the untrained imaging method.”

Fig. S23: Reconstructions of the untrained learning with or without interpolation in the objective.

References:

[S82] Zhou, J., Zhu, R., Jiang, G., Zhao, L., Cheng, B.: A precise wavenumber domain algorithm for near range microwave imaging by cross MIMO array. IEEE Trans. Microw. Theory Techn.

67(4), 1316–1326 (2019)

2. Robustness of the statistically array optimization method: The topology of the optimized sparse array may be inherently influenced by the distribution of samples. To test the robustness of the array optimization method, we have conducted numerous experiments with various conditions, such as clothing material and thickness, subject position, body shape, target shape, position and status, etc. Results indicate that the generated sparse array aligns with the specific requirements of security-check scenarios, further verify the robustness of the proposed sparse array optimization method.

Location: Supplement Note 10

“Additional details of applicability and generalization

We conducted a series of experiments to reveal the applicability and generalization of the system. The experiments involved spatial resolution, clothing, body shape, subject position, target position and status, and element assembly error. The reference images are reconstructed from full-sampled echoes by RMA, while the sparsely sampled images are reconstructed by the untrained method.

Clothing material and thickness

As shown in Fig. S11, we can detect hidden objects under cotton, synthetic fiber, and blended fabric clothes. While other materials exhibit good reconstruction quality and facilitate concealed object identification, leather impedes MMW, making it hard to identify hidden targets. Wool, on the other hand, is somewhat MMW-penetrable. However, it interferes with the detection of MMW-absorbing materials such as EPM. So we recommend operators request the subject take off wool, leather, and fur clothing when performing MMW security checking. Further, we tested the reconstruction and detection results in cases of various clothing thicknesses, as shown in Fig. S12. For MMW-penetrated clothing materials, it does not affect the reconstructed image quality and detection accuracy whether the subject is wearing a single jacket or the common layering of multiple thick garments (Down jacket + sweater + T-shirt).

Fig. S11: Reconstructed images and detection results with different types of clothing material. The leather cloth impedes MMW, resulting in the inability or misleading to identify hidden targets. The wool cloth may affect the detection of MMW-absorbing targets such as EPM.

Fig. S12: Reconstructed images and detection results with different clothing thicknesses. The proposed scheme is not sensitive to the thicknesses of MMW-penetrable clothes.

Body shape

We tested several cases, including heights ranging from 160cm to 189cm and weights ranging from 45kg to 95kg. The reconstruction and detection results are shown in Fig. S13. The imaging coverage area of our system is set at around 185cm in height and 100cm in width, which can be validated by the reconstruction results. Even in the case where subjects have a height exceeding 185cm, the target hidden in the clothes can also be detected. The proposed system can successfully image and detect the concealed targets of different people with common body shapes. We aim to gather an even more expansive dataset that encompasses a greater variety of body shapes to further promote the applicability of the proposed system across a wider range of real-world use cases.

Fig. S13: Reconstructed images and detection results in cases of various body shapes with heights ranging from 160cm to 189cm and weights ranging from 45kg to 95kg. The proposed scheme can deal with subjects with various body shapes.

Subject position

We conducted a series of experiments to reveal the performance boundary in cases where the

subject is positioned off-center or rotates away from the central position. The space for the human movement was expanded to 115cm (azimuth) \times 80cm (range). Due to the inherent width and thickness of the human body, the feasible range for left and right movement of the subject is ± 40 cm, and the range for forward and backward movement is ± 20 cm. Besides, we tested situations with the subject facing forward (0°) and at angular deviations of 10° , 20° , 30° , and 45° . The numerical evaluations of offset forward/backward/to the left/to the right are shown in Fig. S14. The reconstruction and detection results are shown in Fig. S15 (offset to the right), Fig. S16 (offset forward & backward), and Fig. S17 (subject rotation). We can draw the main conclusions as follows:

- The image quality reconstructed by the random and statistically optimized arrays will decrease when the subject deviates from the central position (Fig. S14). In general, the larger the deviation, the lower the quality.
- However, a few examples may not follow the above rules. This is mainly because the bigger the deviation, the interference of the random array would also deviate, leading to greater sidelobe separation from the primary body. Consequently, some portions of the subject may appear dim, and PSNR may become better.
- As shown in Fig. S15 and Fig. S16, when the subject deviates from the central position, the detectable range is:
 - 25% sampling ratio: forward 10cm, backward 20cm, left and right 40cm;
 - 10% sampling ratio: forward 10cm, backward 20cm, left and right 15cm.
- When the subject's body rotates, the detectable range is within 20° for both 25% and 10% sampling ratios (Fig. S17). In actual use, when it is greater than 20° , the body will show an obvious rotation, and the operator should promptly remind the person being tested to adjust their posture.

Fig. S14: Reconstruction results of the random and statistically optimized sparse arrays in cases where the subject being tested is offset from the central position for inspection.

Fig. S15: Reconstructed images and detection results in cases where the subject is offset to the right from the central position.

Fig. S16: Reconstructed images and detection results in cases where the subject is offset forward or backward from the central position.

Fig. S17: Reconstructed images and detection results in cases where the subject faces different directions. We tested situations with the subject facing forward (0°) and at angular deviations of 10°, 20°, 30°, and 45°.

Target position and status

The utilized YOLO network has demonstrated its effectiveness in handling variations in targets' positions and statuses across natural images [46]. When we migrate it to the MMW image scenario, the network also exhibits robustness to these variations. As shown in Fig. S18, the object can be successfully detected when hidden within the detectable range of the human body. The detection network is robust to changes in object rotation and open-closed status of knives, as shown in Fig. S19.”

Fig. S18: Reconstructed images and detection results in cases of different object positions. The detection network is robust to changes in object positions.

Fig. S19: Reconstructed images and detection results in cases of different object orientations and open-closed statuses. The detection network is robust to changes in the rotation and open-closed status of objects.

[Reviewer 3]

<Comment 1>

The cost of the proposed method is low, which is very important for the security check. Please give more description about the cost of the proposed method compared with the scanning method for example. How many elements are used for the 10% sparse array? A good reconstruction with low sampling ratio data is usually hard, which is depending on the array size. Please analyze the relationship between the array size and the performance of the proposed method.

<Response>

Thank you for your inquiries. The cost-effectiveness of the proposed method, particularly its relevance to security checks, deserves elucidation. 7998 elements are required for 10% sparse array. Our fully sampled array adheres to the Nyquist sampling criterion, with inter-element spacing slightly below the maximum distance mandated by the criterion. Consequently, this full array exhibits relatively small redundancy. Generally speaking, a large array size leads to better imaging performance. We present here a detailed comparison with traditional scanning methods and the relationship between array size and method performance.

Comparison with mechanically scanning arrays: In contrast to the mechanical scanning approach with 1-D linear antenna arrays [X3, X4], [S30] with 860 transmit and receive elements in total (equivalent to 430 antennas), the fully electronically scanned optimized array requires a substantial 7998 elements with 10% sparse configuration. Though with a lower hardware cost, the scanning scheme would lead to distorted images if the subjects shake or jiggle during the echo collection, while our scheme shows the anti-shaking property. In addition, the fully electronic system is promising to achieve high-throughput real-time detection. We have included the relevant revision in the supplement as follows:

Location: Supplement Note 1.1

“However, it is imperative to note that mechanically scanning linear array systems are challenged by the time-consuming scanning procedure, rendering them unsuitable for one-shot imaging. In practice, it is challenging to ensure complete steadiness, especially in cases involving slight movements, such as body swaying or shaking, particularly among elderly individuals. Such movements can significantly degrade image quality (see Supplement Note 5). Given that state-of-the-art

detection and classification methods, including deep neural networks, often rely on precise object edge and feature detection within images [14], blurred images with distorted object features can substantially compromise detection accuracy. Additionally, even slight subject movements during data acquisition lead to blurred images [13], thus compromising the accuracy of subsequent concealed object detection algorithms [14], [S23]. ”

Location: Supplement Note 5

“Comparison with mechanical scanning arrays

The mechanical scanning method avoids the high cost associated with full-array imaging by utilizing a linear array for scanning. However, this scanning process also encounters certain issues, such as scanning time and jiggle problems. In particular, the jiggle caused by human movement during the scanning process greatly affects the accuracy of imaging and target recognition.

As shown in Fig. S5, when the subject experiences a jiggle during scanning ($\sim 2s$), it becomes almost impossible to achieve accurate imaging and detection. Similarly, if there is displacement of body parts during the scanning process, the imaging and detection of the displaced area will be affected. On the other hand, our electronic scanning framework offers extremely short imaging time ($\sim 0.17s$), and even with motion, it has minimal impact on imaging and target recognition.

Fig. S5: The reconstruction and detection results when the subject experiences a jiggle. The electronic scanning system can accurately image and detect the concealed target, while the mechanical scanning system fails.

Relationship between array size and performance: Generally, a more sparse array corresponds to a more degraded reconstruction and detection accuracy. Combining both the statistically sparse array and the untrained learning reconstruction, we can achieve the same spatial resolution

(5mm) of 25% sampling ratio as that of the full array, and a 6mm resolution of 10% sampling ratio (Fig. 5). We have conducted a series of experiments of reconstruction and detection under 5%, 10%, and 25% sampling ratios. Considering both reconstruction and detection accuracy as shown in Fig. X1, 10% is the minimum accuracy we can currently achieve. Under a 5% sampling ratio, both reconstruction and detection accuracy decrease further, making detection unfeasible in many instances.

Figure X1: Comparisons of the untrained learning reconstruction and detection results under 5%, 10%, and 25% sampling ratios.

We have included the relevant revisions in the main text and supplement as follows:

Location: Main text, Figure 5

Fig. 5: The MMW imaging prototype. **a**, The working system for human security inspection. **b**, The block diagram of the transceiver system. **c**, The resolution target and full-sampled imaging results of a resolution target. Groups 4, 5, 6, and 7 correspond to 4mm, 5mm, 6mm, and 7mm resolutions. **d**, The untrained reconstructions of random and statistically optimized sparse arrays. The resolutions of these arrays are 10% random > 7mm, 10% statistically optimized 6mm, 25% random 6mm, 25% statistically optimized 5mm, and full array 5mm.

“Relationship between the sparse array topology and imaging results

Fig. S4: Configuration of the planar sparse array for wideband near-field imaging.

This section establishes a connection between the sparse array topology and the resulting images. We first elucidate the relationship between array size and its influence on imaging results, drawing inspiration from the work of Wang et al. [25]. Consider a wideband monostatic sparse array characterized by arbitrary element positions, as illustrated in Fig. S4. These positions are sampled from a 2D dense (full) array comprising N elements. In the context of an imaging scenario featuring Q scatterers and relying on the Born approximation [S74], the expression for the received scattered field of the dense array can be represented as per Kay’s fundamentals [S75]:

$$s(k, \mathbf{r}') = \sum_{q=1}^Q \sigma_q \frac{e^{-j2k\|\mathbf{r}'-\mathbf{r}_q\|}}{16\pi^2 \|\mathbf{r}'-\mathbf{r}_q\|^2}, \quad (\text{S9})$$

where \mathbf{r}' signifies the element position within the dense array, \mathbf{r}_q and σ_q denote the position and scattering coefficient, respectively, of the q th target. The wavenumber k is expressed as $k = \frac{2\pi f}{c}$. The matrix representation of Eq. (S9) is articulated as follows:

$$\mathbf{s}_k = \mathbf{A}_k \boldsymbol{\sigma}, \quad (\text{S10})$$

where

$$\mathbf{s}_k = [s(k, \mathbf{r}'_1), s(k, \mathbf{r}'_2), \dots, s(k, \mathbf{r}'_N)]^T, \quad (\text{S11})$$

$$\mathbf{A}_k = \begin{bmatrix} \frac{e^{-j2k\|\mathbf{r}'_1-\mathbf{r}_1\|}}{16\pi^2\|\mathbf{r}'_1-\mathbf{r}_1\|^2} & \cdots & \frac{e^{-j2k\|\mathbf{r}'_1-\mathbf{r}_Q\|}}{16\pi^2\|\mathbf{r}'_1-\mathbf{r}_Q\|^2} \\ \vdots & \ddots & \vdots \\ \frac{e^{-j2k\|\mathbf{r}'_N-\mathbf{r}_1\|}}{16\pi^2\|\mathbf{r}'_N-\mathbf{r}_1\|^2} & \cdots & \frac{e^{-j2k\|\mathbf{r}'_N-\mathbf{r}_Q\|}}{16\pi^2\|\mathbf{r}'_N-\mathbf{r}_Q\|^2} \end{bmatrix}_{N \times N_Q}, \quad (\text{S12})$$

$$\boldsymbol{\sigma} = [\sigma_1, \sigma_2, \dots, \sigma_Q]^T. \quad (\text{S13})$$

The sparse array can be derived through subsampling from \mathbf{s}_k and can be represented as follows:

$$\mathbf{s}'_k = \boldsymbol{\Psi} \mathbf{s}_k, \quad (\text{S14})$$

where

$$\mathbf{s}'_k = [s'(k, \mathbf{r}'_1), s'(k, \mathbf{r}'_2), \dots, s'(k, \mathbf{r}'_N)]^T, \quad (\text{S15})$$

$$\boldsymbol{\Psi} = \begin{bmatrix} w_1 & & & & \\ & w_2 & & & \\ & & \ddots & & \\ & & & & w_N \end{bmatrix}_{N \times N}, \quad (\text{S16})$$

The matrix $\boldsymbol{\Psi}$ is predominantly a diagonal matrix with the majority of its diagonal elements set to zero. Assuming that there are a total of N' sampling points distributed across the imaging region, the imaging result corresponding to k due to the array aperture can be expressed as follows:

$$\mathbf{I}_k = \boldsymbol{\Phi} \mathbf{s}'_k, \quad (\text{S17})$$

where $\boldsymbol{\Phi}$ represents the imaging operator, which can encompass methods such as Back Projection (BP) [15], RMA, Compressive Sensing with Conjugate Gradient (CS-CG) [36], and the proposed untrained learning technique, among others. The vector \mathbf{I}_k is defined as:

$$\mathbf{I}_k = [I(k, \mathbf{r}''_1), I(k, \mathbf{r}''_2), \dots, I(k, \mathbf{r}''_{N'})]^T, \quad (\text{S18})$$

Here, $\mathbf{r}_{N'}''$ represents the position of the N' th imaging pixel. Given that Ψ is a diagonal matrix, and \mathbf{s}_k is a vector, we can rewrite Eq. (S17) as:

$$\mathbf{I}_k = \Phi \mathbf{S}_k \mathbf{w}, \quad (\text{S19})$$

where

$$\mathbf{I}_k = \begin{bmatrix} s(k, \mathbf{r}'_1) & & & & \\ & s(k, \mathbf{r}'_2) & & & \\ & & \ddots & & \\ & & & \ddots & \\ & & & & s(k, \mathbf{r}'_N) \end{bmatrix}_{N \times N}, \quad (\text{S20})$$

$$\mathbf{w} = [w_1, w_2, \dots, w_N]^T. \quad (\text{S21})$$

The sub-imaging results across all wavenumbers in the bandwidth need to be integrated to generate the final image:

$$\mathbf{I} = \mathbf{B} \mathbf{w}, \quad (\text{S22})$$

where

$$\mathbf{I} = \sum_k \mathbf{I}_k, \quad (\text{S23})$$

$$\mathbf{B} = \sum_k \Phi \mathbf{S}_k. \quad (\text{S24})$$

To summarize, regardless of the specific imaging method employed, it is evident that the design of sparse arrays has a direct impact on the resulting images, as explicitly demonstrated in Eq. (S22). By integrating RMA with Eq. (S22), and using a reference array (which can be a full array), we can derive the averaged imaging result \mathbf{I}_{ref} , serving as the baseline by consolidating all the echoes. Subsequently, we can determine the propagation operator that quantifies the loss incurred by arbitrary imaging methods:

$$L_{\text{info}} = \chi \cdot \|\mathbf{I}_{\text{ref}} - \mathbf{B} \mathbf{w}\|_2 \quad (\text{S25})$$

where χ signifies the image quality indicator, which can encompass metrics like RMSE, PSNR, SSIM, and others. This allows us to readily obtain the averaged propagation matrix or operator L_{info} , encapsulating the statistical significance for the specific subject under investigation.

In practice, we utilize the imaging result of a particular method I for quantitative assessment of information loss by comparing it with the referenced result I_{ref} . To facilitate this assessment, we employ straightforward metrics, specifically the RMSE, PSNR, and SSIM. These criteria provide a robust foundation for establishing a relationship between the quality of reconstruction results and spatial redundancy, a pivotal aspect of our research. Table 1, Supplement Note 6, and Supplement Note 7 demonstrate the superior performance of the reported statistically optimized sparse array.

Our full array comprises 79980 elements, and existing methods [S10, S30, S34, S35, S38, S39, S76–S78] struggle to address the challenges posed by arrays with an enormous number of elements. Hence, we propose a statistically sparse array design strategy, rooted in the principle of ranking the combination of magnitude and inverse phase gradient of echoes. This strategy effectively tailors the sparse array to meet the requirements of security-check scenarios.”

References:

- [X3] Sheen, D., McMakin, D., Hall, T.: Near-field three-dimensional radar imaging techniques and applications. *Applied Optics* 49(19), 83–93 (2010)
- [X4] Wang, S., Li, S., Zhao, G., Amin, M.G.: Efficient wavenumber domain processing for near-field imaging with polyline arrays. *IEEE Transactions on Microwave Theory and Techniques* 70(10), 4544–4555 (2022)
- [S30] Yang, B., Zhuge, X., Yarovoy, A., Ligthart, L.: UWB MIMO antenna array topology design using PSO for through dress near-field imaging. In: *Eur. Microw. Conf.*, pp. 1620–1623 (2008). IEEE
- [14] Kupyn, O., Budzan, V., Mykhailych, M., Mishkin, D., Matas, J.: DeblurGAN: Blind motion deblurring using conditional adversarial networks. In: *IEEE/CVF Conf. Comput. Vis. Pattern Recogn.*, pp. 8183–8192 (2018)
- [13] Liu, H., Wang, S., Jing, H., Li, S., Zhao, G., Sun, H.: Millimeter-wave image deblurring via cycle-consistent adversarial network. *Electronics* 12(3), 741 (2023)
- [S23] Tao, X., Gao, H., Shen, X., Wang, J., Jia, J.: Scale-recurrent network for deep image deblur-

ring. In: CVPR, pp. 8174–8182 (2018)

[25] Wang, S., Li, S., Ren, B., Miao, K., Zhao, G., Sun, H.: Convex optimization-based design of sparse arrays for 3-d near-field imaging. *IEEE Sensors Journal* (2023)

[S74] Morse, P.M., Feshbach, H.: *Methods of theoretical physics*. *Amer. J. Phys.* 22(6), 410–413 (1954)

[S75] Kay, S.M.: *Fundamentals of Statistical Signal Processing: Estimation Theory*. Prentice-Hall, Inc., New Jersey (1993)

[15] Desai, M.D., Jenkins, W.K.: Convolution backprojection image reconstruction for spotlight mode synthetic aperture radar. *IEEE Trans. Image Process* 1(4), 505–517 (1992)

[36] Li, S., Zhao, G., Sun, H., Amin, M.: Compressive sensing imaging of 3-d object by a holographic algorithm. *IEEE Trans. Antennas Propag.* 66(12), 7295–7304 (2018)

[S10] Tan, K., Chen, X.: Fast 3-D image reconstruction on nonregular UWB sparse MIMO planar array using scaling techniques. *IEEE Trans. Microw. Theory Techn.* 69(1), 222–234 (2020)

[S34] Tan, K., Wu, S., Wang, Y., Ye, S., Chen, J., Fang, G.: A novel two-dimensional sparse MIMO array topology for UWB short-range imaging. *IEEE Antennas Wireless Propag. Lett.* 15, 702–705 (2015)

[S35] Tan, K., Wu, S., Wang, Y., Ye, S., Chen, J., Liu, X., Fang, G., Yan, S.: On sparse MIMO planar array topology optimization for UWB near-field high-resolution imaging. *IEEE Trans. Antennas Propag.* 65(2), 989–994 (2016)

[S38] An, Q., Hoorfar, A., Lv, H., Wang, J.: Task-specific sparse MIMO array design for twri using multi-objective CMA-ES. In: *General Assem. Scientific Symp. Int. Union Radio Sci.*, pp. 1–4 (2021). IEEE

[S39] Wang, S., Li, S., Hoorfar, A., Miao, K., Zhao, G., Sun, H.: Compressive sensing based sparse MIMO array synthesis for wideband near-field millimeter-wave imaging. *IEEE Trans. Aerosp. Electron. Syst.* (2023)

[S76] Valdes, B., Allan, G., YR-Vaqueiro, Y.A., Mantzavinos, S., Nickerson, M., et al.: Sparse array optimization using simulated annealing and 1065 compressed sensing for near-field millimeter wave imaging. *IEEE Trans. Antennas Propag.* 62(4), 1716–1722 (2014)

[S77] Napier, P.J., Thompson, A.R., Ekers, R.D.: The very large array: Design and performance of a modern synthesis radio telescope. *Proc. IEEE* 71(11), 1295–1320 (1983)

[S78] Schwartz, J.L., Steinberg, B.D.: Ultrasparse, ultrawideband arrays. *IEEE Trans. Ultrason. Ferroelect. Freq. Control* 45(2), 376–393 (1998)

<Comment 2>

The time cost should also be considered for the security check application. 30s is a very long time if the proposed method cannot be improved efficiently. There are several steps during the security check, such as detection, data processing, imaging and target recognition, etc. A time cost table is suggested to show how the time is used. The potential analysis of the improvement in the time cost for the proposed method is better given.

<Response>

Thanks for the reviewer’s comment. We develop a complete MMW-based security check workflow including sparse echo acquisition, untrained learning reconstruction, and detection. The running time of each step is summarized in Tab. S7. The most time-consuming step is to reconstruct 3D scenes. The original running time of 30 seconds per 100 iterations was measured on an RTX 3090 with the FP32 precision network. After enhancing efficiency (utilizing RTX 4090 + BF16) without compromising accuracy, the runtime per 100 iterations stands at approximately 10.4 seconds. Through analysis and experimentation, we have determined that utilizing the advanced H100 GPU can reduce this process to 0.78 seconds, fully meeting the high-throughput demands of MMW security imaging. Here we list the revision of the relevant experiments and analysis in the supplement as follows:

Location: Supplement Note 13

“In addition, the consumer GPU RTX 4090, used for our reconstruction exhibits a significant disparity in computational power compared to professional GPUs. As illustrated in Fig. S25 a, the half-precision computational power of professional GPU NVIDIA H100 is around 12 times bigger than our RTX 4090. Due to the limitations of the available devices, we do not have access to the most advanced H100 currently. Instead, we have implemented our algorithm on various GPUs including RTX 3060, RTX 2080Ti, RTX 3090, A40, and RTX 4090. Figure S25 b demonstrates the fact that the computational power is approximately inversely proportional to running time. On

the basis of our calculations across various GPUs, this relationship can be fitted as

$$y = \frac{1544.1}{x} \quad (R^2 = 0.9889), \quad (\text{S30})$$

where x denotes the computational power (unit: TFLOPS), and y is the running time per 100 iterations (unit: second). Using this approximate model, we estimate that the required time for H100 is less than 1.6s. The computational power of GPUs has been rapidly increasing in recent years (Fig. S25 a). Despite the roughness of this approximation model, we would like to emphasize that it is highly likely that a computing platform capable of high-throughput imaging, which satisfies our algorithm, is emerging in the present or near future. Therefore, we have provided a promising framework for high-throughput MMW security imaging and detection in this work.

Table S7: Running time of each step during the MMW-based security check. Here we consider the number of iterations in the reconstruction process to be 200.

GPU	Reconstruction	Detection	Total Time
RTX 4090	20.8s	9ms	< 21s
H100 (prediction)	1.56s	< 1ms	< 1.6s

a. Recent advancements of GPU

b. Running time w.r.t Computational power

Fig. S25: The improvement in computational power facilitates the rapid calculation of our algorithm. **a.** The development in computational power of typical NVIDIA GPUs in recent years. **b.** The relationship between the running time per 100 iterations and GPU computational power. The horizontal axis is represented as the reciprocal axis. The computational time of RTX 3060, RTX 2080 Ti, RTX 3090, A40, and RTX 4090 exhibits an approximate inverse proportionality to computational power, which allows us to estimate the computation time per 100 iterations for H100 to be 0.78s.

<Comment 3>

In Section 2.2, the verification platforms are different for the traditional method and the proposed method. Please analyze the computing power directly. It seems the proposed method needs more computing power if the parallel computing is not used.

<Response>

Thanks for your comments. We have included the computing power analysis in the revised supplement. Even though our algorithm is not the least complex, our validation confirms that using consumer-grade RTX 4090 GPU allows us to achieve reconstruction in 10.4 seconds per 100 iterations, and with professional-grade H100 GPU, we can achieve reconstruction in 0.78 seconds per 100 iterations. Further, our algorithm exhibits the highest accuracy and commendable robustness among the compared methods, including compressive sensing and deep learning methods (Fig. 2, Tab. 1, and Supplement Note 8). Although our algorithm may not have the lowest complexity, it is promising to achieve accurate and high-throughput MMW imaging-based security inspection.

Location: Supplement Note 13

“For convolutional neural networks, such as the one mentioned in this work, the main computational burden arises from the convolutional layers. Therefore, we will now attempt to approximate the complexity of the complex neural network reported by utilizing the complexity of the convolutional layers. As shown in Fig. S24, one single convolutional layer with bias has $C_i K^2 H W C_o$ multiplications and $C_i K^2 H W C_o$ additions, where (H, W) denotes the size of output, C_i and C_o are the channels of input x and output y , and K is the size of kernels w . The complex convolutional layer is composed of two real-valued convolutional layers and two addition operations for output features, as illustrated in Eq. (6) in the main text. Thus, the number of Floating Point Operations (FLOPs) of a complex convolutional layer is denoted as

$$(4C_i K^2 + 2) H W C_o. \quad (\text{S27})$$

Fig. S24: The workflow of a typical convolutional layer.

When performing the gradient-based optimization for the proposed network, updating a convolutional layer requires calculating the gradient of input x_{grad} and kernels w_{grad} ,

$$\begin{aligned} x_{grad} &= \text{TransConv}(y_{grad}, w), \\ w_{grad} &= \text{Conv}(y_{grad}, x). \end{aligned} \quad (\text{S28})$$

For the proposed network, the size (H, W) of output features remains unchanged. We can simplify

the time complexity of computing these gradients for a complex convolutional layer as

$$\begin{aligned} x_{grad} &: (4C_oK^2 + 2)HWC_i, \\ w_{grad} &: (4C_oHW + 2)K^2C_i. \end{aligned} \tag{S29}$$

The time complexity for updating a complex convolutional layer via the gradient descent algorithm is $8C_iK^2HWC_o + 2HWC_i + 2K^2C_i$.

We note that the complexity of other operations in CCN (including ReLU, BN, and the loss function) is much lower than that of the convolutional layers. So we can approximate the total time complexity of CCN by the complexity of these convolutional layers. The proposed CCN consists of seven complex convolutional layers. The input size is $N_x \times N_z \times N_f$, where $N_x = 430$, $N_z = 186$, and $N_f = 50$. The kernel size K is 3. The number of kernels in the hidden convolutional layers C is 128. Following Eqs. (S27, S29), we summarize the forward and backward complexity of each layer in Tab. S5. The complexity of one iteration to update CCN is the summation of all the complexities in Tab. S5. Ignoring minor terms, the total FLOPs of one iteration can be expressed as $24CK^2N_xN_zN_f + 60C^2K^2N_xN_z$.

Table S5: Time complexity of each complex convolutional layer in CCN.

Step		Complexity (FLOPs)
Forward	layer #1	$(4N_fK^2 + 2)N_xN_zC$
	layer #2-#6	$(4CK^2 + 2)N_xN_zC$
	layer #7	$(4CK^2 + 2)N_xN_zN_f$
Backward	layer #1	$8CK^2N_xN_zN_f + 2N_xN_zN_f + 2K^2N_f$
	layer #2-#6	$8C^2K^2N_xN_z + 2N_xN_zC + 2K^2C$
	layer #7	$8CK^2N_xN_zN_f + 2N_xN_zC + 2K^2C$

We summarize the complexity of all the methods we tested in the main text in Tab. S6. Compared with CS-CG, The ADMM converges with fewer iterations, thus its running time is less. Note that the complexity of the untrained method is a theoretical upper limit without any optimization techniques applied. In practical usage, we can effectively reduce the network complexity through quantization, parallel computing, and other methods, allowing the successful reconstruction within an acceptable computation time. As shown in Fig. S22 a, The network gradually converged be-

tween 100 to 200 iterations and took less than 21s on an NVIDIA RTX 4090 GPU.

Table S6: Comparison of the time complexity of different algorithms. N_x, N_z : Number of antennas along azimuth and height direction, respectively. N_f : Number of frequencies. N_{iter} : Number of iterations. C : Kernel number of CCN hidden layers. K : Kernel size.

Algorithm	Complexity (FLOPs)	Running time
RMA	$N_x N_z N_f (20 + 5 \log N_x N_z + 5 \log N_x N_z N_f)$	2s (2 × Intel E5-2687W)
CS-CG	$N_{\text{iter}} N_x N_z N_f (8 N_f + 10 \log N_x N_z)$	610s (2 × Intel E5-2687W)
ADMM	$N_{\text{iter}} N_x N_z N_f (62 + 16 N_f + 20 \log N_x N_z)$	66s (2 × Intel E5-2687W)
Untrained	$N_{\text{iter}} (24 C K^2 N_x N_z N_f + 60 C^2 K^2 N_x N_z)$	< 21s (NVIDIA RTX 4090)

<Comment 4>

In Section 2.3, there are five kinds of common targets in the network training. Are the rotation direction, the opening and closing status, and the spatial attitude considered? How many validation samples are used? How about the generalization ability of the proposed method?

<Response>

Thank you for the reviewer’s suggestions. The test dataset contains 200 different samples. Besides, we conducted more than 300 sets of additional experiments to reveal the applicability and generalization of the reported system. These experiments explored reconstruction and detection accuracy under various conditions, including clothing material and thickness, subject position, body shape, target shape, position and status, etc. Based on our findings, we conclude that:

- **Clothing:** We tested various materials (cotton, synthetic fiber, blended fabric, feather, and wool) and thicknesses (jeans, jacket, sweater, down jacket, and down jacket + sweater + T-shirt) of clothing, as shown in Fig. S11 and Fig. S12, respectively. Our system performs well in all cases except when leather and wool clothes are involved. MMW is hard to penetrate leather items, which obstructs the reconstruction and detection of concealed targets. Wool is somewhat MMW-penetrable. However, it interferes with the detection of MMW-absorbing materials such as explosive powdered material (EPM). Therefore, we recommend operators request subjects remove wool, leather, and fur clothing during MMW security screening.
- **Body shape:** According to the system settings and the reconstruction results, the imaging coverage area is around 185cm in height and 100cm in width. We tested several cases, in-

cluding heights ranging from 160cm to 189cm and weights ranging from 45kg to 95kg. The reconstruction and detection results are shown in Fig. S13. The proposed system can successfully image and detect the concealed targets of different people with common body shapes.

- **Subject position:** We tested the subjects standing offset (forward, backward, left, right) and rotating from the central position. To establish a baseline for comparison, we utilized the imaging result obtained with the subject in the central position as the benchmark. The numerical results in Fig. S14 indicate that the reconstruction quality is degrading as these deviations become larger. As shown in Fig. S15, Fig. S16, and Fig. S17, the detectable ranges of the proposed statistically optimized arrays corresponding to these deviations are:
 - 25% sampling ratio: forward 10cm, backward 20cm, left and right 40cm, rotation 20°;
 - 10% sampling ratio: forward 10cm, backward 20cm, left and right 15cm, rotation 20°.

In practice, we can position subjects within the detectable range to ensure successful detection.

- **Target shape, position, and status:** Five different kinds of objects are involved as targets, including EPM, knife, phone, gun, and wrench (Fig. 5 c). The length of these targets ranges from 7 to 15 cm, while their width varies from 2 to 11 cm. We tested various target objects in different positions and statuses. The object can be successfully imaged and detected when hidden within the detectable range of the human body (Fig. S16). The detection network is robust to changes in object rotation and the open-closed status of knives (Fig. S18).
- **Spatial resolution:** We tested the spatial resolution of both random and statistically optimized sparse arrays. The reconstruction results are shown in Fig. 5, indicating that the statistically optimized sparse array exhibits a higher spatial resolution than the random array under the same sampling ratio. The spatial resolution of the full array is 5mm. The statistically optimized arrays, coupled with the untrained learning reconstruction, attain 6mm resolution at a 10% sampling ratio and 5mm at a 25% sampling ratio.

We have added the detailed descriptions and relevant revisions as follows:

Location: Main text, Section 2.4, Line 274 - 317

“Applicability and generalization

We have conducted extensive experiments to assess our system’s applicability and generalization in reconstruction and detection, encompassing clothing, subject positioning, body shape, target positioning and status, etc. The detailed reconstruction and detection results can be found in Supplement Note 10. We summarize the main conclusions as follows.

Clothing: As shown in Fig. 4 a and Supplement Note 10.1, common materials such as cotton, synthetic fibers, and blended fabrics are penetrable by MMW, meaning that wearing clothes made of these materials does not affect the accuracy of reconstruction and detection. Woolen products, to some extent, may allow MMW penetration, but potentially affect the detection of MMW-absorbing targets. MMW is hard to penetrate leather items, which obstructs the reconstruction and detection of concealed targets. Thicker garments made from materials penetrable by MMWs (like sweaters, down jackets, etc.), or layering multiple garments (e.g., T-shirt + sweater + down jacket), do not hinder reconstruction and detection processes. Hence, we advise operators to instruct individuals undergoing scans to remove clothing composed of leather, wool, or fur materials.

Body shape: The imaging coverage area of our system is set at around 185cm in height and 100cm in width. We tested multiple subjects ranging from 160cm to 189cm in height and from 45kg to 95kg in weight. We experimentally validated that the proposed system can successfully image and detect the concealed targets of different people with common body shapes (Supplement Note 10.2).

Subject position: Variations in the orientation of the subject have an impact on the detection accuracy. We adopt the orientation of the subject facing and parallel to the antenna array as the reference. When the targets are hidden on the subject’s chest, the detectable range extends to a left and right rotation of 20° under both 10% and 25% sampling ratios (Fig. 4 b, Supplement Note 10.3). The detectable ranges for forward, backward, and lateral movements are as follows: 10cm for forward movements, 20cm for backward movements, and 40cm (25% sampling ratio) / 15cm (10% sampling ratio) for lateral movements, as depicted in Fig. 4 c and Supplement Note 10.3. In practice, we can position subjects within the detectable range to ensure successful detection.

Target shape, position and status: Our prototype is designed for detecting multiple classes of concealed targets, including EPM, knives, phones, guns, and wrenches (Fig. 3 c). The length of these targets ranges from 7 to 15 cm, while their width varies from 2 to 11 cm. The employed

YOLOv8 network has proven its effectiveness in handling variations in targets' positions and statuses across natural images [46]. In addition, we have conducted a series of experiments (Supplement Note 10.4) to verify that within the detectable range, regardless of the position, rotation angle, or open/closed status of the targets, the detection network is capable of successfully detecting hidden targets.

Fig. 4 Applicability and generalization of the system. **a**, MMW-penetrable clothing includes commonly used materials such as cotton, synthetic fiber, and blended fabric. While wool does offer some degree of MMW penetration, it can potentially impact detection accuracy. MMW is hard to penetrate leather items. It does not affect the reconstructed image quality or detection accuracy in cases of common layering of multiple MMW-penetrated clothes. **b**, The detectable range under subject rotation. The detectable range is between 20° to the left and right, with the human body facing the array as the reference. **c**, The reconstruction robustness (PSNR/dB) and detectable range under varying subject positions. We show the difference between the PSNR values at offset positions and the PSNR value at the original position. The red box indicates the range within which hidden targets can be detected by the detection network. More details are in Supplement Note 10.

Location: Supplement Note 10

“Additional details of applicability and generalization

We conducted a series of experiments to reveal the applicability and generalization of the system. The experiments involved spatial resolution, clothing, body shape, subject position, target position and status, and element assembly error. The reference images are reconstructed from full-sampled echoes by RMA, while the sparsely sampled images are reconstructed by the untrained method.

Clothing material and thickness

As shown in Fig. S11, we can detect hidden objects under cotton, synthetic fiber, and blended fabric clothes. While other materials exhibit good reconstruction quality and facilitate concealed object identification, MMW is hard to penetrate leather items, which obstructs the reconstruction and detection of concealed targets. Wool, on the other hand, is somewhat MMW-penetrable. However, it interferes with the detection of MMW-absorbing materials such as EPM. So we recommend operators request the subject take off wool, leather, and fur clothing when performing MMW security checking. Further, we tested the reconstruction and detection results in cases of various clothing thicknesses, as shown in Fig. S12. For MMW-penetrated clothing materials, it does not affect the reconstructed image quality and detection accuracy whether the subject is wearing a single jacket or the common layering of multiple thick garments (Down jacket + sweater + T-shirt).

Fig. S11: Reconstructed images and detection results with different types of clothing material. The leather cloth impedes MMW, resulting in the inability or misleading to identify hidden targets. The wool cloth may affect the detection of MMW-absorbing targets such as EPM.

Fig. S12: Reconstructed images and detection results with different clothing thicknesses. The proposed scheme is not sensitive to the thicknesses of MMW-penetrable clothes.

Body shape

We tested several cases, including heights ranging from 160cm to 189cm and weights ranging from 45kg to 95kg. The reconstruction and detection results are shown in Fig. S13. The imaging coverage area of our system is set at around 185cm in height and 100cm in width, which can be validated by the reconstruction results. Even in the case where subjects have a height exceeding 185cm, the target hidden in the clothes can also be detected. The proposed system can successfully image and detect the concealed targets of different people with common body shapes. We aim to gather an even more expansive dataset that encompasses a greater variety of body shapes to further promote the applicability of the proposed system across a wider range of real-world use cases.

Fig. S13: Reconstructed images and detection results in cases of various body shapes with heights ranging from 160cm to 189cm and weights ranging from 45kg to 95kg. The proposed scheme can deal with subjects with various body shapes.

Subject position

We conducted a series of experiments to reveal the performance boundary in cases where the

subject is positioned off-center or rotates away from the central position. The space for the human movement was expanded to 115cm (azimuth) \times 80cm (range). Due to the inherent width and thickness of the human body, the feasible range for left and right movement of the subject is ± 40 cm, and the range for forward and backward movement is ± 20 cm. Besides, we tested situations with the subject facing forward (0°) and at angular deviations of 10° , 20° , 30° , and 45° . The numerical evaluations of offset forward/backward/to the left/to the right are shown in Fig. S14. The reconstruction and detection results are shown in Fig. S15 (offset to the right), Fig. S16 (offset forward & backward), and Fig. S17 (subject rotation). We can draw the main conclusions as follows:

- The image quality reconstructed by the random and statistically optimized arrays will decrease when the subject deviates from the central position (Fig. S14). In general, the larger the deviation, the lower the quality.
- However, a few examples may not follow the above rules. This is mainly because the bigger the deviation, the interference of the random array would also deviate, leading to greater sidelobe separation from the primary body. Consequently, some portions of the subject may appear dim, and PSNR may become better.
- As shown in Fig. S15 and Fig. S16, when the subject deviates from the central position, the detectable range is:
 - 25% sampling ratio: forward 10cm, backward 20cm, left and right 40cm;
 - 10% sampling ratio: forward 10cm, backward 20cm, left and right 15cm.
- When the subject's body rotates, the detectable range is within 20° for both 25% and 10% sampling ratios (Fig. S17). In actual use, when it is greater than 20° , the body will show an obvious rotation, and the operator should promptly remind the person being tested to adjust their posture.

Fig. S14: Reconstruction results of the random and statistically optimized sparse arrays in cases where the subject being tested is offset from the central position for inspection.

Fig. S15: Reconstructed images and detection results in cases where the subject is offset to the right from the central position.

Fig. S16: Reconstructed images and detection results in cases where the subject is offset forward or backward from the central position.

Fig. S17: Reconstructed images and detection results in cases where the subject faces different directions. We tested situations with the subject facing forward (0°) and at angular deviations of 10°, 20°, 30°, and 45°.

Target position and status

The utilized YOLO network has demonstrated its effectiveness in handling variations in targets' positions and statuses across natural images [46]. When we migrate it to the MMW image scenario, the network also exhibits robustness to these variations. As shown in Fig. S18, the object can be successfully detected when hidden within the detectable range of the human body. The detection network is robust to changes in object rotation and open-closed status of knives, as shown in Fig. S19.”

Fig. S18: Reconstructed images and detection results in cases of different object positions. The detection network is robust to changes in object positions.

Fig. S19: Reconstructed images and detection results in cases of different object orientations and open-closed statuses. The detection network is robust to changes in the rotation and open-closed status of objects.

References:

[46] Jocher, G., Chaurasia, A., Qiu, J.: YOLO by Ultralytics. <https://github.com/ultralytics>

[S81] Feko, A.: Altair engineering, Inc., www.altairhyperworks.com/feko (2018)

<Comment 5>

Is there any improvement of the untrained reconstruction technique in the proposed method? What is the contribution in addition to the usage of the untrained reconstruction technique?

<Response>

Thanks for your comment. To our knowledge, we, for the first time, introduce the untrained learning-based reconstruction into 3D MMW imaging. To accommodate the complex character-

istics inherent to MMW imaging modalities, we have integrated complex-valued convolutional networks (CCN) into the untrained learning framework for reconstruction. Further, we have explored a wide variety of network optimization techniques from various perspectives, including network scale, quantization, and regularization. Through experiments as depicted in Fig. S22, we have established the optimal optimization strategy by balancing accuracy and efficiency. This refined methodology involves a 5-res-block network scale, complex-valued total variation (CTV) regularization, and BF16 quantization, which presents a pathway toward the efficient integration of complex-valued untrained reconstruction frameworks within high-throughput MMW security imaging pipelines. We have included the relevant statement in the revised supplement, which is provided below:

Location: Supplement Note 11

“Additional details of the untrained learning reconstruction

To our knowledge, we first introduce the untrained learning-based reconstruction into 3D MMW imaging. To accommodate the complex characteristics inherent to MMW imaging modalities, we have integrated Complex-valued Convolutional Networks (CCNs) into the untrained learning framework for reconstruction. Here, we list some contributions to improve the performance of the untrained reconstruction technique in MMW imaging.

- **Network scale:** We have conducted a series of experiments to explore the optimal network scale that balances accuracy and complexity. CCN contains the fixed input and output convolution layers, as well as multiple hidden Res-blocks. Changing Res-block Numbers (RN) can vary the network scale. In general, a larger scale represents higher accuracy, but also a longer running time, as depicted in Fig. S22 **a** and **b**. It can be observed that with fewer iterations, when RN=5, CCN achieves a balance between efficiency and reconstruction accuracy (Fig. S22 **c**). Therefore, we set the network size to RN=5.
- **Regularization:** The proposed Complex TV (CTV) regularization can achieve higher reconstruction accuracy and more stable convergence. The computational complexity of CTV is very low, so whether there is CTV or not almost has no impact on running time.
- **Quantization:** We observed that compared to traditional quantization of Floating Point 32-bit (FP32), quantization of BFloat 16-bit (BF16) can improve efficiency by 100% (Fig. S22

f) without affecting precision (Fig. S22 e). When using quantization with a lower bitwidth than BF16, the network may fail to converge. Therefore, we adopted BF16 quantization.

Fig. S22: Analysis of the reconstruction network. **a** and **b** show the accuracy and efficiency under different Res-block Numbers (RN). **c**, The accuracy w.r.t. RN at 100 and 200 iterations. **d**, Comparison of reconstruction accuracy with and without CTV regularization. **e** and **f** show the accuracy and efficiency of BF16 and FP32 networks.

<Comment 6>

How about the transmit and receive isolation of the detection system shown in Fig.4?

<Response>

Thank you for your insightful question. The transmit and receive isolation is 40 dB. Here we added the relevant description in the revised supplement as follows:

Location: Supplement Note 15

“Discussion about transmit and receive antenna isolation and the calibration procedure

The element spacing between two adjacent transmit (receive) antennas in our system is 10mm, while the spacing between the nearest transmit and receive antennas is 18.9mm. In general, when the spacing between two adjacent antennas falls below half of the wavelength, the isolation between them tends to deteriorate, potentially leading to a significant coupling effect. In the context of the detection system illustrated in Fig. 5 of the main text, the working frequency’s wavelength is approximately 10mm, which is nearly half of the spacing between adjacent transmit and receive antennas. Consequently, the coupling effect among elements is not pronounced, and we can assert that the transmit and receive isolation is adequately maintained, which is practically 40 dB.

Furthermore, to mitigate the impact of coupling among elements and address channel inconsistencies, we employ the following calibration methods:

1. Placement of a Metal Plate: A metal plate is positioned in parallel to the antenna array.
2. Theoretical Reference Echo: We establish the theoretical reference echo of the metal plate

as

$$S_{\text{theory}}(k, y) = e^{-jkr}, \quad (\text{S36})$$

where k denotes the wavenumber, and r signifies the distance between the metal plate and the antenna array.

3. Calibration Factor: The calibration factor is computed as

$$S_{\text{cali}}(k, y) = \frac{S_{\text{theory}}(k, y)}{S_{\text{metal}}(k, z)}, \quad (\text{S37})$$

where $S_{\text{metal}}(k, z)$ represents the echo obtained from the metal plate.

4. Echo Calibration in Application: For practical echo calibration, we employ the equation

$$S_{\text{calied}}(k, y) = [S_{\text{mea}}(k, y) - S_{\text{air}}(k)] \cdot S_{\text{cali}}(k, y), \quad (\text{S38})$$

where $S_{\text{mea}}(k, y)$ denotes the measured echo, and $S_{\text{air}}(k)$ corresponds to the echo generated in the air.

By following this calibration procedure, we effectively minimize the coupling effects among the transmit and receive antennas while simultaneously reducing channel inconsistencies.”

<Comment 7>

The resolution of the proposed imaging system is good. Please analyze how much position error can be tolerant during the data detection.

<Response>

Thank you for your inquiry. We consider two kinds of position errors, one for the subject position error and the other for the assembly position error of array elements.

Subject position: We tested the subjects standing offset (forward, backward, left, right) and rotating from the central position. To establish a baseline for comparison, we utilized the imaging result obtained with the subject in the central position as the benchmark. The numerical results in Fig. S14 indicate that the reconstruction quality is degrading as these deviations become larger. As shown in Fig. S15, Fig. S16, and Fig. S17, the detectable ranges of the proposed statistically optimized arrays corresponding to these deviations are:

- 25% sampling ratio: forward 10cm, backward 20cm, left and right 40cm, rotation 20°;
- 10% sampling ratio: forward 10cm, backward 20cm, left and right 15cm, rotation 20°.

In practice, we can position subjects within the detectable range to ensure successful detection.

Assembly position error of array elements: We tested various element position errors by utilizing FEKO software [S81]. The images reconstructed by the untrained learning method exhibit slight distortion under large element position errors, while RMA and CS-CG yield distorted images. The proposed untrained learning method exhibits robustness in the face of varying element position errors.

Location: Supplement Note 10.3

“Subject position

We conducted a series of experiments to reveal the performance boundary in cases where the subject is positioned off-center or rotates away from the central position. The space for the human movement was expanded to 115cm (azimuth) \times 80cm (range). Due to the inherent width and thickness of the human body, the feasible range for left and right movement of the subject is ± 40 cm, and the range for forward and backward movement is ± 20 cm. Besides, we tested situations with the subject facing forward (0°) and at angular deviations of 10° , 20° , 30° , and 45° . The numerical evaluations of offset forward/backward/to the left/to the right are shown in Fig. S14. The reconstruction and detection results are shown in Fig. S15 (offset to the right), Fig. S16 (offset forward & backward), and Fig. S17 (subject rotation). We can draw the main conclusions as follows:

- The image quality reconstructed by the random and statistically optimized arrays will decrease when the subject deviates from the central position (Fig. S14). In general, the larger the deviation, the lower the quality.
- However, a few examples may not follow the above rules. This is mainly because the bigger the deviation, the interference of the random array would also deviate, leading to greater sidelobe separation from the primary body. Consequently, some portions of the subject may appear dim, and PSNR may become better.
- As shown in Fig. S15 and Fig. S16, when the subject deviates from the central position, the detectable range is:
 - 25% sampling ratio: forward 10cm, backward 20cm, left and right 40cm;
 - 10% sampling ratio: forward 10cm, backward 20cm, left and right 15cm.
- When the subject’s body rotates, the detectable range is within 20° for both 25% and 10% sampling ratios (Fig. S17). In actual use, when it is greater than 20° , the body will show an obvious rotation, and the operator should promptly remind the person being tested to adjust their posture.”

Fig. S14: Reconstruction results of the random and statistically optimized sparse arrays in cases where the subject being tested is offset from the central position for inspection.

Fig. S15: Reconstructed images and detection results in cases where the subject is offset to the right from the central position.

Fig. S16: Reconstructed images and detection results in cases where the subject is offset forward or backward from the central position.

Fig. S17: Reconstructed images and detection results in cases where the subject faces different directions. We tested situations with the subject facing forward (0°) and at angular deviations of 10°, 20°, 30°, and 45°.

Location: Supplement Note 10.5

“Assembly position error of array elements

To gain insights into element position error, we have employed full-wave electromagnetic simulation utilizing FEKO software [S81], wherein the physical optics method is applied for expediency. The imaging target is a resolution board with some hollow stripes. We have configured a square array, the parameters of which are detailed in Tab. S4. The original element spacing of the full array (D) is 5mm. We introduced a random deviation to the array element positions, characterized by a standard deviation $V \times D$. We tested both the full array and sparse array (sampling ratio is 25%), as shown in Fig. S20. The reconstructed images by untrained learning are shown in Fig. S21, while presuming that the elements are correctly positioned. We also present the reference imaging results of the full arrays by RMA under the same element position errors.

The imaging results obtained through the proposed untrained learning method exhibit gradual and slight distortion as the element position error increases. Specifically, the amplitude exhibits minor fluctuations when V reaches 50%. Remarkably, under the same element position error, the imaging results achieved through the proposed untrained learning method closely resemble those produced by full arrays. Even if $V = 100\%$, we can still distinguish the texture and structure of the target. In contrast, the other methods applied to the same sparse arrays yield distorted images. Consequently, we can conclude that the proposed untrained learning method exhibits robustness in the face of varying element position errors.”

Table S4: FEKO simulation parameters for 2D sparse array

Parameters	Values
Imaging distance (R_0)	0.6 m
Start frequency	30 GHz
Stop frequency	35 GHz
Number of frequency steps	51
Number of antennas of the full array	128×128
Element spacing of the original full array (D)	5 mm
Sampling ratio of the sparse array	25%
Azimuth/Height resolution	5 mm
Range resolution	30 mm
Standard deviation of element position error	$V \times D, V = 0\% \rightarrow 100\%$

**Fig. S20:** FEKO simulation arrays of various element position variance ratios ($V = 0\%$, 50% , and 100%) for the full array (first row) and the sparse array (second row).

Fig. S21: Reconstructions of the FEKO simulations with diverse element position variance ratios. The first, second, and third columns correspond to the reconstructed images of RMA, CS-CG, and untrained learning under a 25% sampling ratio. The fourth column (reference) images are reconstructed by RMA from full-sampled echoes. The element position variance ratios (V) correspond to 0%, 30%, 50%, 80%, and 100%. The proposed untrained learning method stands as the most robust approach for mitigating element position error among the aforementioned techniques.

References:

[S81] Feko, A.: Altair engineering, Inc., www.altairhyperworks.com/feko (2018)

REVIEWER COMMENTS

Reviewer #1 (Remarks to the Author):

I thank the authors for their responses. Most of the concerns in the first round review are well addressed, but two major remains.

1. This concern of the influence of clothing, position, body shapes is shared among all the reviewers. The performance of the system under different configuration of these factors are discussed and the examples shown in Figures during rebuttal demonstrate the influence of these factors qualitatively. I appreciate these results and analyses. However, the quantitative results and metrics are not provided. As the paper is based on learning and statistical methods, I think only some visual examples are not convincing enough.

2. I still have some concerns on the effectiveness of the statistically optimized arrays compared to regular grid arrays. Sure, regular grid arrays will result in repeat ghost patterns as shown in Fig. S6. Random and statistically optimized arrays also have other noise patterns. If they can be denoised and reconstructed, the repeat ghost pattern may be removed from deep learning method as well? The repeat pattern should be easier to be removed than the noise from the randomly sampled arrays using learning based methods with collected paired data? Correct me if I am wrong. From Table 1 in the response, the deep learning method works very well in statically optimized arrays, almost similar to the untrained method proposed in the paper while it does work that well (though still quite well) for random arrays. I guess the reason is that "random" noise resulting from the random arrays makes it difficult to learn while the statistically optimized arrays result in noise with certain distribution other than the "random" noise. Given this assumption, regular grid arrays + deep learning method may achieve better reconstruction results than statistically optimized arrays + untrained and statistically optimized arrays + deep learning.

Fig. S6 did provide examples for CS-CSG and untrained for the regular array, untrained method works in mitigating the repeat pattern to some extent. How about other method? what's the quantitative comparison of the methods mentioned in the paper.

Reviewer #2 (Remarks to the Author):

The answers provided by the authors are remarkably comprehensive and seem to address absolutely all my comments.

I believe that the paper now represents a useful source of information for the scientific community in the development of complete body-imaging systems.

As far as scientific added-value is concerned, the combination of sub-sampling techniques and machine learning reconstruction methods still seems to me a little weak by the standards of a journal like Nature Communications (only considering novelty). However, considering the quality of the results provided by the complete system and the efforts that this research team has had to make to reach this level of development, I believe that this contribution deserves publication.

Reviewer #3 (Remarks to the Author):

My issues has been addressed. I have no further comments.

Dear Editor and Reviewers:

Thank you very much for your valuable comments on our manuscript NCOMMS-23-29557A. Specifically, we appreciate the acceptance advice from Reviewer #2 and Reviewer #3, and we have also supplemented additional experiments according to Reviewer #1's minor comments. For convenience, we highlight the revisions with blue font in the revised manuscript. The point-to-point responses to the comments are listed below.

Sincerely,

All the authors

[Reviewer 1]

<Comment 1>

I thank the authors for their responses. Most of the concerns in the first round review are well addressed.

<Response>

Thanks for the reviewer's recognition of this manuscript and the positive comments.

<Comment 2>

This concern of the influence of clothing, position, body shapes is shared among all the reviewers. The performance of the system under different configuration of these factors are discussed and the examples shown in Figures during rebuttal demonstrate the influence of these factors qualitatively. I appreciate these results and analyses. However, the quantitative results and metrics are not provided. As the paper is based on learning and statistical methods, I think only some visual examples are not convincing enough.

<Response>

We appreciate the reviewer's suggestion. We have incorporated the quantitative metrics, including Root Mean Square Error (RMSE), Peak Signal-to-Noise Ratio (PSNR), and Structural Similarity Index (SSIM) to provide a more comprehensive evaluation under various scenarios. The relevant revisions are provided as follows:

Location: Supplement Note 10

“We conducted a series of experiments to reveal the applicability and generalization of the system. The experiments involved spatial resolution, clothing, body shape, subject position, target position and status, and element assembly error. The reference images are reconstructed from full-sampled echoes by RMA, while the sparsely sampled images are reconstructed by the untrained method. We evaluated the RMSE, PSNR, and SSIM values of under-sampled reconstructions with the above-mentioned references as benchmarks.

10.1 Clothing material and thickness

As shown in Fig. S13, we can detect hidden objects under cotton, synthetic fiber, and blended fabric clothes. While other materials exhibit good reconstruction quality and facilitate concealed object identification, leather impedes MMW, resulting in the inability to identify hidden targets.

Wool, on the other hand, is somewhat MMW-penetrable. However, it interferes with the detection of MMW-absorbing materials such as EPM. So we recommend operators request the subject take off wool, leather, and fur clothing when performing MMW security checking. Further, we tested the reconstruction and detection results in cases of various clothing thicknesses, as shown in Fig. S14. For MMW-penetrated clothing materials, it does not affect the reconstructed image quality and detection accuracy whether the subject is wearing a single jacket or the common layering of multiple thick garments (Down jacket + sweater + T-shirt).

Fig. S13: Reconstructed images and detection results with different types of clothing material. The leather cloth impedes MMW, resulting in the inability or misleading to identify hidden targets. The wool cloth may affect the detection of MMW-absorbing targets such as EPM.

Fig. S14: Reconstructed images and detection results with different clothing thicknesses. The proposed scheme is not sensitive to the thicknesses of MMW-penetrable clothes.

10.2 Body shape

We tested several cases, including heights ranging from 160cm to 189cm and weights ranging from 45kg to 95kg. The reconstruction and detection results are shown in Fig. S15. The imaging coverage area of our system is set at around 185cm in height and 100cm in width, which can be validated by the reconstruction results. Even in the case where subjects have a height exceeding 185cm, the target hidden in the clothes can also be detected. The proposed system can successfully image and detect the concealed targets of different people with common body shapes. We aim to gather an even more expansive dataset that encompasses a greater variety of body shapes to further promote the applicability of the proposed system across a wider range of real-world use cases.

Fig. S15: Reconstructed images and detection results in cases of various body shapes with heights ranging from 160cm to 189cm and weights ranging from 45kg to 95kg. The proposed scheme can deal with subjects with various body shapes.

10.3 Subject position

We conducted a series of experiments to reveal the performance boundary in cases where the subject is positioned off-center or rotates away from the central position. The space for the human movement was expanded to 115cm (azimuth) \times 80cm (range). Due to the inherent width and thickness of the human body, the feasible range for left and right movement of the subject is ± 40 cm, and the range for forward and backward movement is ± 20 cm. Besides, we tested situations with the subject facing forward (0°) and at angular deviations of 10° , 20° , 30° , and 45° . The numerical evaluations of offset forward/backward/to the left/to the right are shown in Fig. S16. The reconstruction and detection results are shown in Fig. S17 (offset to the right), Fig. S18 (offset forward & backward), and Fig. S19 (subject rotation). We can draw the main conclusions as follows:

- In the context of PSNR, the image quality reconstructed by the random and statistically optimized arrays will decrease when the subject deviates from the central position (Fig. S16). In general, the larger the deviation, the lower the quality.
- However, a few examples may not follow the above rules. This is mainly because the bigger the deviation, the interference of the random array would also deviate, leading to greater sidelobe separation from the primary body. Consequently, some portions of the subject may appear dim, and PSNR may become better.
- SSIM typically decreases as the subject deviates from the central position. In the scenarios of the subject moving right and backward, SSIM initially decreases and then increases, as shown in Fig. S17 and Fig. S18. This is because when the deviation exceeds a certain range, the expansion of the black areas in the image without the human body, as the deviation increases, may lead to an increase in SSIM.
- As shown in Fig. S17 and Fig. S18, when the subject deviates from the central position, the detectable range is:
 - 25% sampling ratio: forward 10cm, backward 20cm, left and right 40cm;
 - 10% sampling ratio: forward 10cm, backward 20cm, left and right 15cm.
- When the subject's body rotates, the detectable range is within 20° for both 25% and 10% sampling ratios (Fig. S19). In actual use, when it is greater than 20° , the body will show an

obvious rotation, and the operator should promptly remind the person being tested to adjust their posture.

Fig. S16: Reconstruction results of the random and statistically optimized sparse arrays in cases where the subject being tested is offset from the central position for inspection. The case of the subject standing at the central position was treated as the benchmark.

Fig. S17: Reconstructed images and detection results in cases where the subject is offset to the right from the central position. The case of the subject standing at the central position was treated as the benchmark.

Fig. S18: Reconstructed images and detection results in cases where the subject is offset forward or backward from the central position. The case of the subject standing at the central position was treated as the benchmark.

Fig. S19: Reconstructed images and detection results in cases where the subject faces different directions. We tested situations with the subject facing forward (0°) and at angular deviations of 10°, 20°, 30°, and 45°.

10.4 Target position and status

Generally, the concealed targets are relatively small compared to the human body. Thus, the variations in targets' positions and statuses lead to a relatively minor influence in RMSE, PSNR, and SSIM compared to the variations in the subject's posture and position. As for detection, the utilized YOLO network has demonstrated its effectiveness in handling variations in targets' positions and statuses across natural images [46]. When we migrate it to the MMW image scenario, the network also exhibits robustness to these variations. As shown in Fig. S20, the object can be successfully detected when hidden within the detectable range of the human body. The detection network is robust to changes in object rotation and open-closed status of knives, as shown in Fig. S21."

Fig. S20: Reconstructed images and detection results in cases of different object positions. The detection network is robust to changes in object positions.

Fig. S21: Reconstructed images and detection results in cases of different object orientations and open-closed statuses. The detection network is robust to changes in the rotation and open-closed status of objects.

<Comment 3>

I still have some concerns on the effectiveness of the statistically optimized arrays compared to regular grid arrays. Sure, regular grid arrays will result in repeat ghost patterns as shown in Fig. S6. Random and statistically optimized arrays also have other noise patterns. If they can be de-noised and reconstructed, the repeat ghost pattern may be removed from deep learning method as well? The repeat pattern should be easier to be removed than the noise from the randomly sampled arrays using learning based methods with collected paired data? Correct me if I am wrong. From Table 1 in the response, the deep learning method works very well in statically optimized arrays, almost similar to the untrained method proposed in the paper while it does work that well (though still quite well) for random arrays. I guess the reason is that “random” noise resulting from the

random arrays makes it difficult to learn while the statistically optimized arrays result in noise with certain distribution other than the "random" noise. Given this assumption, regular grid arrays + deep learning method may achieve better reconstruction results than statistically optimized arrays + untrained and statistically optimized arrays + deep learning.

Fig. S6 did provide examples for CS-CG and untrained for the regular array, untrained method works in mitigating the repeat pattern to some extent. How about other methods? What's the quantitative comparison of the methods mentioned in the paper?

<Response>

We appreciate the reviewer's careful review and suggestions. In security check scenarios, the ghost would not appear independently, but would alias with the target. Consequently, attempts to eliminate these ghost artifacts may lead to image deterioration. Typically, relevant works focus on designing optimized arrays to avoid the appearance of ghosts, rather than remove them [R1-R3, 21, 23].

In considering your discussion regarding the impact of random arrays introducing more noise compared to regular arrays, it's important to note that regular arrays can lead to the emergence of periodical grating lobes [R4, R5]. These create significant overlaps between the actual target and the ghost artifacts in such low sampling ratios (10% and 25%). On the contrary, random arrays tend to suppress grating lobes. The statistically optimized sparse array navigates a trade-off between the introduction of noise and grating lobes.

We conducted a series of experiments to compare different arrays (random, regular, and statistically optimized) with various reconstruction algorithms including RMA, CS-CG, ADMM, deep learning (DL), and untrained learning. We applied the network architecture as in Supplement Note 7 for DL reconstruction. We trained the networks with various array forms following the configurations in Supplement Note 7 to ensure convergence. The evaluations on the test dataset (200 echoes in total) in Tab. S4 and Fig. S6 indicate that statistically optimized arrays outperform random and regular arrays across all the reconstruction algorithms. Furthermore, visual comparisons in Fig. S7 and Fig. S8 reveal that even with DL and untrained learning algorithms, regular arrays tend to introduce ghost artifacts. DL is also hard to reconstruct clear targets under the presence of severe ghost artifacts. The statistically optimized array achieves a delicate balance between noise reduction and minimizing grating lobes, making it outperform the random and regular configurations we

tested.

We have included the relevant revisions as follows:

Location: Supplement Note 6

“This section conducts a comparative analysis between the proposed statistically optimized sparse array and other prevalent sparse array configurations, including both random and regular topologies. The random sparse array is generated through direct random sampling from a full array scheme [S39], [25]. Random arrays can introduce sidelobes [S39], manifesting as high noise levels.

Regular grids, characterized by uniform element spacing, hold inherent characteristics that warrant exploration. One crucial factor to consider is that uniform element spacing, when not aligned with the Nyquist sampling criterion, often leads to the emergence of grating lobe artifacts within the reconstructed images. These artifacts manifest as ghost targets and can potentially compromise the overall image quality. This phenomenon is particularly evident when the element interval restricts the imaging region, contributing to the presence of grating lobes. The azimuth and height arrays encompass transmitting and receiving apertures, respectively, and their unambiguous distance in the azimuth direction is influenced by the array’s design parameters.

As articulated in ref. [19], when $\frac{\lambda_c}{\Delta_x} \leq \Theta$, the azimuth unambiguous distance D_x is given by the formula:

$$D_x = \frac{\lambda_c \cdot R_0}{\Delta_x}, \quad (\text{S26})$$

where D_x represents the azimuth unambiguous distance, λ_c signifies the wavelength of the center operating frequency, R_0 pertains to the imaging distance, Δ_x denotes the element interval, and Θ corresponds to the antenna’s beamwidth. Conversely, when $\frac{\lambda_c}{\Delta_x} > \Theta$, the presence of ghost targets becomes less conspicuous; however, it comes at the cost of reduced azimuth resolution. The manifestation of ghost targets is intricately tied to the interplay between λ_c and Δ_x relative to the antenna beamwidth Θ .

Considering the parameters used in our experiments, the azimuth unambiguous distance, D_x , exceeds 0.75 meters. This is sufficient for effective security checks in near-field imaging scenarios. However, when half of the azimuth elements are removed (sampling ratio of 25%), reducing the element spacing, D_x drops to 0.375 meters. This reduction signifies a severe aliasing phenomenon, wherein ghost effects become prominent. Therefore, the ambiguity phenomenon is

intricately linked to the specific parameter settings.

Furthermore, we conducted a series of experiments to compare different arrays (random, regular, and statistically optimized) with various reconstruction algorithms, including RMA, CS-CG, ADMM, DL, and untrained learning. The evaluations on the test dataset (200 echoes in total) in Tab. S4 and Fig. S6 indicate that statistically optimized arrays outperform random and regular arrays across all reconstruction algorithms. Additionally, visual comparisons in Fig. S7 and Fig. S8 validate the aforementioned analysis, indicating that random arrays tend to introduce high levels of noise, while regular arrays tend to produce ghost artifacts. Regular arrays introduce grating lobes, which interfere with primary body regions, resulting in image distortion and yielding inferior metrics in most cases. The statistically optimized arrays achieve a delicate balance between noise reduction and minimizing grating lobes, thus outperforming the random and regular arrays we tested.”

Table S4: The quantitative comparisons of reconstructed data with different sampling strategies (regular, random, and statistically optimized) and reconstruction methods (RMA, CS-CG, ADMM, DL, and Untrained). We used all the 200 echoes of the test dataset for reconstruction. The values specified to the right of the forward slash indicate the variances.

Method	SR	10%			25%		
	Metrics	RMSE	PSNR	SSIM	RMSE	PSNR	SSIM
RMA	Regular	67.05/8.13	11.67/1.06	0.15/0.07	65.83/6.27	11.80/0.84	0.27/0.07
	Random	63.66/9.63	12.13/1.32	0.12/0.05	53.33/8.49	13.75/1.36	0.19/0.06
	Statistically optimized	49.92/5.97	14.26/1.00	0.32/0.05	33.89/4.86	17.65/1.25	0.45/0.07
CS-CG	Regular	67.85/16.07	11.73/1.96	0.03/0.011	66.70/9.54	11.73/1.20	0.06/0.02
	Random	59.10/13.49	12.91/1.72	0.05/0.05	44.90/7.39	15.22/1.25	0.16/0.07
	Statistically optimized	50.93/9.75	14.28/1.52	0.44/0.14	35.94/3.28	17.07/0.77	0.60/0.07
ADMM	Regular	46.09/6.02	14.93/1.13	0.12/0.04	47.65/7.83	14.69/1.44	0.46/0.07
	Random	43.58/5.65	15.41/1.09	0.13/0.04	47.15/7.96	14.90/1.45	0.56/0.06
	Statistically optimized	40.82/5.67	16.11/1.00	0.24/0.03	40.91/7.22	16.09/1.54	0.61/0.05
DL	Regular	67.05/14.72	11.80/1.84	0.16/0.16	59.65/12.96	12.81/1.79	0.17/0.19
	Random	46.62/8.23	14.88/1.45	0.26/0.13	35.65/10.09	17.40/2.28	0.49/0.16
	Statistically optimized	32.76/8.11	18.07/2.05	0.60/0.11	28.05/8.86	19.55/2.50	0.64/0.15
Untrained	Regular	50.84/6.22	14.07/1.07	0.47/0.09	45.65/6.15	15.02/1.13	0.46/0.09
	Random	40.05/5.84	16.17/1.23	0.53/0.06	28.23/4.20	19.30/1.24	0.59/0.06
	Statistically optimized	31.53/5.42	18.40/1.16	0.65/0.05	23.90/3.31	20.72/1.15	0.72/0.05

Fig. S6: Average PSNR of reconstructed data with different sampling strategies (regular, random, and statistically optimized) and reconstruction methods (RMA, CS-CG, ADMM, DL, and Untrained) on the test dataset. All the reconstruction methods obtained the best average PSNR on the statistically optimized array and the worst average PSNR on the regular array at both 10% and 25% sampling ratios.

Fig. S7: Reconstruction results by RMA, ADMM, CS-CG, DL, and the untrained learning technique under 25% sampling ratio of different sparse arrays including random, regular, and statistically optimized arrays. The reconstructions of the regular sparse array contain severe artifacts. The reported statistically optimized array achieves superior reconstruction performance compared to random and regular arrays.

Fig. S8: Reconstruction results by RMA, ADMM, CS-CG, DL, and the untrained learning technique under 10% sampling ratio of different sparse arrays including random, regular, and statistically optimized arrays. The reconstructions of the regular sparse array contain severe artifacts. The reported statistically optimized array achieves superior reconstruction performance compared to random and regular arrays.

Reference:

- [R1] Yang, B., Zhuge, X., Yarovoy, A., Ligthart, L.: UWB MIMO antenna array topology design using PSO for through dress near-field imaging. In: Eur. Microw. Conf., pp. 1620–1623 (2008). IEEE
- [R2] Tan, K., Wu, S., Wang, Y., Ye, S., Chen, J., Fang, G.: A novel two dimensional sparse MIMO array topology for UWB short-range imaging. *IEEE Antennas Wireless Propag. Lett.* 15, 702–705 (2015)
- [R3] Wang, S., Li, S., Hoorfar, A., Miao, K., Zhao, G., Sun, H.: Compressive sensing based sparse MIMO array synthesis for wideband near-field millimeter-wave imaging. *IEEE Trans. Aerosp. Electron. Syst.*, 59(6), 7681 - 7697 (2023)
- [R4] Davidsen R. E., Jensen J. A., Smith S W.: Two-dimensional random arrays for real time volumetric imaging. *Ultrason. Imag.*, 16(3): 143-163 (1994)
- [R5] Turnbull D H, Foster F S.: Beam steering with pulsed two dimensional transducer arrays. *IEEE Trans. Ultrason. Ferroelect. Freq. Control*, 38(4), 320–333 (1991)
- [21] Gonzalez-Valdes, B., Allan, G., Rodriguez-Vaqueiro, Y., Alvarez, Y., Mantzavinos, S., Nickerson, M., Berkowitz, B., Marti, J., Las-Heras, F., Rappaport, C.M., et al.: Sparse array optimization using simulated annealing and compressed sensing for near-field millimeter wave imaging. *IEEE Trans. Antennas Propag.* 62(4), 1716–1722 (2013)
- [23] Tan, K., Wu, S., Wang, Y., Ye, S., Chen, J., Liu, X., Fang, G., Yan, S.: On sparse MIMO planar array topology optimization for UWB near-field high-resolution imaging. *IEEE Trans. Antennas Propag.* 65(2), 989–994928 (2016)

[Reviewer 2]

<Comment 1>

Considering the quality of the results provided by the complete system and the efforts that this research team has had to make to reach this level of development, I believe that this contribution deserves publication.

<Response>

Thank you for the positive feedback. We sincerely appreciate your dedicated efforts in reviewing the manuscript.

[Reviewer 3]

<Comment 1>

My issues has been addressed. I have no further comments.

<Response>

Thanks for your positive comments. We extend our sincere appreciation for your dedicated efforts in reviewing the manuscript.

REVIEWERS' COMMENTS

Reviewer #1 (Remarks to the Author):

I thank the authors for the responses and they have addressed my concerns.